# A Finite-Sample Analysis of Payoff-Based Independent Learning in Zero-Sum Stochastic Games

**Zaiwei Chen**[1,*]**, Kaiqing Zhang**[2]**, Eric Mazumdar**[1,†]**, Asuman Ozdaglar**[3]**, Adam Wierman**[1,‡]

[1]CMS, Caltech, *`zchen458@caltech.edu`, †`mazumdar@caltech.edu`, ‡`adamw@caltech.edu`
[2]ECE & ISR, University of Maryland, College Park, `kaiqing@umd.edu`
[3]EECS, MIT, `asuman@mit.edu`

## Abstract

In this work, we study two-player zero-sum stochastic games and develop a variant of the smoothed best-response learning dynamics that combines independent learning dynamics for matrix games with the minimax value iteration for stochastic games. The resulting learning dynamics are payoff-based, convergent, rational, and symmetric between the two players. Our theoretical results present to the best of our knowledge the first last-iterate finite-sample analysis of such independent learning dynamics. To establish the results, we develop a coupled Lyapunov drift approach to capture the evolution of multiple sets of coupled and stochastic iterates, which might be of independent interest.

## 1 Introduction

Recent years have seen remarkable successes in reinforcement learning (RL) in a variety of applications, such as board games [1], autonomous driving [2], robotics [3], and city navigation [4]. A common feature of these applications is that there are *multiple* decision makers interacting with each other in a common environment. Although empirical successes have shown the potential of multi-agent reinforcement learning (MARL) [5, 6], the training of MARL agents is largely based on heuristics and parameter tuning and, therefore, is not always reliable. In particular, many practical MARL algorithms are directly extended from their single-agent counterparts and lack guarantees because of the adaptive strategies of multiple agents.

A growing literature seeks to provide theoretical insights to substantiate the empirical success of MARL and inform the design of efficient and provably convergent algorithms. Work along these lines can be broadly categorized into work on cooperative MARL where agents seek to reach a common goal [7–10], and work on competitive MARL where agents have individual (and possibly misaligned) objectives [11–22]. While some earlier work focused on providing guarantees on the asymptotic convergence, the more recent ones share an increasing interest in understanding the finite-time/sample behavior. This follows from a line of recent advances in establishing finite-sample guarantees of single-agent RL algorithms, see e.g., [23–26] and many others.

In this paper, we focus on the benchmark setting of two-player[1] zero-sum stochastic games, and develop best-response-type learning dynamics with provable finite-sample guarantees. Crucially, our learning dynamics are independent (requiring no coordination between the agents in learning) and rational (each agent will converge to the best response to the opponent if the opponent plays an (asymptotically) stationary policy [27]), and therefore capture learning in settings with multiple game-theoretic agents. Indeed, learning dynamics with self-interested agents should not enforce information communication or coordination among agents. Furthermore, we focus on the more challenging but practically relevant setting of payoff-based learning, where each agent can only

---

[1]Hereafter, we may use player and agent interchangeably.

observe their realized payoff at each stage, without observing the policy or even the action taken by the opponent. For learning dynamics with such properties, we establish to the best of our knowledge the first last-iterate finite-sample guarantees. We detail our contributions as follows.

## 1.1 Contributions

We first consider zero-sum matrix games and provide the last-iterate finite-sample guarantees for the smoothed best-response dynamics proposed in [28]. Then, we extend the algorithmic idea to the setting of stochastic games and develop an algorithm called value iteration with smoothed best-response dynamics (VI-SBR) that also enjoys last-iterate finite-sample convergence.

**Two-Player Zero-Sum Matrix Games.** We start with the smoothed best-response dynamics in [28] and establish the last-iterate finite-sample bounds when using stepsizes of various decay rates. The result implies a sample complexity of $\mathcal{O}(\epsilon^{-1})$ in terms of the last iterate to find the Nash distribution [29], which is also known as the quantal response equilibrium in the literature [30]. To our knowledge, this is the first last-iterate finite-sample result for best-response learning dynamics that are payoff-based, rational, and symmetric in zero-sum matrix games.

**Two-Player Zero-Sum Stochastic Games.** Building on the algorithmic ideas for matrix games, we develop best-response-type learning dynamics for stochastic games called VI-SBR, which uses a single trajectory of Markovian samples. Our learning dynamics consist of two loops and can be viewed as a combination of the smoothed best-response dynamics for an induced auxiliary matrix game (conducted in the inner loop) and an independent way of performing minimax value iteration (conducted in the outer loop). In particular, in the inner loop, the iterate of the outer loop, i.e., the value function, is fixed, and the players learn the approximate Nash equilibrium of an auxiliary matrix game induced by the value function; then the outer loop is updated by approximating the minimax value iteration updates for the stochastic game, with only local information.

We establish the last-iterate finite-sample bounds for VI-SBR when using both constant stepsizes and diminishing stepsizes of $\mathcal{O}(1/k)$ decay rate. To the best of our knowledge, this appears to be the first last-iterate finite-sample analysis of best-response-type independent learning dynamics that are convergent and rational for stochastic games. Most existing MARL algorithms are either symmetric across players, but not payoff-based, e.g., [31–35], or not symmetric and thus not rational, e.g., [14, 36–38], or do not have last-iterate finite-time/sample guarantees, e.g., [39, 15, 40].

## 1.2 Challenges & Techniques

The main challenge in analyzing our learning dynamics is that it maintains multiple sets of stochastic iterates and updates them in a coupled manner. To overcome this challenge, we develop a novel *coupled Lyapunov drift approach*. Specifically, we construct a Lyapunov function for each set of the stochastic iterates and establish a Lyapunov drift inequality for each. We then carefully combine the coupled Lyapunov drift inequalities to establish the finite-sample bounds. Although a more detailed analysis is provided in the appendices, we briefly give an overview of the main challenges in analyzing the payoff-based independent learning dynamics in stochastic games, as well as our techniques to overcome them.

**Time-Inhomogeneous Markovian Noise.** The fact that our learning dynamics are payoff-based presents major challenges in handling the stochastic errors in the update. In particular, due to the best-response nature of the dynamics, the behavior policy for sampling becomes time-varying. In fact, the sample trajectory used for learning forms a time-inhomogeneous Markov chain. This makes it challenging to establish finite-sample guarantees, as time-inhomogeneity prevents us from directly exploiting the uniqueness of stationary distributions and the fast mixing of Markov chains. Building on existing work [23, 24, 41, 42], we overcome this challenge by tuning the algorithm parameters (in particular, the stepsizes) and developing a refined conditioning argument.

**Non-Zero-Sum Payoffs Due to Independent Learning.** As illustrated in Section 1.1, the inner loop of VI-SBR is designed to approximately learn the Nash equilibrium of an auxiliary matrix game induced by the value functions for the two players, which we denote by $v_t^1$ and $v_t^2$, where $t$ is the iteration index of the outer loop. Importantly, the value functions $v_t^1$ and $v_t^2$ are maintained individually by players 1 and 2, and therefore do not necessarily satisfy $v_t^1 + v_t^2 = 0$ due to independent learning. As a result, the auxiliary matrix game from the inner loop does *not* necessarily

admit a zero-sum structure during learning. The error induced from such a non-zero-sum structure appears in existing work [15, 43], and was handled by designing a novel truncated Lyapunov function. However, the truncated Lyapunov function was sufficient to establish the asymptotic convergence, but did not provide the explicit rate at which the induced error goes to zero. To enable finite-sample analysis, we introduce $\|v_t^1 + v_t^2\|_\infty$ as a Lyapunov function in our coupled Lyapunov framework, which is customized to capture the behavior of the induced error from the non-zero-sum structure of the inner-loop auxiliary matrix game.

**Coupled Lyapunov Drift Inequalities.** In the existing literature of stochastic iterative algorithms [44, 24, 45, 46], when using a Lyapunov approach for finite-sample analysis, once the Lyapunov drift inequality is established, the finite-sample bound follows straightforwardly by repeatedly invoking the result. However, since our learning dynamics (for stochastic games) maintain multiple sets of stochastic iterates and update them in a coupled manner, the Lyapunov drift inequalities we establish are also highly coupled. Decoupling these Lyapunov inequalities is a major challenge. To overcome it, we develop a systematic approach for decoupling, which crucially relies on a bootstrapping argument. Specifically, we first use the Lyapunov inequalities in a direct way to establish a crude bound for each Lyapunov function. Then, we substitute the crude bounds back into the Lyapunov drift inequalities to obtain tighter bounds. See Appendix B.4 for a more detailed illustration of our decoupling procedure.

## 1.3 Related Work

Due to space constraints, here we only discuss related work in independent learning in matrix games and stochastic games, and single-agent RL. See Appendix A for a more detailed literature review.

**Independent Learning in Matrix Games.** Independent learning has been well-studied in the literature on learning in matrix games. Fictitious play (FP) [47] may be viewed as the earliest one of this kind, and its convergence analysis for the zero-sum setting is provided in [48]. Smoothed versions of FP have been developed [49, 50] to make the learning dynamics consistent [51, 52]. It was shown that the behavior of smoothed FP is captured by an ODE known as the smoothed best-response dynamics, which were also studied extensively in the literature [53]. Note that the Lyapunov function used to study the smoothed best-response dynamics is the regularized Nash gap [54, 53, 29], a variant of which is also used in our Lyapunov framework. To make the learning dynamics payoff-based, [28] developed a two time-scale variant of the smoothed best-response dynamics and established the asymptotic convergence. Moreover, no-regret learning algorithms, extensively studied in online learning, can also be used as independent learning dynamics for matrix games [55]. It is known that they are both convergent and rational by the definition in [27], and are usually implemented in a symmetric way. See [55] for a detailed introduction to no-regret learning in games.

**Independent Learning in Stochastic Games.** For stochastic games, independent and symmetric policy gradient methods have been developed in recent years, mostly for the case of Markov potential games [18, 19, 56]. The zero-sum case is more challenging since there is no single Lyapunov function to capture the learning dynamics (which is why we need to develop a coupled Lyapunov approach with multiple Lyapunov functions), such as the potential function in Markov potential games. For non-potential game settings, symmetric variants of policy gradient methods have been proposed, but have only been studied under the *full-information* setting without finite-sample guarantees [31, 32, 57, 33–35], with the exception of [58, 59]. However, the learning algorithm in [58] requires some coordination between the players when sampling, and is thus not completely independent; that in [59] is extragradient-based and needs some stage-based sampling process that also requires coordination across players. Best-response-type independent learning for stochastic games has been studied recently in [39, 15, 43, 60, 40, 61, 62], with [15, 43, 40, 61] tackling the zero-sum setting. However, only asymptotic convergence was established in these works, which motivated this work.

**Finite-Sample Analysis in Single-Agent RL.** The most related works (in single-agent RL) to our paper are those that perform finite-sample analysis for RL in infinite-horizon discounted Markov decision processes following a single trajectory of Markovian samples. See [63, 23, 41, 24–26, 64–67, 10] and the references therein. Among these works, [23, 24] established finite-sample bounds for temporal difference (TD)-learning (with linear function approximation), and [25, 64, 65] established finite-sample bounds for $Q$-learning. In both cases, the behavior policy for sampling is a *stationary* policy. For *non-stationary* behavior policies as we consider, [41] established finite-sample bounds for SARSA, an on-policy variant of $Q$-learning, and [10] provided finite-sample bounds for off-policy actor-critic, which is established based on a general framework of contractive stochastic approximation with time-inhomogeneous Markovian noise.

## 2 Zero-Sum Matrix Games

We begin by considering zero-sum matrix games. This section introduces algorithmic and technical ideas that are important for the setting of stochastic games. For $i \in \{1, 2\}$, let $\mathcal{A}^i$ be the finite action space of player $i$, and let $R_i \in \mathbb{R}^{|\mathcal{A}^i| \times |\mathcal{A}^{-i}|}$ (where $-i$ denotes the index of player $i$'s opponent) be the payoff matrix of player $i$. Note that in a zero-sum game we have $R_1 + R_2^\top = 0$. Since there are finitely many actions for each player, we assume without loss of generality that $\max_{a^1, a^2} |R_1(a^1, a^2)| \leq 1$. Furthermore, we denote $A_{\max} = \max(|\mathcal{A}^1|, |\mathcal{A}^2|)$.

The decision variables here are the policies $\pi^i \in \Delta(\mathcal{A}^i)$, $i \in \{1, 2\}$, where $\Delta(\mathcal{A}^i)$ denotes the probability simplex supported on $\mathcal{A}^i$. Given a joint policy $(\pi^1, \pi^2)$, the expected reward received by player $i$ is $\mathbb{E}_{A^i \sim \pi^i, A^{-i} \sim \pi^{-i}}[R_i(A^i, A^{-i})] = (\pi^i)^\top R_i \pi^{-i}$, where $i \in \{1, 2\}$. Both players aim to maximize their rewards against their opponents. Unlike in the single-agent setting, since the performance of player $i$'s policy depends on its opponent $-i$'s policy, there is, in general, no universal optimal policy. Instead, we use the *Nash gap* and the *regularized Nash gap* as measurements of the performance of the learning dynamics, as formally defined below.

**Definition 2.1** (Nash Gap in Matrix Games). Given a joint policy $\pi = (\pi^1, \pi^2)$, the Nash gap $\mathrm{NG}(\pi^1, \pi^2)$ is defined as $\mathrm{NG}(\pi^1, \pi^2) = \sum_{i=1,2} \max_{\hat{\pi}^i \in \Delta(\mathcal{A}^i)} (\hat{\pi}^i - \pi^i)^\top R_i \pi^{-i}$.

Note that $\mathrm{NG}(\pi^1, \pi^2) = 0$ if and only if $(\pi^1, \pi^2)$ is in a Nash equilibrium of the matrix game (which may not be unique), in which no player has the incentive to change its policy.

**Definition 2.2** (Regularized Nash Gap in Matrix Games). Given a joint policy $\pi = (\pi^1, \pi^2)$ and a constant $\tau > 0$, the entropy-regularized Nash gap $\mathrm{NG}_\tau(\pi^1, \pi^2)$ is defined as $\mathrm{NG}_\tau(\pi^1, \pi^2) = \sum_{i=1,2} \{ \max_{\hat{\pi}^i \in \Delta(\mathcal{A}^i)} (\hat{\pi}^i - \pi^i)^\top R_i \pi^{-i} + \tau \nu(\hat{\pi}^i) - \tau \nu(\pi^i) \}$, where $\nu(\cdot)$ is the entropy function defined as $\nu(\pi^i) = -\sum_{a^i \in \mathcal{A}^i} \pi^i(a^i) \log(\pi^i(a^i))$ for $i \in \{1, 2\}$.

A joint policy $(\pi^1, \pi^2)$ satisfying $\mathrm{NG}_\tau(\pi^1, \pi^2) = 0$ is called the Nash distribution [29] or the quantal response equilibrium [30], which, unlike Nash equilibria, is unique in zero-sum matrix games. As $\tau$ approaches 0, the corresponding Nash distribution approximates a Nash equilibrium [68].

### 2.1 The Learning Dynamics in Zero-Sum Matrix Games

We start by presenting in Algorithm 1 (from the perspective of player $i$, where $i \in \{1, 2\}$) the independent learning dynamics for zero-sum matrix games, which was first proposed in [28]. Given $\tau > 0$, we use $\sigma_\tau : \mathbb{R}^{|\mathcal{A}^i|} \mapsto \mathbb{R}^{|\mathcal{A}^i|}$ for the softmax function with temperature $\tau$, that is, $[\sigma_\tau(q^i)](a^i) = \exp(q^i(a^i)/\tau) / \sum_{\tilde{a}^i \in \mathcal{A}^i} \exp(q^i(\tilde{a}^i)/\tau)$ for all $a^i \in \mathcal{A}^i$, $q^i \in \mathbb{R}^{|\mathcal{A}^i|}$, and $i \in \{1, 2\}$.

---

**Algorithm 1** Independent Learning Dynamics in Zero-Sum Matrix Games

1: **Input:** Integer $K$, initializations $q_0^i = 0 \in \mathbb{R}^{|\mathcal{A}^i|}$ and $\pi_0^i = \mathrm{Unif}(\mathcal{A}^i)$.
2: **for** $k = 0, 1, \cdots, K-1$ **do**
3: $\quad \pi_{k+1}^i = \pi_k^i + \beta_k(\sigma_\tau(q_k^i) - \pi_k^i)$
4: $\quad$ Play $A_k^i \sim \pi_{k+1}^i(\cdot)$ (against $A_k^{-i}$), and receive reward $R_i(A_k^i, A_k^{-i})$
5: $\quad q_{k+1}^i(a^i) = q_k^i(a^i) + \alpha_k \mathbb{1}_{\{a^i = A_k^i\}} (R_i(A_k^i, A_k^{-i}) - q_k^i(A_k^i))$ for all $a^i \in \mathcal{A}^i$
6: **end for**

---

To make this paper self-contained, we next provide a detailed interpretation of Algorithm 1, which also motivates our algorithm for stochastic games in Section 3. At a high level, Algorithm 1 can be viewed as a discrete and smoothed variant of the best-response dynamics, where each player constructs an approximation of the best response to its opponent's policy using the $q$-function. The update for the $q$-function is in the spirit of the TD-learning algorithm in RL [69].

**The Policy Update.** To understand the update equation for the policies (cf. Algorithm 1 Line 3), consider the discrete version of the smoothed best-response dynamics:

$$\pi_{k+1}^i = \pi_k^i + \beta_k(\sigma_\tau(R_i \pi_k^{-i}) - \pi_k^i), \quad i \in \{1, 2\}. \tag{1}$$

In Eq. (1), each player updates its policy $\pi_k^i$ incrementally towards the smoothed best response to its opponent's current policy. While the dynamics in Eq. (1) provably converge for zero-sum matrix

games, see e.g., [70], implementing it requires player $i$ to compute $\sigma_\tau(R_i \pi_k^{-i})$. Note that $\sigma_\tau(R_i \pi_k^{-i})$ involves the exact knowledge of the opponent's policy and the reward matrix, both of which cannot be accessed in payoff-based independent learning. This leads to the update equation for the $q$-functions, which estimate the quantity $R_i \pi_k^{-i}$ that is needed for implementing Eq. (1).

**The $q$-Function Update.** Suppose for now that we are given a *stationary* joint policy $\pi = (\pi^1, \pi^2)$. Fix $i \in \{1, 2\}$, the problem of player $i$ estimating $R_i \pi^{-i}$ can be viewed as a *policy evaluation* problem, which is usually solved with TD-learning in RL [69]. Specifically, the two players repeatedly play the matrix game with the joint policy $\pi = (\pi^1, \pi^2)$ and produce a sequence of joint actions $\{(A_k^1, A_k^2)\}_{k \geq 0}$. Then, player $i$ forms an estimate of $R_i \pi^{-i}$ through the following iterative algorithm:

$$q_{k+1}^i(a^i) = q_k^i(a^i) + \alpha_k \mathbb{1}_{\{a^i = A_k^i\}}(R_i(A_k^i, A_k^{-i}) - q_k^i(A_k^i)), \quad \forall \, a^i \in \mathcal{A}^i, \tag{2}$$

with an arbitrary initialization $q_0^i \in \mathbb{R}^{|\mathcal{A}^i|}$, where $\alpha_k > 0$ is the stepsize. To understand Eq. (2), suppose that $q_k^i$ converges to some $\bar{q}^i$. Then Eq. (2) should be "stationary" at the limit point $\bar{q}^i$ in the sense that $\mathbb{E}_{A^i \sim \pi^i(\cdot), A^{-i} \sim \pi^{-i}(\cdot)}[\mathbb{1}_{\{a^i = A^i\}}(R_i(A^i, A^{-i}) - \bar{q}^i(A^i))] = 0$ for all $a^i \in \mathcal{A}^i$, which implies $\bar{q}^i = R_i \pi^{-i}$, as desired. Although Eq. (2) is motivated by the case when the joint policy $(\pi^1, \pi^2)$ is stationary, the joint policy $\pi_k = (\pi_k^1, \pi_k^2)$ from Eq. (1) is time-varying. A natural approach to address this issue is to make sure that the policies evolve at a *slower* time-scale compared to that of the $q$-functions, so that $\pi_k$ is close to being *stationary* from the perspectives of $q_k^i$.

*Remark.* In [28], where Algorithm 1 was first proposed, the authors require $\beta_k = o(\alpha_k)$ to establish the asymptotic convergence, making Algorithm 1 a *two time-scale* algorithm. In this work, for finite-sample analysis and easier implementation, we update $\pi_k^i$ and $q_k^i$ on a *single time scale* with only a multiplicative constant difference in their stepsizes, i.e., $\beta_k = c_{\alpha,\beta} \alpha_k$ for some $c_{\alpha,\beta} \in (0, 1)$.

## 2.2 Finite-Sample Analysis

In this section, we present the finite-sample analysis of Algorithm 1 for the convergence to the Nash distribution [29]. We consider using either constant stepsizes, i.e., $\alpha_k \equiv \alpha$ and $\beta_k \equiv \beta = c_{\alpha,\beta}\alpha$, or diminishing stepsizes with $\mathcal{O}(1/k)$ decay rate, i.e., $\alpha_k = \alpha/(k+h)$ and $\beta_k = \beta/(k+h) = c_{\alpha,\beta}\alpha/(k+h)$. Let $\ell_\tau = [(A_{\max} - 1)\exp(2/\tau) + 1]^{-1}$ and $L_\tau = \tau/\ell_\tau + A_{\max}^2/\tau$. The requirement for choosing the stepsizes is stated in the following condition.

**Condition 2.1.** When using either constant or diminishing stepsizes, we choose $\tau \leq 1$, $\alpha_0 < \frac{2}{\ell_\tau}$, $\beta_0 < \min(2, \frac{\tau}{128 A_{\max}^2})$, and $c_{\alpha,\beta} = \beta_k/\alpha_k \leq \min\left(\frac{\tau \ell_\tau^3}{32}, \frac{\ell_\tau \tau^3}{128 A_{\max}^2}, \frac{2\sqrt{2}}{L_\tau^{1/2}}\right)$.

We next state the finite-sample bounds of Algorithm 1. See Appendix C for the proof.

**Theorem 2.1.** *Suppose that both players follow the learning dynamics presented in Algorithm 1, and the stepsizes $\{\alpha_k\}$ and $\{\beta_k\}$ are chosen such that Condition 2.1 is satisfied. Then we have the following results.*

*(1) When using constant stepsizes, i.e., $\alpha_k \equiv \alpha$ and $\beta_k \equiv \beta$, we have*

$$\mathbb{E}[NG_\tau(\pi_K^1, \pi_K^2)] \leq B_{in}\left(1 - \frac{\beta}{4}\right)^K + 8L_\tau\beta + \frac{64\alpha}{c_{\alpha,\beta}},$$

*where $B_{in} = 4 + 2\tau \log(A_{\max}) + 2A_{\max}$.*

*(2) When using $\alpha_k = \alpha/(k+h)$ and $\beta_k = \beta/(k+h)$, by choosing $\beta > 4$, we have*

$$\mathbb{E}[NG_\tau(\pi_K^1, \pi_K^2)] \leq B_{in}\left(\frac{h}{K+h}\right)^{\beta/4} + \left(64eL_\tau\beta + \frac{512e\alpha}{c_{\alpha,\beta}}\right)\frac{1}{K+h}.$$

The convergence bounds in Theorem 2.1 are qualitatively consistent with the existing results on the finite-sample analysis of general stochastic approximation algorithms [44, 46, 24, 65, 23, 42]. Specifically, when using constant stepsizes, the bound consists of a geometrically decaying term (known as the optimization error) and two constant terms (known as the statistical error) that are proportional to the stepsizes. When using diminishing stepsizes with suitable hyperparameters, both the optimization error and the statistical error achieve an $\mathcal{O}(1/K)$ rate of convergence.

Although Theorem 2.1 is stated in terms of the expectation of the regularized Nash gap, it implies the mean-square convergence of the policy iterates $(\pi_k^1, \pi_k^2)$. To see this, note that the regularized Nash gap $NG_\tau(\pi^1, \pi^2)$ has a unique minimizer, i.e., the Nash distribution and is denoted by $(\pi_{*,\tau}^1, \pi_{*,\tau}^2)$. In addition, fixing $\pi^1$ (respectively, $\pi^2$), the function $NG_\tau(\pi^1, \cdot)$ (respectively, $NG_\tau(\cdot, \pi^2)$) is a $\tau$-strongly convex function with respect to $\pi^2$ (respectively, $\pi^1$). See Lemma D.7 for a proof. Therefore, by the quadratic growth property of strongly convex functions, we have

$$NG_\tau(\pi_k^1, \pi_k^2) = NG_\tau(\pi_k^1, \pi_k^2) - NG_\tau(\pi_{*,\tau}^1, \pi_k^2) + NG_\tau(\pi_{*,\tau}^1, \pi_k^2) - NG_\tau(\pi_{*,\tau}^1, \pi_{*,\tau}^2)$$
$$\geq \frac{\tau}{2}(\|\pi_k^1 - \pi_{*,\tau}^1\|_2^2 + \|\pi_k^2 - \pi_{*,\tau}^2\|_2^2).$$

As a result, up to a multiplicative constant, the convergence bound for $\mathbb{E}[NG_\tau(\pi_k^1, \pi_k^2)]$ implies a convergence bound of $\mathbb{E}[\|\pi_k^1 - \pi_{*,\tau}^1\|_2^2] + \mathbb{E}[\|\pi_k^2 - \pi_{*,\tau}^2\|_2^2]$.

Based on Theorem 2.1, we next derive the sample complexity of Algorithm 1 in the following corollary. See Appendix C.5 for the proof.

**Corollary 2.1.1.** *Given $\epsilon > 0$, to achieve $\mathbb{E}[NG_\tau(\pi_K^1, \pi_K^2)] \leq \epsilon$, the sample complexity is $\mathcal{O}(\epsilon^{-1})$.*

To the best of our knowledge, Theorem 2.1 and Corollary 2.1.1 present the first last-iterate finite-sample analysis of Algorithm 1 [28]. Importantly, with only feedback in the form of realized payoffs, we achieve a sample complexity of $\mathcal{O}(\epsilon^{-1})$ to find the Nash distribution. In general, for smooth and strongly monotone games, the lower bound for the sample complexity of payoff-based or zeroth-order algorithms is $\mathcal{O}(\epsilon^{-2})$ [71]. We have an improved $\mathcal{O}(\epsilon^{-1})$ sample complexity due to the bilinear structure of the game (up to a regularizer). In particular, with bandit feedback, the $q$-function is constructed as an efficient estimator for the marginalized payoff $R_i \pi_k^{-i}$, which can also be interpreted as the gradient. Therefore, Algorithm 1 enjoys the fast $\mathcal{O}(\epsilon^{-1})$ sample complexity that is comparable to the first-order method [72].

**The Dependence on the Temperature $\tau$.** Although our finite-sample bound enjoys the $\mathcal{O}(1/K)$ rate of convergence, the stepsize ratio $c_{\alpha,\beta}$ appears as $c_{\alpha,\beta}^{-1}$ in the bound. Since $c_{\alpha,\beta} = o(\ell_\tau)$ (cf. Condition 2.1) and $\ell_\tau$ is exponentially small in $\tau$, the finite-sample bound is actually exponentially large in $\tau^{-1}$. To illustrate this phenomenon, consider the update equation for the $q$-functions (cf. Algorithm 1 Line 5). Observe that the $q$-functions are updated asynchronously because only one component (which corresponds to the action taken at time step $k$) of the vector-valued $q_k^i$ is updated in the $k$-th iteration. Suppose that an action $a^i$ is never taken in the algorithm trajectory, which means that $q_k^i(a^i)$ is never updated during learning. Then, in general, we cannot expect the convergence of $q_k^i$ or $\pi_k^i$. Similarly, suppose that an action is rarely taken in the learning dynamics, we would expect the overall convergence rate to be slow. Therefore, the finite-sample bound should depend on the quantity $\min_{i \in \{1,2\}} \min_{0 \leq k \leq K} \min_{a^i} \pi_k^i(a^i)$, which captures the exploration abilities of Algorithm 1. Due to the exponential nature of softmax functions, the parameter $\ell_\tau$, which we establish in Lemma C.2 as a lower bound of $\min_{i \in \{1,2\}} \min_{0 \leq k \leq K} \min_{a^i} \pi_k^i(a^i)$, is also exponentially small in $\tau$. This eventually leads to the exponential dependence in $\tau^{-1}$ in the finite-sample bound.

A consequence of having such an exponential factor of $\tau^{-1}$ in the sample complexity bound is that, if we want to have convergence to a Nash equilibrium rather than to the Nash distribution, the sample complexity can be exponentially large. To see this, note that the following bound holds regarding the Nash gap and the regularized Nash gap:

$$NG(\pi^1, \pi^2) \leq NG_\tau(\pi^1, \pi^2) + 2\tau \log(A_{\max}), \quad \forall (\pi^1, \pi^2), \tag{3}$$

which, after combining with Theorem 2.1, gives the following corollary. For simplicity of presentation, we only state the result for using constant stepsizes.

**Corollary 2.1.2.** *Under the same conditions stated in Theorem 2.1 (1), we have*

$$\mathbb{E}[NG(\pi_K^1, \pi_K^2)] \leq B_{in}\left(1 - \frac{\beta}{4}\right)^K + 8L_\tau\beta + \frac{64\alpha}{c_{\alpha,\beta}} + 2\tau \log(A_{\max}). \tag{4}$$

The last term on the RHS of Eq. (4) can be viewed as the bias due to using smoothed best-response. In view of Eq. (4), to achieve $\mathbb{E}[NG(\pi_K^1, \pi_K^2)] \leq \epsilon$, we need $\tau = \mathcal{O}(\epsilon)$. Since $c_{\alpha,\beta}$ appears in the denominator of our finite-sample bound and is exponentially small in $\tau$, the overall sample complexity

for the convergence to a Nash equilibrium can be exponentially large in $\epsilon^{-1}$. In Appendix F, we conduct numerical experiments to investigate the impact of $\tau$ on this smoothing bias.

In light of the discussion before Corollary 2.1.2, the reason for such an exponentially large sample complexity for finding a Nash equilibrium is due to the limitation of using the softmax policies in smoothed best-response for exploration. We kept the softmax policy without further modification to preserve the "naturalness" of the learning dynamics, which is part of the motivation for studying independent learning in games [73]. A future direction of this work is to remove such an exponential dependence on $\tau$ by designing an improved exploration strategy.

# 3  Zero-Sum Stochastic Games

Moving to the setting of stochastic games, we consider an infinite-horizon discounted two-player zero-sum stochastic game $\mathcal{M} = (\mathcal{S}, \mathcal{A}^1, \mathcal{A}^2, p, R_1, R_2, \gamma)$, where $\mathcal{S}$ is a finite state space, $\mathcal{A}^1$ (respectively, $\mathcal{A}^2$) is a finite action space for player 1 (respectively, player 2), $p$ represents the transition probabilities, in particular, $p(s' \mid s, a^1, a^2)$ is the probability of transitioning to state $s'$ after player 1 taking action $a^1$ and player 2 taking action $a^2$ simultaneously at state $s$, $R_1 : \mathcal{S} \times \mathcal{A}^1 \times \mathcal{A}^2 \mapsto \mathbb{R}$ (respectively, $R_2 : \mathcal{S} \times \mathcal{A}^2 \times \mathcal{A}^1 \mapsto \mathbb{R}$) is player 1's (respectively, player 2's) reward function, and $\gamma \in (0, 1)$ is the discount factor. Note that we have $R_1(s, a^1, a^2) + R_2(s, a^2, a^1) = 0$ for all $(s, a^1, a^2)$. We assume without loss of generality that $\max_{s,a^1,a^2} |R_1(s, a^1, a^2)| \leq 1$, and denote $A_{\max} = \max(|\mathcal{A}^1|, |\mathcal{A}^2|)$.

Given a joint policy $\pi = (\pi^1, \pi^2)$, where $\pi^i : \mathcal{S} \mapsto \Delta(\mathcal{A}^i)$, $i \in \{1, 2\}$, we define the local $q$-function $q_\pi^i \in \mathbb{R}^{|\mathcal{S}||\mathcal{A}^i|}$ of player $i$ as $q_\pi^i(s, a^i) = \mathbb{E}_\pi \left[ \sum_{k=0}^\infty \gamma^k R_i(S_k, A_k^i, A_k^{-i}) \mid S_0 = s, A_0^i = a^i \right]$ for all $(s, a^i)$, where we use the notation $\mathbb{E}_\pi[\cdot]$ to indicate that the actions are chosen according to the joint policy $\pi$. In addition, we define the global value function $v_\pi^i \in \mathbb{R}^{|\mathcal{S}|}$ as $v_\pi^i(s) = \mathbb{E}_{A^i \sim \pi^i(\cdot|s)}[q_\pi^i(s, A^i)]$ for all $s$, and the expected value function $U^i(\pi^i, \pi^{-i}) \in \mathbb{R}$ as $U^i(\pi^i, \pi^{-i}) = \mathbb{E}_{S \sim p_o}[v_\pi^i(S)]$, where $p_o \in \Delta(\mathcal{S})$ is an arbitrary initial distribution on the states. The Nash gap in the case of stochastic games is defined in the following.

**Definition 3.1** (Nash Gap in Zero-Sum Stochastic Games). Given a joint policy $\pi = (\pi^1, \pi^2)$, the Nash gap $\mathrm{NG}(\pi^1, \pi^2)$ is defined as $\mathrm{NG}(\pi^1, \pi^2) = \sum_{i=1,2} \left( \max_{\hat{\pi}^i} U^i(\hat{\pi}^i, \pi^{-i}) - U^i(\pi^i, \pi^{-i}) \right)$.

Similar to the matrix-game setting, a joint policy $\pi = (\pi^1, \pi^2)$ satisfying $\mathrm{NG}(\pi^1, \pi^2) = 0$ is called a Nash equilibrium, which may not be unique.

**Additional Notation.** In what follows, we will frequently work with the real vectors in $\mathbb{R}^{|\mathcal{S}||\mathcal{A}^i|}$, $\mathbb{R}^{|\mathcal{S}||\mathcal{A}^{-i}|}$, and $\mathbb{R}^{|\mathcal{S}||\mathcal{A}^i||\mathcal{A}^{-i}|}$, where $i \in \{1, 2\}$. To simplify the notation, for any $x \in \mathbb{R}^{|\mathcal{S}||\mathcal{A}^i||\mathcal{A}^{-i}|}$, we use $x(s)$ to denote the $|\mathcal{A}^i| \times |\mathcal{A}^{-i}|$ matrix with the $(a^i, a^{-i})$-th entry being $x(s, a^i, a^{-i})$. For any $y \in \mathbb{R}^{|\mathcal{S}||\mathcal{A}^i|}$, we use $y(s)$ to denote the $|\mathcal{A}^i|$-dimensional vector with its $a^i$-th entry being $y(s, a^i)$.

## 3.1  Value Iteration with Smoothed Best-Response Dynamics

Our learning dynamics for stochastic games (cf. Algorithm 2) build on the dynamics for matrix games studied in Section 2.1, with the additional incorporation of minimax value iteration, a well-known approach for solving zero-sum stochastic games [74].

**Algorithmic Ideas.** To motivate the learning dynamics, we first introduce the minimax value iteration. For $i \in \{1, 2\}$, let $\mathcal{T}^i : \mathbb{R}^{|\mathcal{S}|} \mapsto \mathbb{R}^{|\mathcal{S}||\mathcal{A}^i||\mathcal{A}^{-i}|}$ be an operator defined as

$$\mathcal{T}^i(v)(s, a^i, a^{-i}) = R_i(s, a^i, a^{-i}) + \gamma \mathbb{E} \left[ v(S_1) \mid S_0 = s, A_0^i = a^i, A_0^{-i} = a^{-i} \right]$$

for all $(s, a^i, a^{-i})$ and $v \in \mathbb{R}^{|\mathcal{S}|}$. Given $X \in \mathbb{R}^{|\mathcal{A}^i| \times |\mathcal{A}^{-i}|}$, we define $val^i : \mathbb{R}^{|\mathcal{A}^i| \times |\mathcal{A}^{-i}|} \mapsto \mathbb{R}$ as

$$val^i(X) = \max_{\mu^i \in \Delta(\mathcal{A}^i)} \min_{\mu^{-i} \in \Delta(\mathcal{A}^{-i})} \{(\mu^i)^\top X \mu^{-i}\} = \min_{\mu^{-i} \in \Delta(\mathcal{A}^{-i})} \max_{\mu^i \in \Delta(\mathcal{A}^i)} \{(\mu^i)^\top X \mu^{-i}\}.$$

Then, the minimax Bellman operator $\mathcal{B}^i : \mathbb{R}^{|\mathcal{S}|} \mapsto \mathbb{R}^{|\mathcal{S}|}$ is defined as $[\mathcal{B}^i(v)](s) = val^i(\mathcal{T}^i(v)(s))$ for all $s \in \mathcal{S}$, where $\mathcal{T}^i(v)(s)$ is an $|\mathcal{A}^i| \times |\mathcal{A}^{-i}|$ matrix according to our notation. It is known that the operator $\mathcal{B}^i(\cdot)$ is a $\gamma$ – contraction mapping with respect to the $\ell_\infty$-norm [74], hence it admits a unique fixed point, which we denote by $v_*^i$.

A common approach for solving zero-sum stochastic games is to first implement the minimax value iteration $v_{t+1}^i = \mathcal{B}^i(v_t^i)$ until (approximate) convergence to $v_*^i$ [75], and then solve the matrix game $\max_{\mu^i \in \Delta(\mathcal{A}^i)} \min_{\mu^{-i} \in \Delta(\mathcal{A}^{-i})} (\mu^i)^\top \mathcal{T}^i(v_*^i)(s)\mu^{-i}$ for each state $s$ to obtain an (approximate) Nash equilibrium policy. However, implementing this algorithm requires complete knowledge of the underlying transition probabilities. Moreover, since it is an off-policy algorithm, the output is independent of the opponent's policy. Thus, it is not rational by the definition in [27]. To develop a model-free and rational learning dynamics, let us first rewrite the minimax value iteration:

$$v_{t+1}^i = \hat{v}, \text{ where } \hat{v}(s) = \max_{\mu^i \in \Delta(\mathcal{A}^i)} \min_{\mu^{-i} \in \Delta(\mathcal{A}^{-i})} (\mu^i)^\top \mathcal{T}^i(v_t^i)(s)\mu^{-i}, \ \forall \ s \in \mathcal{S}, \tag{5}$$

where $\hat{v} \in \mathbb{R}^{|\mathcal{S}|}$ is a dummy variable. In view of Eq. (5), we need to solve a matrix game with payoff matrix $\mathcal{T}^i(v_t^i)(s)$ for each state $s$ and then update the value of the game to $v_{t+1}^i(s)$. In light of Algorithm 1, we already know how to solve matrix games with independent learning. Thus, what remains to do is to combine Algorithm 1 with value iteration. This leads to Algorithm 2, which is presented from player $i$'s perspective, where $i \in \{1, 2\}$.

---

**Algorithm 2** Value Iteration with Smoothed Best-Response (VI-SBR) Dynamics

---

1: **Input:** Integers $K$ and $T$, initializations $v_0^i = 0 \in \mathbb{R}^{|\mathcal{S}|}$, $q_{0,0}^i = 0 \in \mathbb{R}^{|\mathcal{S}||\mathcal{A}^i|}$, and $\pi_{0,0}^i(\cdot|s) = \text{Unif}(\mathcal{A}^i)$ for all $s \in \mathcal{S}$.
2: **for** $t = 0, 1, \cdots, T-1$ **do**
3:     **for** $k = 0, 1, \cdots, K-1$ **do**
4:         $\pi_{t,k+1}^i(s) = \pi_{t,k}^i(s) + \beta_k(\sigma_\tau(q_{t,k}^i(s)) - \pi_{t,k}^i(s))$ for all $s \in \mathcal{S}$
5:         Play $A_k^i \sim \pi_{t,k+1}^i(\cdot|S_k)$ (against $A_k^{-i}$) and observe $S_{k+1} \sim p(\cdot \mid S_k, A_k^i, A_k^{-i})$
6:         $q_{t,k+1}^i(s,a^i) = q_{t,k}^i(s,a^i) + \alpha_k \mathbb{1}_{\{(s,a^i)=(S_k,A_k^i)\}}(R_i(S_k, A_k^i, A_k^{-i}) + \gamma v_t^i(S_{k+1}) - q_{t,k}^i(S_k, A_k^i))$ for all $(s, a^i)$
7:     **end for**
8:     $v_{t+1}^i(s) = \pi_{t,K}^i(s)^\top q_{t,K}^i(s)$ for all $s \in \mathcal{S}$
9:     $S_0 = S_K$, $q_{t+1,0}^i = q_{t,K}^i$, and $\pi_{t+1,0}^i = \pi_{t,K}^i$
10: **end for**

---

**Algorithm Details.** For each state $s$, the inner loop of Algorithm 2 is designed to solve a matrix game with payoff matrices $\mathcal{T}^1(v_t^1)(s)$ and $\mathcal{T}^2(v_t^2)(s)$ for each state $s \in \mathcal{S}$, which reduces to Algorithm 1 when (1) the stochastic game has only one state, and (2) $v_t^1 = v_t^2 = 0$. However, in general, since $v_t^1$ and $v_t^2$ are *independently* maintained by players 1 and 2, the quantity

$$\mathcal{T}^1(v_t^1)(s, a^1, a^2) + \mathcal{T}^2(v_t^2)(s, a^2, a^1) = \gamma \sum_{s'} p(s' \mid s, a^1, a^2)(v_t^1(s') + v_t^2(s'))$$

is in general *non-zero* during learning. As a result, the auxiliary matrix game (with payoff matrices $\mathcal{T}^1(v_t^1)(s)$ and $\mathcal{T}^2(v_t^2)(s)$) at state $s$ that the inner loop of Algorithm 2 is designed to solve is not necessarily a zero-sum matrix game, which presents a major challenge in the finite-sample analysis, as illustrated previously in Section 1.2.

The outer loop of Algorithm 2 is an "on-policy" variant of minimax value iteration. To see this, note that, ideally, we would synchronize $v_{t+1}^i(s)$ with $\pi_{t,K}^i(s)^\top \mathcal{T}^i(v_t^i)(s)\pi_{t,K}^{-i}(s)$, which is an approximation of $[\mathcal{B}^i(v)](s) = val^i(\mathcal{T}^i(v_t^i)(s))$ by design of our inner loop. However, player $i$ has no access to $\pi_{t,K}^{-i}$ in independent learning. Fortunately, the $q$-function $q_{t,K}^i$ is precisely constructed as an *estimate* of $\mathcal{T}^i(v_t^i)(s)\pi_{t,K}^{-i}(s)$, as illustrated in Section 2.1, which leads to the outer loop of Algorithm 2. In Algorithm 2 Line 8, we set $S_0 = S_K$ to ensure that the initial state of the next inner loop is the last state of the previous one; hence Algorithm 2 is driven by a single trajectory of Markovian samples.

## 3.2 Finite-Sample Analysis

We now state our main results, which, to the best of our knowledge, provide the first last-iterate finite-sample bound for best-response-type payoff-based independent learning dynamics in zero-sum stochastic games. Our results are based on the following assumption.

**Assumption 3.1.** There exists a joint policy $\pi_b = (\pi_b^1, \pi_b^2)$ such that the Markov chain $\{S_k\}_{k \geq 0}$ induced by $\pi_b$ is irreducible and aperiodic.

One challenge in our finite-sample analysis is that the behavior policies used for taking the actions are time-varying, due to the best-response nature of the dynamics. Most, if not all, existing finite-sample guarantees of RL algorithms under time-varying behavior policies assume that the induced Markov chain of any policy, or any policy encountered along the algorithm trajectory, is uniformly geometrically ergodic [41, 76, 77, 59, 78–80]. Assumption 3.1 is weaker, since it assumes only the existence of one policy that induces an irreducible and aperiodic Markov chain.

We consider using either constant stepsizes, i.e., $\alpha_k \equiv \alpha$ and $\beta_k \equiv \beta = c_{\alpha,\beta}\beta$, or diminishing stepsizes of $\mathcal{O}(1/k)$ decay rate, i.e., $\alpha_k = \alpha/(k+h)$ and $\beta_k = \beta/(k+h) = c_{\alpha,\beta}\alpha/(k+h)$, where $c_{\alpha,\beta} \in (0,1)$ is the stepsize ratio. In the stochastic-game setting, we redefine $\ell_\tau = [1 + (A_{\max} - 1)\exp(2/[(1-\gamma)\tau])]^{-1}$, which, analogous to the matrix-game setting, is a uniform lower bound on the entries of the policies generated by Algorithm 2 (cf. Lemma D.1). We next state our requirement for choosing the stepsizes.

**Condition 3.1.** When using either constant or diminishing stepsizes, we choose $\tau \leq 1/(1-\gamma)$ and the stepsize ratio $c_{\alpha,\beta}$ to satisfy $c_{\alpha,\beta} \leq \min\left(\frac{1}{60L_p|\mathcal{S}|A_{\max}}, \frac{c_\tau\tau(1-\gamma)^2}{34|\mathcal{S}|A_{\max}^2}, \frac{c_\tau\ell_\tau^2\tau^3(1-\gamma)^2}{144A_{\max}^2}\right)$, where $c_\tau \propto \ell_\tau$ and $L_p > 0$ are defined in Appendix B.3. In addition, when using $\alpha_k \equiv \alpha$ and $\beta_k \equiv \beta$, we require $\alpha < 1/c_\tau$ and $\beta < 1$, and when using $\alpha_k = \alpha/(k+h)$ and $\beta_k = \beta/(k+h)$, we require[2] $\beta = 4$, $\alpha > 1/c_\tau$, and $h > 1$ such that $\alpha_0 < 1/c_\tau$ and $\beta_0 < 1$.

We next state the finite-sample bound of Algorithm 2. For simplicity of presentation, we use $a \lesssim b$ to mean that there exists an *absolute* constant $c > 0$ such that $a \leq bc$.

**Theorem 3.1.** *Suppose that both players follow Algorithm 2, Assumption 3.1 is satisfied, and the stepsizes $\{\alpha_k\}$ and $\{\beta_k\}$ satisfy Condition 3.1. Then, we have the following results.*

*(1) When using constant stepsizes, there exists $z_\beta = \mathcal{O}(\log(1/\beta))$ such that the following inequality holds as long as $K \geq z_\beta$:*

$$
\mathbb{E}[NG(\pi_{T,K}^1, \pi_{T,K}^2)] \lesssim \underbrace{\frac{A_{\max}^2 T}{\tau(1-\gamma)^3}\left(\frac{1+\gamma}{2}\right)^{T-1}}_{:=\mathcal{E}_1} + \underbrace{\frac{A_{\max}^2 L_{in}(K-z_\beta)^{1/2}}{\tau(1-\gamma)^4}\left(1-\frac{\beta}{2}\right)^{\frac{K-z_\beta-1}{2}}}_{:=\mathcal{E}_2}
$$

$$
+ \underbrace{\frac{|\mathcal{S}|A_{\max}}{(1-\gamma)^4 c_{\alpha,\beta}}z_\beta^2\alpha^{1/2}}_{:=\mathcal{E}_3} + \underbrace{\frac{\tau\log(A_{\max})}{(1-\gamma)^2}}_{:=\mathcal{E}_4},
$$

*where $L_{in} = \frac{4}{(1-\gamma)} + 2\tau\log(A_{\max}) + \frac{8|\mathcal{S}|A_{\max}}{(1-\gamma)^2}$.*

*(2) When using $\alpha_k = \alpha/(k+h)$ and $\beta_k = \beta/(k+h)$, there exists $k_0 > 0$ such that the following inequality holds as long as $K \geq k_0$:*

$$
\mathbb{E}[NG(\pi_{T,K}^1, \pi_{T,K}^2)] \lesssim \frac{A_{\max}^2 T}{\tau(1-\gamma)^3}\left(\frac{1+\gamma}{2}\right)^{T-1} + \frac{L_{in}|\mathcal{S}|A_{\max}z_K^2\alpha_K^{1/2}}{(1-\gamma)^4\alpha_{k_0}^{1/2}c_{\alpha,\beta}} + \frac{\tau\log(A_{\max})}{(1-\gamma)^2},
$$

*where $z_K = \mathcal{O}(\log(K))$.*

*Remark.* Analogous to [29, 15], our learning dynamics are symmetric between the two players in the sense that there is no time-scale separation between the two players, that is, they both implement the algorithm with the same stepsizes.

A detailed proof sketch of Theorem 3.1 is provided in Appendix B and the complete proof is provided in Appendix D. Next, we discuss the result in Theorem 3.1 (1). The bound in Theorem 3.1 (1) involves a value iteration error term $\mathcal{E}_1$, an optimization error term $\mathcal{E}_2$, a statistical error term $\mathcal{E}_3$, and a smoothing bias term $\mathcal{E}_4$ due to the use of smoothed best-response in the learning dynamics. Note that $\mathcal{E}_1$ would be the only error term if we were able to perform minimax value iteration to solve

---

[2]The proof works as long as $\beta > 2$. We here use $\beta = 4$ to simplify the statement of the results.

the game. Since minimax value iteration converges geometrically, the term $\mathcal{E}_1$ also goes to zero at a geometric rate. Notably, the terms $\mathcal{E}_2$ and $\mathcal{E}_3$ are orderwise larger compared to their matrix-game counterparts, see Corollary 2.1.2. Intuitively, the reason is that the induced auxiliary matrix game (with payoff matrices $\mathcal{T}^1(v_t^1)(s)$ and $\mathcal{T}^2(v_t^2)(s)$) that the inner loop of Algorithm 2 aims at solving does not necessarily have a zero-sum structure (see the discussion in Section 3.1 after Algorithm 2). Consequently, the error due to such a "non-zero-sum" structure propagates through the algorithm and eventually undermines the convergence bound.

Recall that in the matrix game setting, we proved convergence to the Nash distribution (or the Nash equilibrium of the entropy-regularized matrix game). In the stochastic-game setting, we do not have convergence to the Nash equilibrium of the entropy-regularized stochastic game. The main reason is that, in order to have such a convergence, our outer loop should be designed to approximate the *entropy-regularized* minimax value iteration rather than the vanilla minimax value iteration as in Algorithm 2 Line 8. However, in the payoff-based setting, since each player does not even observe the actions of their opponent, it is unclear how to construct an estimator of the entropy function of the opponent's policy, which is an interesting future direction to investigate.

Although the transient terms in Theorem 3.1 enjoy a desirable rate of convergence (e.g., geometric in $T$ and $\tilde{\mathcal{O}}(1/K^{1/2})$ in $K$), the stepsize ratio $c_{\alpha,\beta}$ (which is exponentially small in $\tau$) appears as $c_{\alpha,\beta}^{-1}$ in the bound; see Theorem 3.1. Therefore, due to the presence of the smoothing bias (i.e., the term $\mathcal{E}_4$ on the RHS of the bound in Theorem 3.1 (1)), to achieve $\mathbb{E}[\mathrm{NG}(\pi_{T,K}^1, \pi_{T,K}^2)] \leq \epsilon$, the overall sample complexity can also be exponentially large in $\epsilon^{-1}$. This is analogous to Corollary 2.1.2 for zero-sum matrix games. As illustrated in detail in Section 2, the reason here is due to the exploration limitation of using softmax as a means for smoothed best response, which we kept without further modification to preserve the naturalness of the learning dynamics. Removing such exponential factors by developing improved exploration strategies is an immediate future direction.

Finally, we consider the case where the opponent of player $i$ (where $i \in \{1, 2\}$) plays the game with a stationary policy and provide a finite-sample bound for player $i$ to find the best response.

**Corollary 3.1.1.** *[Rationality[3]] Given $i \in \{1, 2\}$, suppose that player $i$ follows the learning dynamics presented in Algorithm 2, but its opponent player $-i$ follows a stationary policy, denoted by $\pi^{-i}$. Then, we have $\max_{\hat{\pi}^i} U^i(\hat{\pi}^i, \pi^{-i}) - \mathbb{E}[U^i(\pi_{T,K}^i, \pi^{-i})] \leq \tilde{\mathcal{O}}\left(\omega_1 T \left(\frac{\gamma+1}{2}\right)^T + \frac{\omega_2}{K^{1/2}} + \tau\right)$, where $\omega_1$ and $\omega_2$ are constants that are exponential in $\tau^{-1}$, but polynpomial in $|\mathcal{S}|$, $A_{\max}$, and $1/(1-\gamma)$.*

Intuitively, the reason that our algorithm is rational is that it performs an *on-policy* update in RL. In contrast to an off-policy update, where the behavior policy can be arbitrarily different from the policy being generated during learning (such as in $Q$-learning [81]), in the on-policy update for games, each player is actually playing with the policy that is moving towards the best response to its opponent. As a result, when the opponent's policy is stationary, it reduces to a single-agent problem and the player naturally finds the best response (also up to a smoothing bias). This is an advantage of using symmetric and independent learning dynamics. One challenge of analyzing such on-policy learning dynamics is that the behavior policy is time-varying.

## 4 Conclusion and Future Directions

In this work, we consider payoff-based independent learning for zero-sum matrix games and stochastic games. In both settings, we establish the last-iterate finite-sample guarantees. Our approach, i.e., the coupled Lyapunov drift argument, provides a number of tools that are likely to be of interest more broadly for dealing with iterative algorithms with multiple sets of coupled and stochastic iterates.

**Limitations and Future Directions.** As mentioned before Corollary 2.1 and after Theorem 3.1, the convergence bounds involve constants that are exponential in $\tau^{-1}$, which arise due to the use of the smoothed best response to preserve the naturalness of the learning dynamics. An immediate future direction of this work is to remove such exponential factors by designing better exploration strategies. In the long term, we are interested to see if the algorithmic ideas and the analysis techniques developed in this work can be used to study other classes of games beyond zero-sum stochastic games.

---

[3]According to the definition in [27], a dynamics being rational means that the player following this dynamics will converge to the best response to its opponent when the opponent uses an *asymptotically* stationary policy. Since we are performing finite-sample analysis, we assume the opponent's policy *is* stationary, because otherwise, the convergence rate (which may be arbitrary) of the opponent's policy will also impact the bound.

## Acknowledgement

The authors would like to thank the anonymous reviewers for the helpful feedback, especially for pointing out several related references. ZC acknowledges support from the PIMCO Postdoctoral Fellowship. KZ acknowledges support from the Northrop Grumman – Maryland Seed Grant Program. EM acknowledges support from NSF CAREER Award 2240110. AW acknowledges support from NSF Grants CNS-2146814, CPS-2136197, CNS-2106403, and NGSDI-2105648.

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

# A    Extended Related Work

Continued from Section 1.3, we here discuss several other existing works that are relevant.

Recently, there has been an increasing study of MARL with sample efficiency guarantees recently [16, 82, 20, 17, 21, 83, 84, 22, 85]. Most of them focus on the finite-horizon episodic setting with online exploration, and perform regret analysis, which differs from our last-iterate finite-sample analysis under the stochastic approximation paradigm. Additionally, these algorithms are episodic due to the finite-horizon nature of the setting and are not best-response-type independent learning dynamics that are repeatedly run for infinitely long, which can be viewed as a non-equilibrating adaptation process. In fact, the primary focus of this line of work is a *self-play* setting where all the players can be controlled to perform centralized learning [86, 16, 82, 20, 17]. Beyond the online setting, finite-sample efficiency has also been established for MARL using a generative model [87, 88] or offline datasets [89–91, 67]. These algorithms tend to be centralized in nature and focus on *equilibrium computation*, instead of performing independent learning.

Finite-sample complexity has also been established for *policy gradient* methods, a popular RL approach when applied to solving zero-sum stochastic games [14, 36–38]. However, to ensure convergence, these methods are *asymmetric* in that the players update their policies at *different* timescales, e.g., one player updates faster than the other with larger stepsizes, or one player fixes its policy while waiting for the other to update. Such asymmetric policy gradient methods are not independent, as some implicit coordination is required to enable such a timescale separation across agents. This style of implicit coordination is also required for the finite-sample analysis of decentralized learning in certain general-sum stochastic games, e.g., [92], which improves the asymptotic convergence in [7]. In contrast, our learning dynamics only require the update of each player's policy to be slower than the update of their $q$-functions, but crucially we do not assume a time-scale separation between the two players, making our learning dynamics symmetric.

# B    Proof Sketch of Theorem 3.1

In this section, we present the key steps and technical ideas used to prove Theorem 3.1. The core challenge here is that Algorithm 2 maintains 3 sets of iterates ($\{q_{t,k}^i\}$, $\{\pi_{t,k}^i\}$, and $\{v_t^i\}$), which are coupled. The coupling of their update equations means that it is not possible to separately analyze them. Instead, we develop a coupled Lyapunov drift approach to establish the finite-sample bounds of Algorithm 2. Specifically, we first show that the expected Nash gap can be upper bounded by a sum of properly defined Lyapunov functions, one for each set of the iterates (i.e., the $v$-functions, the policies, and the $q$-functions). Then, we establish a set of coupled Lyapunov drift inequalities – one for each Lyapunov function. Finally, we decouple the Lyapunov drift inequalities to establish the overall finite-sample bounds. We outline the key steps in the argument below.

To begin with, we show in Lemma D.1 that the $q$-functions $\{q_{t,k}^i\}$ and the $v$-functions $\{v_t^i\}$ generated by Algorithm 2 are uniformly bounded from above in $\ell_\infty$-norm by $1/(1 - \gamma)$ and the entries of the policies $\{\pi_{t,k}^i\}$ are uniformly bounded below by $\ell_\tau > 0$. This result will be frequently used in our analysis. We next introduce the Lyapunov functions we use to analyze Algorithm 2. Specifically, for any $t, k \geq 0$, let $\bar{q}_{t,k}^i \in \mathbb{R}^{|\mathcal{S}||\mathcal{A}^i|}$ be defined as $\bar{q}_{t,k}^i(s) = \mathcal{T}^i(v_t^i)(s)\pi_{t,k}^{-i}(s)$ for all $s \in \mathcal{S}$. Let

$$\mathcal{L}_{\text{sum}}(t) = \|v_t^1 + v_t^2\|_\infty, \quad \mathcal{L}_v(t) = \sum_{i=1,2} \|v_t^i - v_*^i\|_\infty,$$

$$\mathcal{L}_q(t,k) = \sum_{i=1,2} \sum_{s \in \mathcal{S}} \|q_{t,k}^i(s) - \mathcal{T}^i(v_t^i)(s)\pi_{t,k}^{-i}(s)\|_2^2 = \sum_{i=1,2} \|q_{t,k}^i - \bar{q}_{t,k}^i\|_2^2,$$

$$\mathcal{L}_\pi(t,k) = \max_{s \in \mathcal{S}} \sum_{i=1,2} \max_{\mu^i \in \Delta(\mathcal{A}^i)} \left\{ (\mu^i - \pi_{t,k}^i(s))^\top \mathcal{T}^i(v_t^i)(s)\pi_{t,k}^{-i}(s) + \tau\nu(\mu^i) - \tau\nu(\pi_{t,k}^i(s)) \right\}.$$

Note that $\mathcal{L}_{\text{sum}}(t)$ is introduced to deal with the fact that the induced matrix game the inner loop of Algorithm 2 is designed to solve may not be a zero-sum game due to independent learning. See the discussion in Section 1.2 and the paragraph after Algorithm 2. At the core of our argument is the following inequality (cf. Lemma D.4):

$$\text{NG}(\pi_{T,K}^1, \pi_{T,K}^2) \leq \frac{4}{1 - \gamma} \left( 2\mathcal{L}_{\text{sum}}(T) + \mathcal{L}_v(T) + \mathcal{L}_\pi(T, K) + 2\tau \log(A_{\max}) \right), \tag{6}$$

which motivates us to bound all the Lyapunov functions.

## B.1 Analysis of the Outer Loop: $v$-Function Update

Motivated by Eq. (6), we need to bound $\mathcal{L}_{\text{sum}}(T)$ and $\mathcal{L}_v(T)$. To achieve that, we establish Lyapunov drift inequalities for them. Specifically, we show in Lemmas D.5 and D.6 that

$$\mathcal{L}_v(t+1) \leq \underbrace{\gamma \mathcal{L}_v(t)}_{\text{Drift}} + \underbrace{4\mathcal{L}_{\text{sum}}(t) + 2\mathcal{L}_q^{1/2}(t,K) + 4\mathcal{L}_\pi(t,K) + 6\tau \log(A_{\max})}_{\text{Additive Errors}}, \tag{7}$$

$$\mathcal{L}_{\text{sum}}(t+1) \leq \underbrace{\gamma \mathcal{L}_{\text{sum}}(t)}_{\text{Drift}} + \underbrace{2\mathcal{L}_q(t,K)^{1/2}}_{\text{Additive Errors}}, \quad \forall\, t \geq 0. \tag{8}$$

Suppose that the *Additive Errors* in the previous two inequalities were only functions of $v_t^1$ and $v_t^2$, then these two Lyapunov drift inequalities can be repeatedly used to obtain a convergence bound for $\mathcal{L}_{\text{sum}}(T)$ and $\mathcal{L}_v(T)$. However, the coupled nature of Eqs. (7) and (8) requires us to analyze the policies and the $q$-functions in the inner loop, and establish their Lyapunov drift inequalities.

## B.2 Analysis of the Inner Loop: Policy Update

As illustrated in Section 2.1 and Section 3.1, for each state $s$, the update equation of the policies can be viewed as a discrete and stochastic variant of the smoothed best-response dynamics for solving matrix games [29]. Typically, the following Lyapunov function is used to study such dynamics [53]:

$$V_X(\mu^1, \mu^2) = \sum_{i=1,2} \max_{\hat{\mu}^i \in \Delta(\mathcal{A}^i)} \{(\hat{\mu}^i - \mu^i)^\top X_i \mu^{-i} + \tau\nu(\hat{\mu}^i) - \tau\nu(\mu^i)\}, \tag{9}$$

where $X_1$ and $X_2$ are the payoff matrices for player 1 and player 2, respectively, and $\nu(\cdot)$ is the entropy function. Specialized to our case, given a joint $v$-function $v = (v^1, v^2)$ from the outer loop[4] and a state $s \in \mathcal{S}$, we would like to use

$$V_{v,s}(\pi^1(s), \pi^2(s)) = \sum_{i=1,2} \max_{\hat{\mu}^i \in \Delta(\mathcal{A}^i)} \{(\hat{\mu}^i - \pi^i(s))^\top \mathcal{T}^i(v^i)(s)\pi^{-i}(s) + \tau\nu(\hat{\mu}^i) - \tau\nu(\pi^i(s))\}$$

as our Lyapunov function. Note that $\max_{s \in \mathcal{S}} V_{v_t,s}(\pi_{t,k}^1(s), \pi_{t,k}^2(s)) = \mathcal{L}_\pi(t,k)$. A sequence of properties (e.g., strong convexity, smoothness, etc.) regarding the Lyapunov function $V_X(\cdot, \cdot)$ is established in Lemma D.7. In the end, we show in Lemma D.8 that

$$\mathbb{E}_t[\mathcal{L}_\pi(t,k+1)] \leq \underbrace{\left(1 - \frac{3\beta_k}{4}\right)\mathbb{E}_t[\mathcal{L}_\pi(t,k)]}_{\text{Drift}}$$

$$+ \underbrace{2L_\tau\beta_k^2 + \frac{32A_{\max}^2\beta_k}{\tau^3\ell_\tau^2(1-\gamma)^2}\mathbb{E}_t[\mathcal{L}_q(t,k)] + \frac{16A_{\max}^2\beta_k}{\tau}\mathcal{L}_{\text{sum}}(t)^2}_{\text{Additive Errors}}, \tag{10}$$

where $\mathbb{E}_t[\cdot]$ stands for conditional expectation conditioned on the history up to the beginning of the $t$-th outer loop. To interpret the above, suppose that we were considering the continuous-time smoothed best-response dynamics. Then, the additive error term would disappear in the sense that the time-derivative of the Lyapunov function along the trajectory of the ODE is strictly negative. Thus, the three terms in the *Additive Errors* can be interpreted as (1) the discretization error in the update equation, (2) the stochastic error in the $q$-function estimate, and (3) the error due to the non-zero-sum structure of the inner-loop auxiliary matrix game.

## B.3 Analysis of the Inner Loop: $q$-Function Update

Our next focus is the $q$-function update. The $q$-function update equation is in the same spirit as TD-learning in RL, and a necessary condition for the convergence of TD-learning is that the behavior

---

[4]Due to the nested-loop structure of Algorithm 2, conditioned on the history up to the beginning of the $t$-th outer loop, the $v$-functions $v_t^1$ and $v_t^2$ are constants.

policy (i.e., the policy used to collect samples) should enable the agent to sufficiently explore the environment. To achieve this goal, since we show in Lemma D.1 that all joint policies from the algorithm trajectory have uniformly lower-bounded entries (with lower bound $\ell_\tau > 0$), it is enough to restrict our attention to a "soft" policy class $\Pi_\tau := \{\pi = (\pi^1, \pi^2) \mid \min_{s,a^1} \pi^1(a^1|s) \geq \ell_\tau, \min_{s,a^2} \pi^2(a^2|s) \geq \ell_\tau\}$. The following lemma, which is an extension of [10, Lemma 4], establishes a uniform exploration property under Assumption 3.1.

To present the result, we need the following notation. Under Assumption 3.1, the Markov chain induced by the joint policy $\pi_b$ has a unique stationary distribution $\mu_b \in \Delta(\mathcal{S})$ [93], the minimum component of which is denoted by $\mu_{b,\min}$. In addition, there exists $\rho_b \in (0,1)$ such that $\max_{s\in\mathcal{S}} \left\| P_{\pi_b}^k(s,\cdot) - \mu_b(\cdot) \right\|_{\mathrm{TV}} \leq 2\rho_b^k$ for all $k \geq 0$ [93], where $P_{\pi_b}$ is the transition probability matrix of the Markov chain $\{S_k\}$ under $\pi_b$. We also define the mixing time in the following. Given a joint policy $\pi = (\pi^1, \pi^2)$ and an accuracy level $\eta > 0$, the $\eta$ – mixing time of the Markov chain $\{S_k\}$ induced by $\pi$ is defined as

$$t_{\pi,\eta} = \min\left\{ k \geq 0 \; : \; \max_{s\in\mathcal{S}} \| P_\pi^k(s,\cdot) - \mu_\pi(\cdot) \|_{\mathrm{TV}} \leq \eta \right\}, \tag{11}$$

where $P_\pi$ is the $\pi$-induced transition probability matrix and $\mu_\pi$ is the stationary distribution of $\{S_k\}$ under $\pi$, provided that it exists and is unique. When the induced Markov chain mixes at a geometric rate, it is easy to see that $t_{\pi,\eta} = \mathcal{O}(\log(1/\eta))$.

**Lemma B.1** (An Extension of Lemma 4 in [10]). *Suppose that Assumption 3.1 is satisfied. Then we have the following results.*

(1) *For any $\pi = (\pi^1, \pi^2) \in \Pi_\tau$, the Markov chain $\{S_k\}$ induced by the joint policy $\pi$ is irreducible and aperiodic, hence admits a unique stationary distribution $\mu_\pi \in \Delta(\mathcal{S})$.*

(2) *It holds that $\sup_{\pi\in\Pi_\tau} \max_{s\in\mathcal{S}} \| P_\pi^k(s,\cdot) - \mu_\pi(\cdot) \|_{\mathrm{TV}} \leq 2\rho_\tau^k$ for any $k \geq 0$, where $\rho_\tau = \rho_b^{\ell_\tau^{2r_b}\mu_{b,\min}}$ and $r_b := \min\{k \geq 0 \; : \; P_{\pi_b}^k(s,s') > 0, \; \forall\, (s,s')\}$. As a result, we have*

$$t(\ell_\tau, \eta) := \sup_{\pi\in\Pi_\tau} t_{\pi,\eta} \leq \frac{t_{\pi_b,\eta}}{\ell_\tau^{2r_b}\mu_{b,\min}}, \tag{12}$$

*where we recall that $t_{\pi_b,\eta}$ is the $\eta$ – mixing time of the Markov chain $\{S_k\}$ induced by $\pi_b$.*

(3) *There exists $L_p \geq 1$ (which was used in the statement of Theorem 3.1) such that*

$$\|\mu_\pi - \mu_{\bar{\pi}}\|_1 \leq L_p\big( \max_{s\in\mathcal{S}} \|\pi^1(s) - \pi^2(s)\|_1 + \max_{s\in\mathcal{S}} \|\bar{\pi}^1(s) - \bar{\pi}^2(s)\|_1 \big)$$

*for all $\pi = (\pi^1, \pi^2), \bar{\pi} = (\bar{\pi}^1, \bar{\pi}^2) \in \Pi_\tau$.*

(4) *$\mu_{\min} := \inf_{\pi\in\Pi_\tau} \min_{s\in\mathcal{S}} \mu_\pi(s) > 0$.*

*Remark.* Lemma B.1 (1), (3), and (4) were previous established in [10, Lemma 4]. Lemma B.1 (2) enables us to see the explicit dependence of the "uniform mixing time" on the margin $\ell_\tau$ and the mixing time of the benchmark exploration policy $\pi_b$.

In view of Lemma B.1 (2), we have fast mixing for all policies in $\Pi_\tau$ if *(i)* the margin $\ell_\tau$ is large, and *(ii)* the Markov chain $\{S_k\}$ induced by the benchmark exploration policy $\pi_b$ is well-behaved. By "well-behaved" we mean the mixing time is small (i.e., small $t_{\pi_b,\eta}$) and the stationary distribution is relatively well-balanced (i.e., large $\mu_{b,\min}$). Point *(i)* agrees with our intuition as a large margin encourages more exploration. To make sense of point *(ii)*, since $\pi(a|s) \geq \ell_\tau^2 \pi_b(a|s)$ for all $s$ and $a = (a^1, a^2)$, we can write $\pi$ as a convex combination between $\pi_b$ and some residual policy $\tilde{\pi}$: $\pi(\cdot|s) = \ell_\tau^2 \pi_b(\cdot|s) + (1 - \ell_\tau^2)\tilde{\pi}(\cdot|s)$ for all $s \in \mathcal{S}$. Therefore, since any $\pi \in \Pi_\tau$ has a portion of the benchmark exploration policy $\pi_b$ in it, it makes intuitive sense that fast mixing of $\{S_k\}$ under $\pi_b$ implies, to some extent, fast mixing of $\{S_k\}$ under $\pi \in \Pi_\tau$. Note that, as the margin $\ell_\tau$ approaches zero, the uniform mixing time in Lemma B.1 (2) goes to infinity. This is not avoidable in general, as demonstrated by a simple MDP example constructed in Appendix E.

We define $c_\tau = \mu_{\min}\ell_\tau$, which was used in the statement of Theorem 3.1. With Lemma B.1 in hand, we are now able to analyze the behavior of the $q$-functions. We model the $q$-function update as a

stochastic approximation algorithm driven by time-inhomogeneous Markovian noise, and use the norm-square function

$$\sum_{i=1,2} \sum_s \|q^i(s) - \mathcal{T}^i(v^i)(s)\pi^{-i}(s)\|_2^2$$

as the Lyapunov function to study its behavior. Note that $\mathcal{L}_q(t,k) = \sum_{i=1,2} \sum_s \|q_{t,k}^i(s) - \mathcal{T}^i(v_t^i)(s)\pi_{t,k}^{-i}(s)\|_2^2$. The key challenge to establishing a Lyapunov drift inequality is to control a difference of the form

$$\mathbb{E}[F^i(q^i, S_k, A_k^i, A_k^{-i}, S_{k+1})] - \mathbb{E}[F^i(q^i, \hat{S}, \hat{A}^i, \hat{A}^{-i}, \hat{S}')] \tag{13}$$

for any $q^i \in \mathbb{R}^{|\mathcal{S}||\mathcal{A}^i|}$ and $i \in \{1,2\}$, where $F^i(\cdot)$ is some appropriately defined operator that captures the dynamics of the update equation; see Appendix D.5.2 for its definition. In the term (13), the random tuple $(S_k, A_k^i, A_k^{-i}, S_{k+1})$ is the $k$-th sample from the time-inhomogeneous Markov chain $\{(S_k, A_k^i, A_k^{-i}, S_{k+1})\}_{k \geq 0}$ generated by the time-varying joint policies $\{\pi_k\}_{k \geq 0}$, and $(\hat{S}, \hat{A}^i, \hat{A}^{-i}, \hat{S}')$ is a random tuple such that $S \sim \mu_k(\cdot)$, $A^i \sim \pi_k^i(\cdot|S)$, $A^{-i} \sim \pi_k^{-i}(\cdot|S)$, and $S' \sim p(\cdot|S, A^i, A^{-i})$, where $\mu_k(\cdot)$ denotes the unique stationary distribution of the Markov chain $\{S_n\}_{n \geq 0}$ induced by the joint policy $\pi_k$. Lemma B.1 implies that $\mu_k$ exists and is unique.

In the existing literature, when $\{(S_k, A_k^i, A_k^{-i}, S_{k+1})\}$ is sampled either in an i.i.d. manner or forms an ergodic time-homogeneous Markov chain, there are techniques that successfully bound the term (13) [94, 24, 23]. To deal with time-inhomogeneous Markovian noise, building upon existing conditioning results [23, 24, 41, 76] and also Lemma B.1, we develop a refined conditioning argument to show that

$$(13) = \mathcal{O}\left(z_k \sum_{n=k-z_k}^{k-1} \alpha_n\right), \tag{See Lemma D.11}$$

where $z_k = t(\ell_\tau, \beta_k)$ is a uniform upper bound on the $\beta_k$ – the mixing time (i.e., the uniform mixing time with accuracy $\beta_k$, see Eq. (12)) of the Markov chain $\{S_n\}_{n \geq 0}$ induced by an arbitrary joint policy from the algorithm trajectory. Suppose that we are using diminishing stepsizes of $\mathcal{O}(1/k)$ decay rate (similar results hold for using constant stepsizes). Then, the uniform mixing property from Lemma B.1 (2) implies that $z_k = \mathcal{O}(\log(1/k))$. As a result, we have $\lim_{k \to \infty}(13) \leq \lim_{k \to \infty} z_k \sum_{n=k-z_k}^{k-1} \alpha_n = 0$, which provides us a way to control the term in (13). After successfully handling (13), we are able to establish a Lyapunov drift inequality of $\mathcal{L}_q(t,k)$:

$$\mathbb{E}_t[\mathcal{L}_q(t, k+1)] \leq \underbrace{(1 - \alpha_k c_\tau)\mathbb{E}_t[\mathcal{L}_q(t,k)]}_{\text{Drift}} + \underbrace{C_0 z_k \alpha_k \alpha_{k-z_k, k-1} + \frac{\beta_k}{4}\mathbb{E}[\mathcal{L}_\pi(t,k)]}_{\text{Additive Errors}}, \tag{14}$$

where $C_0$ is a (problem-dependent) constant, and we use the notation $\alpha_{k_1, k_2} := \sum_{k=k_1}^{k_2} \alpha_k$ to simplify the notation. See Lemma D.12 for more details. When $k$ is large, it can be shown that $\alpha_{k-z_k, k-1} \leq 2\alpha_k z_k$ [65, Appendix 1.8].

## B.4 Solving Coupled Lyapunov Drift Inequalities

Until this point, we have established the Lyapunov drift inequalities for the individual $v$-functions, the sum of the $v$-functions, the policies, and the $q$-functions in Eqs. (7), (8), (10), and (14), respectively. The last challenge is to find a way of using these coupled inequalities to derive the finite-sample bound. To elaborate, we first restate all the Lyapunov drift inequalities in the following:

$$\mathcal{L}_v(t+1) \leq \gamma \mathcal{L}_v(t) + 4\mathcal{L}_{\text{sum}}(t) + 4\mathcal{L}_\pi(t, K) + 2\mathcal{L}_q^{1/2}(t, K) + 6\tau \log(A_{\max}), \tag{15}$$

$$\mathcal{L}_{\text{sum}}(t+1) \leq \gamma \mathcal{L}_{\text{sum}}(t) + 2\mathcal{L}_q^{1/2}(t, K), \tag{16}$$

$$\mathbb{E}_t[\mathcal{L}_\pi(t, k+1)] \leq (1 - 3\beta_k/4)\mathbb{E}_t[\mathcal{L}_\pi(t, k)] + C_1(\beta_k^2 + \beta_k \mathbb{E}_t[\mathcal{L}_q(t,k)] + \beta_k \mathcal{L}_{\text{sum}}^2(t)), \tag{17}$$

$$\mathbb{E}_t[\mathcal{L}_q(t, k+1)] \leq (1 - c_\tau \alpha_k)\mathbb{E}_t[\mathcal{L}_q(t,k)] + \beta_k \mathbb{E}_t[\mathcal{L}_\pi(t,k)]/4 + C_3 z_k^2 \alpha_k^2. \tag{18}$$

To decouple the Lyapunov inequalities stated above, our high-level ideas are (1) using the Lyapunov drift inequalities in a combined way instead of in a separate manner, and (2) a *bootstrapping* procedure

where we first derive a crude bound and then substitute the crude bound back into the Lyapunov drift inequalities to derive a tighter bound. We next present our approach.

For ease of presentation, for a scalar-valued quantity $W$ that is a function of $k$ and/or $t$, we say $W = o_k(1)$ if $\lim_{k\to\infty} W = 0$ and $W = o_t(1)$ if $\lim_{t\to\infty} W = 0$. The explicit convergence rates of the $o_k(1)$ term and the $o_t(1)$ term will be revealed in the complete proof in Appendix D.6, but is not important for the illustration here.

**Step 1.** Adding up Eq. (17) and (18), using Condition 3.1, and then repeatedly using the resulting inequality, we obtain:

$$\mathbb{E}_t[\mathcal{L}_\pi(t, k)] \leq \mathbb{E}_t[\mathcal{L}_\pi(t, k) + \mathcal{L}_q(t, k)] = o_k(1) + \mathcal{O}(1)\mathcal{L}_{\text{sum}}^2(t), \ \forall \, t, k. \tag{19}$$

**Step 2.** Substituting the bound for $\mathbb{E}_t[\mathcal{L}_\pi(t, k)]$ in Eq. (19) into Eq. (18) and repeatedly using the resulting inequality, and we obtain:

$$\mathbb{E}_t[\mathcal{L}_q(t, K)] = o_K(1) + \mathcal{O}(c_{\alpha,\beta})\mathcal{L}_{\text{sum}}^2(t), \ \forall \, t,$$

which in turn implies (by first using Jensen's inequality and then taking total expectation) that:

$$\mathbb{E}[\mathcal{L}_q^{1/2}(t, K)] = o_K(1) + \mathcal{O}(c_{\alpha,\beta}^{1/2})\mathbb{E}[\mathcal{L}_{\text{sum}}(t)], \ \forall \, t, \tag{20}$$

where we recall that $c_{\alpha,\beta} = \beta_k/\alpha_k$ is the stepsize ratio. The fact that we are able to get a factor of $\mathcal{O}(c_{\alpha,\beta}^{1/2})$ in front of $\mathbb{E}[\mathcal{L}_{\text{sum}}(t)]$ is crucial for the decoupling procedure.

**Step 3.** Taking total expectation on both sides of Eq. (16) and then using the upper bound of $\mathbb{E}[\mathcal{L}_q^{1/2}(t, K)]$ we obtained in Eq. (20), we obtain

$$\mathbb{E}[\mathcal{L}_{\text{sum}}(t + 1)] \leq (\gamma + \mathcal{O}(c_{\alpha,\beta}^{1/2}))\mathbb{E}[\mathcal{L}_{\text{sum}}(t)] + o_K(1), \ \forall \, t.$$

By choosing $c_{\alpha,\beta}$ so that $\mathcal{O}(c_{\alpha,\beta}^{1/2}) \leq (1 - \gamma)/2$, the previous inequality implies

$$\mathbb{E}[\mathcal{L}_{\text{sum}}(t + 1)] \leq \left(1 - \frac{1 - \gamma}{2}\right)\mathbb{E}[\mathcal{L}_{\text{sum}}(t)] + o_K(1), \ \forall \, t, \tag{21}$$

which can be repeatedly used to obtain

$$\mathbb{E}[\mathcal{L}_{\text{sum}}(t)] = o_t(1) + o_K(1). \tag{22}$$

Substituting the previous bound on $\mathbb{E}[\mathcal{L}_{\text{sum}}(t)]$ into Eq. (19), we have

$$\max(\mathbb{E}[\mathcal{L}_\pi(t, K)], \mathbb{E}[\mathcal{L}_q(t, K)]) = o_t(1) + o_K(1). \tag{23}$$

**Step 4.** Substituting the bounds we obtained for $\mathbb{E}[\mathcal{L}_\pi(t, K)]$, $\mathbb{E}[\mathcal{L}_q(t, K)]$, and $\mathbb{E}[\mathcal{L}_{\text{sum}}(t)]$ in Eqs. (22) and (23) into Eq. (15), and then repeatedly using the resulting inequality from $t = 0$ to $t = T$, we have

$$\mathbb{E}[\mathcal{L}_v(T)] = o_T(1) + o_K(1) + \mathcal{O}(\tau).$$

Now that we have obtained finite-sample bounds for $\mathbb{E}[\mathcal{L}_v(T)]$, $\mathbb{E}[\mathcal{L}_{\text{sum}}(T)]$, $\mathbb{E}[\mathcal{L}_\pi(T, K)]$, and $\mathbb{E}[\mathcal{L}_q(T, K)]$, using them in Eq. (6), we finally obtain the desired finite-sample bound for the expected Nash gap.

Looking back at the decoupling procedure, Steps 2 and 3 are crucial. In fact, in Step 1 we already obtain a bound on $\mathbb{E}_t[\mathcal{L}_q(t, k)]$, where the additive error is $\mathcal{O}(1)\mathbb{E}[\mathcal{L}_{\text{sum}}(t)]$. However, directly using this bound on $\mathbb{E}_t[\mathcal{L}_q(t, k)]$ in Eq. (16) would result in an expansive inequality for $\mathbb{E}[\mathcal{L}_{\text{sum}}(t)]$. By performing Step 2, we are able to obtain a tighter bound for $\mathbb{E}_t[\mathcal{L}_q(t, k)]$, with the additive error being $\mathcal{O}(c_{\alpha,\beta}^{1/2})\mathbb{E}[\mathcal{L}_{\text{sum}}(t)]$. Furthermore, we can choose $c_{\alpha,\beta}$ to be small enough so that after using the bound from Eq. (20) in Eq. (16), the additive error $\mathcal{O}(c_{\alpha,\beta}^{1/2})\mathbb{E}[\mathcal{L}_{\text{sum}}(t)]$ is dominated by the negative drift in Eq. (21).

## C  Proof of Theorem 2.1

The proof is divided into 4 steps. In Appendix C.1, we prove an important boundedness property regarding the iterates generated by Algorithm 1. In Appendices C.2 and C.3, we analyze the evolution of the policies and the $q$-functions by establishing the negative drift inequalities with respect to their associated Lyapunov functions. In Appendix C.4, we solve the coupled Lyapunov drift inequalities to prove Theorem 2.1. Moreover, we prove Corollary 2.1.1 in Appendix C.5. The statement and proof of all supporting lemmas used in this section are presented in Appendix C.6.

## C.1 Boundedness of the Iterates

In this subsection, we show that the $q$-functions generated by Algorithm 1 are uniformly bounded from above, and the entries of the policies are uniformly bounded from below. The following lemma is needed to establish the result.

**Lemma C.1.** *For any $i \in \{1, 2\}$ and $q^i \in \mathbb{R}^{|\mathcal{A}|^i}$, we have*

$$\min_{a^i \in \mathcal{A}^i} [\sigma_\tau(q^i)](a^i) \geq \frac{1}{(A_{\max} - 1) \exp(2\|q^i\|_\infty/\tau) + 1}.$$

*Proof of Lemma C.1.* Given $i \in \{1, 2\}$, for any $q^i \in \mathbb{R}^{|\mathcal{A}|^i}$ and $a^i \in \mathcal{A}^i$, we have

$$\begin{aligned}
[\sigma_\tau(q^i)](a^i) &= \frac{\exp(q^i(a^i)/\tau)}{\sum_{\bar{a}^i \in \mathcal{A}^i} \exp(q^i(\bar{a}^i)/\tau)} \\
&= \frac{1}{\sum_{\bar{a}^i \neq a^i} \exp((q^i(\bar{a}^i) - q^i(a^i))/\tau) + 1} \\
&\geq \frac{1}{(|\mathcal{A}^i| - 1) \exp(2\|q^i\|_\infty/\tau) + 1} \\
&\geq \frac{1}{(A_{\max} - 1) \exp(2\|q^i\|_\infty/\tau) + 1}.
\end{aligned}$$

Since the RHS of the previous inequality does not depend on $a^i$, we have the desired inequality. $\square$

We next derive the boundedness property in the following lemma.

**Lemma C.2.** *It holds for all $k \geq 0$ and $i \in \{1, 2\}$ that $\|q_k^i\|_\infty \leq 1$ and $\min_{a^i \in \mathcal{A}^i} \pi_k^i(a^i) \geq \ell_\tau$, where $\ell_\tau = [(A_{\max} - 1) \exp(2/\tau) + 1]^{-1}$.*

*Proof of Lemma C.2.* We prove the results by induction. Since $q_0^i = 0$ and $\pi_0^i$ is initialized as a uniform distribution on $\mathcal{A}^i$, we have the base case. Now suppose that the results hold for some $k \geq 0$. Using the update equation for $q_k^i$ in Algorithm 1 Line 5, we have

$$\begin{aligned}
|q_{k+1}^i(a^i)| &= |(1 - \alpha_k \mathbb{1}_{\{a^i = A_k^i\}}) q_k^i(a^i) + \alpha_k \mathbb{1}_{\{a^i = A_k^i\}} R_i(A_k^i, A_k^{-i})| \\
&\leq \max(|q_k^i(a^i)|, (1 - \alpha_k)|q_k^i(a^i)| + \alpha_k |R_i(A_k^i, A_k^{-i})|) \\
&\leq 1
\end{aligned}$$

for any $a^i \in \mathcal{A}^i$, where the last line follows from the induction hypothesis $\|q_k^i\|_\infty \leq 1$ and $|R_i(a^i, a^{-i})| \leq 1$ for all $(a^i, a^{-i})$. As for $\pi_{k+1}^i$, using the update equation for $\pi_k^i$ in Algorithm 1 Line 3, we have

$$\begin{aligned}
\pi_{k+1}^i(a^i) &= (1 - \beta_k) \pi_k^i(a^i) + \beta_k [\sigma_\tau(q_k^i)](a^i) \\
&\geq (1 - \beta_k) \ell_\tau + \frac{\beta_k}{(A_{\max} - 1) \exp(2\|q_k^i\|_\infty/\tau) + 1} && \text{(Lemma C.1)} \\
&\geq (1 - \beta_k) \ell_\tau + \beta_k \ell_\tau && (\|q_k^i\|_\infty \leq 1 \text{ by induction hypothesis}) \\
&= \ell_\tau.
\end{aligned}$$

The induction is complete. $\square$

## C.2 Analysis of the Policies

Let $V_R : \Delta(\mathcal{A}^1) \times \Delta(\mathcal{A}^2) \mapsto \mathbb{R}$ be the regularized Nash gap defined as

$$V_R(\mu^1, \mu^2) = \sum_{i=1,2} \max_{\hat{\mu}^i \in \Delta(\mathcal{A}^i)} \left\{ (\hat{\mu}^i - \mu^i)^\top R_i \mu^{-i} + \tau \nu(\hat{\mu}^i) - \tau \nu(\mu^i) \right\},$$

where $\nu(\cdot)$ is the entropy function. A sequence of properties regarding $V_R(\cdot, \cdot)$ are provided in Lemma C.7. For simplicity of notation, we use $\nabla_1 V_R(\cdot, \cdot)$ (respectively, $\nabla_2 V_R(\cdot, \cdot)$) to represent the gradient with respect to the first argument (respectively, the second argument).

**Lemma C.3.** *It holds for all $k \geq 0$ that*

$$\mathbb{E}[V_R(\pi_{k+1}^1, \pi_{k+1}^2)] \leq \left(1 - \frac{\beta_k}{2}\right)\mathbb{E}[V_R(\pi_k^1, \pi_k^2)] + \frac{\ell_\tau \alpha_k}{4}\sum_{i=1,2}\mathbb{E}[\|q_k^i - R_i\pi_k^{-i}\|_2^2] + 2L_\tau\beta_k^2,$$

*where we recall that $L_\tau = \tau/\ell_\tau + A_{\max}^2/\tau$.*

*Proof of Lemma C.3.* Using the smoothness property of $V_R(\cdot, \cdot)$ (cf. Lemma C.7 (1)) and the update equation in Algorithm 1 Line 3, we have for any $k \geq 0$ that

$$V_R(\pi_{k+1}^1, \pi_{k+1}^2) \leq V_R(\pi_k^1, \pi_k^2) + \beta_k\langle \nabla_2 V_R(\pi_k^1, \pi_k^2), \sigma_\tau(q_k^2) - \pi_k^2\rangle$$
$$+ \beta_k\langle \nabla_1 V_R(\pi_k^1, \pi_k^2), \sigma_\tau(q_k^1) - \pi_k^1\rangle + \frac{L_\tau\beta_k^2}{2}\sum_{i=1,2}\|\sigma_\tau(q_k^i) - \pi_k^i\|_2^2$$

$$\leq V_R(\pi_k^1, \pi_k^2) + \beta_k\langle \nabla_2 V_R(\pi_k^1, \pi_k^2), \sigma_\tau(R_2\pi_k^1) - \pi_k^2\rangle$$
$$+ \beta_k\langle \nabla_1 V_R(\pi_k^1, \pi_{k+1}^2), \sigma_\tau(R_1\pi_k^2) - \pi_k^1\rangle$$
$$+ \beta_k\langle \nabla_2 V_R(\pi_k^1, \pi_k^2), \sigma_\tau(q_k^2) - \sigma_\tau(R_2\pi_k^1)\rangle$$
$$+ \beta_k\langle \nabla_1 V_R(\pi_k^1, \pi_{k+1}^2), \sigma_\tau(q_k^1) - \sigma_\tau(R_1\pi_k^2)\rangle + 2L_\tau\beta_k^2$$

$$\leq \left(1 - \frac{\beta_k}{2}\right)V_R(\pi_k^1, \pi_k^2) + 4\beta_k\left(\frac{1}{\tau\ell_\tau^2} + \frac{A_{\max}^2}{\tau^3}\right)\sum_{i=1,2}\|q_k^i - R_i\pi_k^{-i}\|_2^2 + 2L_\tau\beta_k^2,$$

where the last line follows from Lemma C.7 (2) and (3). Taking expectations on both sides of the previous inequality and using the condition that $c_{\alpha,\beta} = \frac{\beta_k}{\alpha_k} \leq \min(\frac{\tau\ell_\tau^3}{32}, \frac{\ell_\tau\tau^3}{32A_{\max}^2})$ (cf. Condition 2.1), we have

$$\mathbb{E}[V_R(\pi_{k+1}^1, \pi_{k+1}^2)] \leq \left(1 - \frac{\beta_k}{2}\right)\mathbb{E}[V_R(\pi_k^1, \pi_k^2)] + \frac{\ell_\tau\alpha_k}{4}\sum_{i=1,2}\mathbb{E}[\|q_k^i - R_i\pi_k^{-i}\|_2^2] + 2L_\tau\beta_k^2.$$

The proof is complete. $\qquad\square$

## C.3 Analysis of the $q$-Functions

We study the $q$-functions generated by Algorithm 1 through a stochastic approximation framework. For $i \in \{1, 2\}$, let $F^i : \mathbb{R}^{|\mathcal{A}^i|} \times \mathcal{A}^i \times \mathcal{A}^{-i} \mapsto \mathbb{R}^{|\mathcal{A}^i|}$ be an operator defined as

$$[F^i(q^i, a_0^i, a_0^{-i})](a^i) = \mathbb{1}_{\{a_0^i = a^i\}}\left(R_i(a_0^i, a_0^{-i}) - q^i(a_0^i)\right), \quad \forall \, (q^i, a_0^i, a_0^{-i}) \text{ and } a^i.$$

Then, Algorithm 1 Line 5 can be compactly written as

$$q_{k+1}^i = q_k^i + \alpha_k F^i(q_k^i, A_k^i, A_k^{-i}). \tag{24}$$

Given a joint policy $(\pi^1, \pi^2)$, let $\bar{F}_\pi^i : \mathbb{R}^{|\mathcal{A}^i|} \mapsto \mathbb{R}^{|\mathcal{A}^i|}$ be defined as

$$\bar{F}_\pi^i(q^i) := \mathbb{E}_{A^i \sim \pi^i(\cdot), A^{-i} \sim \pi^{-i}(\cdot)}[F^i(q^i, A^i, A^{-i})] = \text{diag}(\pi^i)(R_i\pi^{-i} - q^i).$$

Then, Eq. (24) can be viewed as a stochastic approximation algorithm for solving the slowly time-varying equation $\bar{F}_{\pi_k}^i(q^i) = 0$.

**Lemma C.4.** *The following inequality holds for all $k \geq 0$:*

$$\sum_{i=1,2}\mathbb{E}[\|q_{k+1}^i - R_i\pi_{k+1}^{-i}\|_2^2] \leq \left(1 - \frac{\ell_\tau\alpha_k}{2}\right)\sum_{i=1,2}\mathbb{E}[\|q_k^i - R_i\pi_k^{-i}\|_2^2] + \frac{\beta_k}{4}\mathbb{E}[V_R(\pi_k^1, \pi_k^2)] + 16\alpha_k^2.$$

*Proof of Lemma C.4.* For any $k \geq 0$ and $i \in \{1, 2\}$, we have

$$\|q_{k+1}^i - R_i\pi_{k+1}^{-i}\|_2^2$$
$$= \|q_{k+1}^i - q_k^i + q_k^i - R_i\pi_k^{-i} + R_i\pi_k^{-i} - R_i\pi_{k+1}^{-i}\|_2^2$$
$$= \|q_{k+1}^i - q_k^i\|_2^2 + \|q_k^i - R_i\pi_k^{-i}\|_2^2 + \|R_i\pi_k^{-i} - R_i\pi_{k+1}^{-i}\|_2^2 + 2\langle q_{k+1}^i - q_k^i, q_k^i - R_i\pi_k^{-i}\rangle$$

$$
+ 2\langle q_{k+1}^i - q_k^i, R_i\pi_k^{-i} - R_i\pi_{k+1}^{-i}\rangle + 2\langle R_i\pi_k^{-i} - R_i\pi_{k+1}^{-i}, q_k^i - R_i\pi_k^{-i}\rangle
$$

$$
= \alpha_k^2\|F^i(q_k^i, A_k^i, A_k^{-i})\|_2^2 + \|q_k^i - R_i\pi_k^{-i}\|_2^2 + \beta_k^2\|R_i(\sigma_\tau(q_k^{-i}) - \pi_k^{-i})\|_2^2
$$
$$
+ 2\alpha_k\langle \bar{F}_{\pi_k}^i(q_k^i), q_k^i - R_i\pi_k^{-i}\rangle + 2\alpha_k\langle F^i(q_k^i, A_k^i, A_k^{-i}) - \bar{F}_{\pi_k}^i(q_k^i), q_k^i - R_i\pi_k^{-i}\rangle
$$
$$
- 2\alpha_k\beta_k\langle F^i(q_k^i, A_k^i, A_k^{-i}), R_i(\sigma_\tau(q_k^{-i}) - \pi_k^{-i})\rangle - 2\beta_k\langle R_i(\sigma_\tau(q_k^{-i}) - \pi_k^{-i}), q_k^i - R_i\pi_k^{-i}\rangle
$$
$$
\text{(Algorithm 1 Lines 3 and 5)}
$$

$$
\leq \alpha_k^2\|F^i(q_k^i, A_k^i, A_k^{-i})\|_2^2 + \|q_k^i - R_i\pi_k^{-i}\|_2^2 + \beta_k^2\|R_i(\sigma_\tau(q_k^{-i}) - \pi_k^{-i})\|_2^2
$$
$$
+ 2\alpha_k\langle \bar{F}_{\pi_k}^i(q_k^i), q_k^i - R_i\pi_k^{-i}\rangle + 2\alpha_k\langle F^i(q_k^i, A_k^i, A_k^{-i}) - \bar{F}_{\pi_k}^i(q_k^i), q_k^i - R_i\pi_k^{-i}\rangle
$$
$$
+ 2\alpha_k\beta_k\|F^i(q_k^i, A_k^i, A_k^{-i})\|_2\|R_i(\sigma_\tau(q_k^{-i}) - \pi_k^{-i})\|_2
$$
$$
+ 2\beta_k\|R_i(\sigma_\tau(q_k^{-i}) - \pi_k^{-i})\|_2\|q_k^i - R_i\pi_k^{-i}\|_2 \qquad \text{(Cauchy–Schwarz inequality)}
$$

$$
\leq \alpha_k^2\|F^i(q_k^i, A_k^i, A_k^{-i})\|_2^2 + \|q_k^i - R_i\pi_k^{-i}\|_2^2 + \beta_k^2\|R_i(\sigma_\tau(q_k^{-i}) - \pi_k^{-i})\|_2^2
$$
$$
+ 2\alpha_k\langle \bar{F}_{\pi_k}^i(q_k^i), q_k^i - R_i\pi_k^{-i}\rangle + 2\alpha_k\langle F^i(q_k^i, A_k^i, A_k^{-i}) - \bar{F}_{\pi_k}^i(q_k^i), q_k^i - R_i\pi_k^{-i}\rangle
$$
$$
+ \frac{\alpha_k\beta_k}{c_1}\|F^i(q_k^i, A_k^i, A_k^{-i})\|_2^2 + \alpha_k\beta_k c_1\|R_i(\sigma_\tau(q_k^{-i}) - \pi_k^{-i})\|_2^2
$$
$$
+ \frac{\beta_k}{c_2}\|R_i(\sigma_\tau(q_k^{-i}) - \pi_k^{-i})\|_2^2 + c_2\beta_k\|q_k^i - R_i\pi_k^{-i}\|_2^2
$$
$$
\text{(This follows from the AM-GM inequality, where } c_1, c_2 > 0 \text{ can be arbitrary.)}
$$

$$
= \left(\alpha_k^2 + \frac{\alpha_k\beta_k}{c_1}\right)\|F^i(q_k^i, A_k^i, A_k^{-i})\|_2^2 + (1 + c_2\beta_k)\|q_k^i - R_i\pi_k^{-i}\|_2^2
$$
$$
+ 2\alpha_k\langle \bar{F}_{\pi_k}^i(q_k^i), q_k^i - R_i\pi_k^{-i}\rangle + 2\alpha_k\langle F^i(q_k^i, A_k^i, A_k^{-i}) - \bar{F}_{\pi_k}^i(q_k^i), q_k^i - R_i\pi_k^{-i}\rangle
$$
$$
+ \left(\beta_k^2 + \frac{\beta_k}{c_2} + \alpha_k\beta_k c_1\right)\|R_i\|_2^2\|\sigma_\tau(q_k^{-i}) - \pi_k^{-i}\|_2^2.
$$

Taking expectations on both sides of the previous inequality, we have

$$
\mathbb{E}[\|q_{k+1}^i - R_i\pi_{k+1}^{-i}\|_2^2]
$$
$$
\leq \left(\alpha_k^2 + \frac{\alpha_k\beta_k}{c_1}\right)\mathbb{E}[\|F^i(q_k^i, A_k^i, A_k^{-i})\|_2^2] + (1 + c_2\beta_k)\mathbb{E}[\|q_k^i - R_i\pi_k^{-i}\|_2^2]
$$
$$
+ 2\alpha_k\mathbb{E}[\langle \bar{F}_{\pi_k}^i(q_k^i), q_k^i - R_i\pi_k^{-i}\rangle] + \left(\beta_k^2 + \frac{\beta_k}{c_2} + \alpha_k\beta_k c_1\right)\|R_i\|_2^2\mathbb{E}[\|\sigma_\tau(q_k^{-i}) - \pi_k^{-i}\|_2^2],
$$

where the term $\mathbb{E}[\langle F^i(q_k^i, A_k^i, A_k^{-i}) - \bar{F}_{\pi_k}^i(q_k^i), q_k^i - R_i\pi_k^{-i}\rangle]$ vanishes due to the tower property of conditional expectations. To proceed, observe

$$
\mathbb{E}[\|F^i(q_k^i, A_k^i, A_k^{-i})\|_2^2] = \mathbb{E}\left[\sum_{a^i \in \mathcal{A}^i} \mathbb{1}_{\{A_k^i = a^i\}}\left(R_i(A_k^i, A_k^{-i}) - q_k^i(A_k^i)\right)^2\right]
$$
$$
= \mathbb{E}[(R_i(a^i, A_k^{-i}) - q_k^i(a^i))^2]
$$
$$
\leq \mathbb{E}[(|R_i(a^i, A_k^{-i})| + |q_k^i(a^i)|)^2]
$$
$$
\leq 4 \qquad \text{(Lemma C.2)}
$$

and

$$
\mathbb{E}[\langle \bar{F}_{\pi_k}^i(q_k^i), q_k^i - R_i\pi_k^{-i}\rangle] = \mathbb{E}[\langle \mathrm{diag}(\pi_k^i)(R_i\pi_k^{-i} - q_k^i), q_k^i - R_i\pi_k^{-i}\rangle]
$$
$$
\leq -\ell_\tau\mathbb{E}[\|q_k^i - R_i\pi_k^{-i}\|_2^2]. \qquad \text{(Lemma C.2)}
$$

In addition, we have

$$
\mathbb{E}[\|\sigma_\tau(q_k^{-i}) - \pi_k^{-i}\|_2^2] \leq 2\mathbb{E}[\|\sigma_\tau(q_k^{-i}) - \sigma_\tau(R_{-i}\pi_k^i)\|_2^2] + 2\mathbb{E}[\|\sigma_\tau(R_{-i}\pi_k^i) - \pi_k^{-i}\|_2^2]
$$
$$
\qquad (a^2 + b^2 \geq 2ab)
$$
$$
\leq \frac{2}{\tau^2}\mathbb{E}[\|q_k^{-i} - R_{-i}\pi_k^i\|_2^2] + \frac{4}{\tau}\mathbb{E}[V_R(\pi_k^1, \pi_k^2)],
$$

where the last line follows from Lemma C.6. Using the previous 4 inequalities all together, we obtain

$$\mathbb{E}[\|q_{k+1}^i - R_i \pi_{k+1}^{-i}\|_2^2] \leq 4\left(\alpha_k^2 + \frac{\alpha_k \beta_k}{c_1}\right) + (1 - 2\ell_\tau \alpha_k + c_2 \beta_k) \mathbb{E}[\|q_k^i - R_i \pi_k^{-i}\|_2^2]$$
$$+ \frac{4A_{\max}^2}{\tau}\left(\beta_k^2 + \frac{\beta_k}{c_2} + \alpha_k \beta_k c_1\right) \mathbb{E}[V_R(\pi_k^1, \pi_k^2)]$$
$$+ \frac{2A_{\max}^2}{\tau^2}\left(\beta_k^2 + \frac{\beta_k}{c_2} + \alpha_k \beta_k c_1\right) \mathbb{E}[\|q_k^{-i} - R_i \pi_k^{-i}\|_2^2].$$

Summing up the previous inequality for $i \in \{1, 2\}$, we have

$$\sum_{i=1,2} \mathbb{E}[\|q_{k+1}^i - R_i \pi_{k+1}^{-i}\|_2^2]$$
$$\leq \left(1 - 2\ell_\tau \alpha_k + c_2 \beta_k + \frac{2A_{\max}^2}{\tau^2}\left(\beta_k^2 + \frac{\beta_k}{c_2} + \alpha_k \beta_k c_1\right)\right) \sum_{i=1,2} \mathbb{E}[\|q_k^i - R_i \pi_k^{-i}\|_2^2]$$
$$+ \frac{8A_{\max}^2}{\tau}\left(\beta_k^2 + \frac{\beta_k}{c_2} + \alpha_k \beta_k c_1\right) \mathbb{E}[V_R(\pi_k^1, \pi_k^2)] + 8\left(\alpha_k^2 + \frac{\alpha_k \beta_k}{c_1}\right)$$
$$= \left(1 - \frac{3\ell_\tau \alpha_k}{2} + \frac{2A_{\max}^2}{\tau^2}\left(2\beta_k^2 + \frac{2\beta_k^2}{\ell_\tau \alpha_k}\right)\right) \sum_{i=1,2} \mathbb{E}[\|q_k^i - R_i \pi_k^{-i}\|_2^2]$$
$$+ \frac{8A_{\max}^2}{\tau}\left(2\beta_k^2 + \frac{2\beta_k^2}{\ell_\tau \alpha_k}\right) \mathbb{E}[V_R(\pi_k^1, \pi_k^2)] + 16\alpha_k^2 \qquad \text{(Choosing } c_1 = \frac{\beta_k}{\alpha_k} \text{ and } c_2 = \frac{\ell_\tau \alpha_k}{2\beta_k})$$
$$\leq \left(1 - \frac{\ell_\tau \alpha_k}{2}\right) \sum_{i=1,2} \mathbb{E}[\|q_k^i - R_i \pi_k^{-i}\|_2^2] + \frac{\beta_k}{4} \mathbb{E}[V_R(\pi_k^1, \pi_k^2)] + 16\alpha_k^2,$$

where the last line follows from $c_{\alpha,\beta} \leq \min(\frac{\tau^2 \ell_\tau}{8A_{\max}^2}, \frac{\tau \ell_\tau}{128 A_{\max}^2})$ and $\beta_0 \leq \frac{\tau}{128 A_{\max}^2}$ (cf. Condition 2.1). The proof is complete. □

## C.4 Solving the Coupled Lyapunov Inequalities

For simplicity of notation, denote $\mathcal{L}_q(k) = \sum_{i=1,2} \mathbb{E}[\|q_k^i - R_i \pi_k^{-i}\|_2^2]$ and $\mathcal{L}_\pi(k) = \mathbb{E}[V_R(\pi_k^1, \pi_k^2)]$. Then, Lemmas C.3 and C.4 state that

$$\mathcal{L}_\pi(k+1) \leq \left(1 - \frac{\beta_k}{2}\right) \mathcal{L}_\pi(k) + \frac{\ell_\tau \alpha_k}{4} \mathcal{L}_q(k) + 2L_\tau \beta_k^2,$$

and

$$\mathcal{L}_q(k+1) \leq \left(1 - \frac{\ell_\tau \alpha_k}{2}\right) \mathcal{L}_q(k) + \frac{\beta_k}{4} \mathcal{L}_\pi(k) + 16\alpha_k^2.$$

Adding up the previous two inequalities, we obtain

$$\mathcal{L}_q(k+1) + \mathcal{L}_\pi(k+1) \leq \left(1 - \frac{\beta_k}{4}\right) \mathcal{L}_\pi(k) + 2L_\tau \beta_k^2 + \left(1 - \frac{\ell_\tau \alpha_k}{4}\right) \mathcal{L}_q(k) + 16\alpha_k^2$$
$$\leq \left(1 - \frac{\beta_k}{4}\right) (\mathcal{L}_\pi(k) + \mathcal{L}_q(k)) + 2L_\tau \beta_k^2 + 16\alpha_k^2, \qquad (25)$$

where the second inequality follows from $c_{\alpha,\beta} \leq \ell_\tau$ (cf. Condition 2.1).

**Constant Stepsizes.** When using constant stepsizes, i.e., $\alpha_k \equiv \alpha$ and $\beta_k \equiv \beta$, repeatedly using Eq. (25), we have for all $k \geq 0$ that

$$\mathcal{L}_q(k) + \mathcal{L}_\pi(k) \leq \left(1 - \frac{\beta}{4}\right)^k (\mathcal{L}_\pi(0) + \mathcal{L}_q(0)) + 8L_\tau \beta + 64\alpha^2/\beta$$
$$\leq \left(1 - \frac{\beta}{4}\right)^k (4 + 2\tau \log(A_{\max}) + 2A_{\max}) + 8L_\tau \beta + 64\alpha^2/\beta$$

$$= B_{\text{in}} \left( 1 - \frac{\beta}{4} \right)^k + 8L_\tau \beta + \frac{64\alpha}{c_{\alpha,\beta}}$$

where the second inequality follows from

$$\mathcal{L}_\pi(0) \leq 4 + 2\tau \log(A_{\max}), \quad \text{and} \quad \mathcal{L}_q(0) \leq 2A_{\max}.$$

Theorem 2.1 (1) follows by observing that $\mathcal{L}_q(k) + \mathcal{L}_\pi(k) \geq \mathcal{L}_\pi(k) = \mathbb{E}[\text{NG}_\tau(\pi_k^1, \pi_k^2)]$.

**Diminishing Stepsizes.** Consider using $\alpha_k = \frac{\alpha}{k+h}$ and $\beta_k = \frac{\beta}{k+h}$. Recursions of the form presented in Eq. (25) have been well studied in the existing literature for the convergence rates of iterative algorithms [44, 24, 65]. Since $\beta > 4$, using the same line of analysis as in [65, Appendix A.2], we have

$$\mathcal{L}_q(k) + \mathcal{L}_\pi(k) \leq B_{\text{in}} \left( \frac{h}{k+h} \right)^{\beta/4} + (64eL_\tau\beta + 512e\alpha/c_{\alpha,\beta}) \frac{1}{k+h}.$$

Theorem 2.1 (2) follows by observing that $\mathcal{L}_q(k) + \mathcal{L}_\pi(k) \geq \mathcal{L}_\pi(k) = \mathbb{E}[\text{NG}_\tau(\pi_k^1, \pi_k^2)]$.

### C.5 Proof of Corollary 2.1.1

We use Theorem 2.1 (1) to derive the sample complexity, and choose $\beta = c_{\alpha,\beta}\alpha$ with $c_{\alpha,\beta}$ satisfying Condition 2.1. To achieve $\mathbb{E}[\text{NG}_\tau(\pi_K^1, \pi_K^2)] \leq \epsilon$, in view of Theorem 2.1 (1), it is sufficient that

$$B_{\text{in}}e^{-\beta K/4} \leq \frac{\epsilon}{3}, \quad 8L_\tau\beta \leq \frac{\epsilon}{3}, \quad 64\alpha/c_{\alpha,\beta} \leq \frac{\epsilon}{3},$$

which implies $\beta = \mathcal{O}(\epsilon)$. It follows that $K = \mathcal{O}\left( \epsilon^{-1} \right)$.

### C.6 Supporting Lemmas

**Lemma C.5.** *For any $i \in \{1, 2\}$ and $\mu_1^i, \mu_2^i \in \{\mu^i \in \Delta(\mathcal{A}^i) \mid \min_{a^i \in \mathcal{A}^i} \mu^i(a^i) \geq \ell_\tau\}$, we have*

$$\|\nabla\nu(\mu_1^i) - \nabla\nu(\mu_2^i)\|_2 \leq \frac{1}{\ell_\tau} \|\mu_1^i - \mu_2^i\|_2.$$

*Proof of Lemma C.5.* For any $i \in \{1, 2\}$ and $\mu^i \in \Delta(\mathcal{A}^i)$ such that $\min_{a^i \in \mathcal{A}^i} \mu^i(a^i) \geq \ell_\tau$, the Hessian of $\nu(\cdot)$ satisfies

$$\text{Hessian}_\nu(\mu^i) = \text{diag}\left(\mu^i\right)^{-1} \leq \frac{I_{|\mathcal{A}^i|}}{\min_{a^i \in \mathcal{A}^i} \mu^i(a^i)} \leq \frac{I_{|\mathcal{A}^i|}}{\ell_\tau}.$$

Therefore, the gradient of the negative entropy function $\nabla\nu(\cdot)$ is $\frac{1}{\ell_\tau}$ – Lipschitz continuous with respect to $\|\cdot\|_2$ on the set $\{\mu^i \in \Delta(\mathcal{A}^i) \mid \min_{a^i \in \mathcal{A}^i} \mu^i(a^i) \geq \ell_\tau\}$, which implies the $\frac{1}{\ell_\tau}$ – smoothness of $\nu(\cdot)$. $\qquad\square$

**Lemma C.6.** *For $i \in \{1, 2\}$, we have for all $\mu^i \in \Delta(\mathcal{A}^i)$ and $\mu^{-i} \in \Delta(\mathcal{A}^{-i})$ that*

$$\|\sigma_\tau(R_i\mu^{-i}) - \mu^i\|_2^2 \leq \frac{2}{\tau} V_R(\mu^1, \mu^2).$$

*Proof of Lemma C.6.* Recall that the negative entropy $\nu(\cdot)$ is 1-strongly concave with respect to $\|\cdot\|_2$. Therefore, given $i \in \{1, 2\}$, fix $\mu^{-i}$, the function

$$\max_{\hat{\mu}^i \in \Delta(\mathcal{A}^i)} \left\{ (\hat{\mu}^i - \mu^i)^\top R_i\mu^{-i} + \tau\nu(\hat{\mu}^i) - \tau\nu(\mu^i) \right\}$$

is $\tau$-strongly convex with respect to $\mu^i$. As a result, by the quadratic growth property of strongly convex functions, we have

$$\|\sigma_\tau(R_i\mu^{-i}) - \mu^i\|_2^2 \leq \frac{2}{\tau} \max_{\hat{\mu}^i \in \Delta(\mathcal{A}^i)} \left\{ (\hat{\mu}^i - \mu^i)^\top R_i\mu^{-i} + \tau\nu(\hat{\mu}^i) - \tau\nu(\mu^i) \right\} \leq \frac{2}{\tau} V_R(\mu^1, \mu^2).$$

$\qquad\square$

Denote $\Pi_\tau = \{(\pi^1, \pi^2) \in \Delta(\mathcal{A}^1) \times \Delta(\mathcal{A}^2) \mid \min_{a^1 \in \mathcal{A}^1} \pi^1(a^1) \geq \ell_\tau, \min_{a^2 \in \mathcal{A}^2} \pi^2(a^2) \geq \ell_\tau\}$.

**Lemma C.7.** *The function $V_R(\cdot, \cdot)$ has the following properties.*

*(1) The function $V_R(\mu^1, \mu^2)$ is $L_\tau$ – smooth on $\Pi_\tau$, where $L_\tau = \frac{\tau}{\ell_\tau} + \frac{A_{\max}^2}{\tau}$.*

*(2) It holds for any $(\mu^1, \mu^2) \in \Pi_\tau$ that*

$$\langle \nabla_1 V_R(\mu^1, \mu^2), \sigma_\tau(R_1 \mu^2) - \mu^1 \rangle + \langle \nabla_2 V_R(\mu^1, \mu^2), \sigma_\tau(R_2 \mu^1) - \mu^2 \rangle \leq -V_R(\mu^1, \mu^2).$$

*(3) For any $q^1 \in \mathbb{R}^{|\mathcal{A}^1|}$ and $q^2 \in \mathbb{R}^{|\mathcal{A}^2|}$, we have for all $(\mu^1, \mu^2) \in \Pi_\tau$ that*

$$\langle \nabla_1 V_R(\mu^1, \mu^2), \sigma_\tau(q^1) - \sigma_\tau(R_1 \mu^2) \rangle + \langle \nabla_2 V_R(\mu^1, \mu^2), \sigma_\tau(q^2) - \sigma_\tau(R_2 \mu^1) \rangle$$

$$\leq \frac{1}{2} V_R(\mu^i, \mu^{-i}) + 4 \left( \frac{1}{\tau \ell_\tau^2} + \frac{A_{\max}^2}{\tau^3} \right) \sum_{i=1,2} \|q^i - R_i \mu^{-i}\|_2^2.$$

*Proof of Lemma C.7.* Recall the definition of $V_R(\cdot, \cdot)$ in the following:

$$V_R(\mu^1, \mu^2) = \sum_{i=1,2} \max_{\hat{\mu}^i \in \Delta(\mathcal{A}^i)} \left\{ (\hat{\mu}^i - \mu^i)^\top R_i \mu^{-i} + \tau \nu(\hat{\mu}^i) - \tau \nu(\mu^i) \right\}.$$

By Danskin's theorem [95], we have

$$\nabla_1 V_R(\mu^1, \mu^2) = -\tau \nabla \nu(\mu^1) + (R_2)^\top \sigma_\tau(R_2 \mu^1),$$
$$\nabla_2 V_R(\mu^1, \mu^2) = -\tau \nabla \nu(\mu^2) + (R_1)^\top \sigma_\tau(R_1 \mu^2),$$

both of which will be frequently used in our analysis.

(1) For any $(\mu^1, \mu^2), (\bar{\mu}^1, \bar{\mu}^2) \in \Pi_\tau$, we have

$$\|\nabla_1 V_R(\mu^1, \mu^2) - \nabla_1 V_R(\bar{\mu}^1, \bar{\mu}^2)\|_2$$
$$= \|\tau \nabla \nu(\bar{\mu}^1) - \tau \nabla \nu(\mu^1) + (R_2)^\top \sigma_\tau(R_2 \mu^1) - (R_2)^\top \sigma_\tau(R_2 \bar{\mu}^1)\|_2$$
$$\leq \tau \|\nabla \nu(\bar{\mu}^1) - \nabla \nu(\mu^1)\|_2 + \|R_2\|_2 \|\sigma_\tau(R_2 \mu^1) - \sigma_\tau(R_2 \bar{\mu}^1)\|_2$$
$$\leq \frac{\tau}{\ell_\tau} \|\mu^1 - \bar{\mu}^1\|_2 + \frac{\|R_2\|_2^2}{\tau} \|\mu^1 - \bar{\mu}^1\|_2$$
$$\leq \left( \frac{\tau}{\ell_\tau} + \frac{A_{\max}^2}{\tau} \right) \|\mu^1 - \bar{\mu}^1\|_2,$$

where the second last inequality follows from Lemma C.5 and $\sigma_\tau(\cdot)$ being $\frac{1}{\tau}$-Lipschitz continuous with respect to $\| \cdot \|_2$ [96], and the last inequality follows from $\|R_i\|_2 \leq \sqrt{|\mathcal{A}^1||\mathcal{A}^2|} \leq A_{\max}$ for $i \in \{1, 2\}$. Similarly, we also have

$$\|\nabla_2 V_R(\mu^1, \mu^2) - \nabla_2 V_R(\bar{\mu}^1, \bar{\mu}^2)\|_2^2 \leq \left( \frac{\tau}{\ell_\tau} + \frac{A_{\max}^2}{\tau} \right) \|\mu^2 - \bar{\mu}^2\|_2.$$

It follows from the previous two inequalities that

$$\|\nabla V_R(\mu^1, \mu^2) - \nabla V_R(\bar{\mu}^1, \bar{\mu}^2)\|_2^2$$
$$= \|\nabla_1 V_R(\mu^1, \mu^2) - \nabla_1 V_R(\bar{\mu}^1, \bar{\mu}^2)\|_2^2 + \|\nabla_2 V_R(\mu^1, \mu^2) - \nabla_2 V_R(\bar{\mu}^1, \bar{\mu}^2)\|_2^2$$
$$\leq \left( \frac{\tau}{\ell_\tau} + \frac{A_{\max}^2}{\tau} \right)^2 \sum_{i=1,2} \|\mu^i - \bar{\mu}^i\|_2^2,$$

which implies that $V_R(\cdot, \cdot)$ is an $L_\tau$ – smooth function on $\Pi_\tau$ [97], where $L_\tau = \frac{\tau}{\ell_\tau} + \frac{A_{\max}^2}{\tau}$.

(2) The result follows from Lemma D.7 by setting $X_i = R_i$, $i \in \{1, 2\}$, and by observing that $R_1 + (R_2)^\top = 0$.

(3) Using the formula of the gradient of $V_R(\cdot, \cdot)$ in the begining of the proof, we have

$$\langle \nabla_1 V_R(\mu^1, \mu^2), \sigma_\tau(q^1) - \sigma_\tau(R_1\mu^2) \rangle$$

$$= \langle -\tau \nabla \nu(\mu^1) + (R_2)^\top \sigma_\tau(R_2\mu^1), \sigma_\tau(q^1) - \sigma_\tau(R_1\mu^2) \rangle$$

$$= \tau \langle \nabla \nu(\sigma_\tau(R_1\mu^2)) - \nabla \nu(\mu^1), \sigma_\tau(q^1) - \sigma_\tau(R_1\mu^2) \rangle$$

$$+ (\sigma_\tau(R_2\mu^1) - \mu^2)^\top R_2(\sigma_\tau(q^1) - \sigma_\tau(R_1\mu^2))$$

(This follows from the order optimality condition: $R_1\mu^2 + \tau \nabla \nu(\sigma_\tau(R_1\mu^2)) = 0$)

$$\leq \frac{\tau}{2c_1} \|\nabla \nu(\sigma_\tau(R_1\mu^2)) - \nabla \nu(\mu^1)\|_2^2 + \frac{\tau c_1}{2} \|\sigma_\tau(q^1) - \sigma_\tau(R_1\mu^2)\|_2^2$$

$$+ \frac{1}{2c_2} \|\sigma_\tau(R_2\mu^1) - \mu^2)\|_2^2 + \frac{c_2}{2} \|R_2(\sigma_\tau(q^1) - \sigma_\tau(R_1\mu^2))\|_2^2$$

(This follows from AM-GM inequality, where $c_1, c_2 > 0$ can be arbitrary)

$$\leq \frac{\tau}{2c_1\ell_\tau^2} \|\sigma_\tau(R_1\mu^2) - \mu^1\|_2^2 + \frac{c_1}{2\tau} \|q^1 - R_1\mu^2\|_2^2$$

$$+ \frac{1}{2c_2} \|\sigma_\tau(R_2\mu^1) - \mu^2)\|_2^2 + \frac{c_2\|R_2\|_2^2}{2\tau^2} \|q^1 - R_1\mu^2\|_2^2 \qquad \text{(Lemma C.5)}$$

$$\leq \left( \frac{1}{c_1\ell_\tau^2} + \frac{1}{\tau c_2} \right) V_R(\mu^1, \mu^2) + \frac{c_1}{2\tau} \|q^1 - R_1\mu^2\|_2^2 + \frac{c_2\|R_2\|_2^2}{2\tau^2} \|q^1 - R_1\mu^2\|_2^2 \quad \text{(Lemma C.6)}$$

$$\leq \frac{1}{4} V_R(\mu^1, \mu^2) + \frac{4}{\tau\ell_\tau^2} \|q^1 - R_1\mu^2\|_2^2 + \frac{4\|R_2\|_2^2}{\tau^3} \|q^1 - R_1\mu^2\|_2^2,$$

where the last line follows by choosing $c_1 = \frac{8}{\ell_\tau^2}$ and $c_2 = \frac{8}{\tau}$. Similarly, we also have

$$\langle \nabla_2 V_R(\mu^1, \mu^2), \sigma_\tau(q^2) - \sigma_\tau(R_2\mu^1) \rangle \leq \frac{1}{4} V_R(\mu^1, \mu^2) + \frac{4}{\tau\ell_\tau^2} \|q^2 - R_2\mu^1\|_2^2$$

$$+ \frac{4\|R_1\|_2^2}{\tau^3} \|q^2 - R_2\mu^1\|_2^2.$$

Summing up the previous two inequalities, we obtain

$$\langle \nabla_1 V_R(\mu^1, \mu^2), \sigma_\tau(q^1) - \sigma_\tau(R_1\mu^2) \rangle + \langle \nabla_2 V_R(\mu^1, \mu^2), \sigma_\tau(q^2) - \sigma_\tau(R_2\mu^1) \rangle$$

$$\leq \frac{1}{2} V_R(\mu^i, \mu^{-i}) + \left( \frac{4}{\tau\ell_\tau^2} + \frac{4A_{\max}^2}{\tau^3} \right) \sum_{i=1,2} \|q^i - R_i\mu^{-i}\|_2^2,$$

where we used $\|R_i\|_2 \leq \sqrt{A_{\max}}$ for $i \in \{1, 2\}$.

$\square$

# D    Proof of Theorem 3.1

We begin by introducing a summary of notation in Appendix D.1. In Appendix D.2, we establish an important boundedness property regarding the $q$-functions, value functions, and the policies generated by Algorithm 2. In Appendix D.3, we bound the Nash gap in terms of the Lyapunov functions. In Appendices D.4 and D.5, we analyze the outer loop and the inner loop of Algorithm 2 and establish the Lyapunov drift inequalities. Finally, in Appendix D.6, we solve the coupled Lyapunov inequalities to obtain the finite-sample bound. The proof of all supporting lemmas are provided in Appendix D.7, and the proof of Corollary 3.1.1 is provided in Appendix D.8.

## D.1    Notation

We begin with a summary of the notation that will be used in the proof.

(1) Given a pair of matrices $\{X_i \in \mathbb{R}^{|\mathcal{A}^i| \times |\mathcal{A}^{-i}|}\}_{i \in \{1,2\}}$ and a pair of distributions $\{\mu^i \in \Delta(\mathcal{A}^i)\}_{i \in \{1,2\}}$, we define

$$V_X(\mu^1, \mu^2) = \sum_{i=1,2} \max_{\hat{\mu}^i \in \Delta(\mathcal{A}^i)} \left\{ (\hat{\mu}^i - \mu^i)^\top X_i \mu^{-i} + \tau \nu(\hat{\mu}^i) - \tau \nu(\mu^i) \right\}, \qquad (26)$$

where $\nu(\cdot)$ is the entropy function. Note that $V_X(\cdot, \cdot)$ is similar to $V_R(\cdot, \cdot)$ defined in Appendix C.2 in the setting of matrix games. However, we do not assume that $X_1 + X_2 = 0$.

(2) Given a pair of value functions $(v^1, v^2)$ and a state $s \in \mathcal{S}$, when $X_i = \mathcal{T}^i(v^i)(s)$, $i \in \{1, 2\}$, we write $V_{v,s}(\cdot, \cdot)$ for $V_X(\cdot, \cdot)$.

(3) For any joint policy $(\pi^1, \pi^2)$ and state $s$, given $i \in \{1, 2\}$, we define $v^i_{*, \pi^{-i}}(s) = \max_{\hat{\pi}^i} v^i_{\hat{\pi}^i, \pi^{-i}}(s)$, $v^i_{\pi^i, *} = \min_{\hat{\pi}^{-i}} v^i_{\pi^i, \hat{\pi}^{-i}}(s)$, $v^{-i}_{\pi^{-i}, *}(s) = \min_{\hat{\pi}^i} v^{-i}_{\pi^{-i}, \hat{\pi}^i}(s)$, and $v^{-i}_{*, \pi^i}(s) = \max_{\hat{\pi}^{-i}} v^{-i}_{\hat{\pi}^{-i}, \hat{\pi}^i}(s)$. Note that we have $v^1_{*, \pi^2} + v^2_{\pi^2, *} = 0$ and $v^1_{\pi^1, *} + v^2_{*, \pi^1} = 0$ because of the zero-sum structure.

(4) For $i \in \{1, 2\}$, denote $v^i_*$ as the unique fixed point of the equation $\mathcal{B}^i(v^i) = v^i$, where $\mathcal{B}^i(\cdot)$ is the minimax Bellman operator defined in Section 3. Note that we have $v^1_* + v^2_* = 0$.

(5) For any $t, k \geq 0$ and $i \in \{1, 2\}$, let $\bar{q}^i_{t,k} \in \mathbb{R}^{|\mathcal{S}||\mathcal{A}^i|}$ be defined as $\bar{q}^i_{t,k}(s) = \mathcal{T}^i(v^i_t)(s)\pi^{-i}_{t,k}(s)$ for all $s \in \mathcal{S}$. In addition, let

$$\mathcal{L}_{\text{sum}}(t) = \|v^1_t + v^2_t\|_\infty, \quad \mathcal{L}_v(t) = \sum_{i=1,2} \|v^i_t - v^i_*\|_\infty,$$

$$\mathcal{L}_q(t, k) = \sum_{i=1,2} \sum_{s \in \mathcal{S}} \|q^i_{t,k}(s) - \mathcal{T}^i(v^i_t)(s)\pi^{-i}_{t,k}(s)\|^2_2 = \sum_{i=1,2} \|q^i_{t,k} - \bar{q}^i_{t,k}\|^2_2,$$

$$\mathcal{L}_\pi(t, k) = \max_{s \in \mathcal{S}} V_{v_t, s}(\pi^1_{t,k}(s), \pi^2_{t,k}(s)),$$

which will be the Lyapunov functions we use in the analysis.

(6) Given $k_1 \leq k_2$, we denote $\beta_{k_1, k_2} = \sum_{k=k_1}^{k_2} \beta_k$ and $\alpha_{k_1, k_2} = \sum_{k=k_1}^{k_2} \alpha_k$.

(7) Recall that $z_k = t(\ell_\tau, \beta_k)$ is the uniform mixing time defined in Lemma B.1 (2), where $\ell_\tau$ is the uniform lower bound of the policies. When using constant stepsizes, $z_k$ is not a function of $k$, and is simply denoted by $z_\beta$. Observe that, due to the uniform geometric mixing property established in Lemma B.1 (2), we have $z_k = \mathcal{O}(\log(k))$ when using $\mathcal{O}(1/k)$ stepsizes and $z_\beta = \mathcal{O}(\log(1/\beta))$ when using constant stepsizes. Let $k_0 = \min k : k \geq z_k$, which is well defined because $z_k$ grows logarithmically with $k$.

### D.2 Boundedness of the Iterates

We first show in the following lemma that the $q$-functions and the $v$-functions generated by Algorithm 2 are uniformly bounded from above, and the policies are uniformly bounded from below. In the context of stochastic games, we redefine $\ell_\tau = [1 + (A_{\max} - 1)\exp(2/[(1 - \gamma)\tau])]^{-1}$.

**Lemma D.1.** *For all $t, k$ and $i \in \{1, 2\}$, we have (1) $\|v^i_t\|_\infty \leq 1/(1 - \gamma)$ and $\|q^i_{t,k}\|_\infty \leq 1/(1 - \gamma)$, and (2) $\min_{s \in \mathcal{S}, a^i \in \mathcal{A}^i} \pi^i_{t,k}(a^i \mid s) \geq \ell_\tau$.*

*Proof of Lemma D.1.* The proof uses induction arguments. Let $i \in \{1, 2\}$.

(1) Given $t \geq 0$, we first show by induction that, if $\|v^i_t\|_\infty \leq \frac{1}{1-\gamma}$ and $\|q^i_{t,0}\|_\infty \leq \frac{1}{1-\gamma}$, we have $\|q^i_{t,k}\|_\infty \leq \frac{1}{1-\gamma}$ for all $k \geq 0$. The base case $\|q^i_{t,0}\|_\infty \leq \frac{1}{1-\gamma}$ holds by our assumption. Suppose that $\|q^i_{t,k}\|_\infty \leq \frac{1}{1-\gamma}$ for some $k \geq 0$. Then, by Algorithm 2 Line 6, we have for all $(s, a^i)$ that

$$|q^i_{t,k+1}(s, a^i)|$$
$$= |q^i_{t,k}(s, a^i) + \alpha_k \mathbb{1}_{\{(s,a^i)=(S_k, A^i_k)\}}(R_i(S_k, A^i_k, A^{-i}_k) + \gamma v^i_t(S_{k+1}) - q^i_{t,k}(S_k, A^i_k))|$$
$$\leq (1 - \alpha_k \mathbb{1}_{\{(s,a^i)=(S_k, A^i_k)\}})|q^i_{t,k}(s, a^i)|$$
$$\quad + \alpha_k \mathbb{1}_{\{(s,a^i)=(S_k, A^i_k)\}}|R_i(S_k, A^i_k, A^{-i}_k) + \gamma v^i_t(S_{k+1})|$$
$$\leq (1 - \alpha_k \mathbb{1}_{\{(s,a^i)=(S_k, A^i_k)\}})\frac{1}{1 - \gamma} + \alpha_k \mathbb{1}_{\{(s,a^i)=(S_k, A^i_k)\}}\left(1 + \frac{\gamma}{1 - \gamma}\right) \tag{27}$$
$$= \frac{1}{1 - \gamma},$$

where Eq. (27) follows from the induction hypothesis $\|q^i_{t,k}\|_\infty \le \frac{1}{1-\gamma}$, our assumption that $\|v^i_t\|_\infty \le \frac{1}{1-\gamma}$, and $\max_{s,a^i,a^{-i}} |R_i(s, a^i, a^{-i})| \le 1$. The induction is now complete and we have $\|q^i_{t,k}\|_\infty \le \frac{1}{1-\gamma}$ for all $k \ge 0$ whenever $\|v^i_t\|_\infty \le \frac{1}{1-\gamma}$ and $\|q^i_{t,0}\|_\infty \le \frac{1}{1-\gamma}$.

We next again use induction to show that $\|v^i_t\|_\infty \le \frac{1}{1-\gamma}$ and $\|q^i_{t,0}\|_\infty \le \frac{1}{1-\gamma}$ for all $t \ge 0$. Our initialization ensures that $\|v^i_0\|_\infty \le \frac{1}{1-\gamma}$ and $\|q^i_{0,0}\|_\infty \le \frac{1}{1-\gamma}$. Suppose that $\|v^i_t\|_\infty \le \frac{1}{1-\gamma}$ and $\|q^i_{t,0}\|_\infty \le \frac{1}{1-\gamma}$ for some $t \ge 0$. Using the update equation for $v^i_{t+1}$ (cf. Algorithm 2 Line 8) and the fact that $\|q^i_{t,k}\|_\infty \le \frac{1}{1-\gamma}$ for all $k \ge 0$ (established in the previous paragraph), we have for all $s \in \mathcal{S}$ that

$$|v^i_{t+1}(s)| = \left| \sum_{a^i \in \mathcal{A}^i} \pi^i_{t,K}(a^i|s) q^i_{t,K}(s, a^i) \right| \le \sum_{a^i \in \mathcal{A}^i} \pi^i_{t,K}(a^i|s) \|q^i_{t,K}\|_\infty \le \frac{1}{1-\gamma},$$

which implies $\|v^i_{t+1}\|_\infty \le \frac{1}{1-\gamma}$. Moreover, we have by Algorithm 2 Line 9 that $\|q^i_{t+1,0}\|_\infty = \|q^i_{t,K}\|_\infty \le \frac{1}{1-\gamma}$. The induction is now complete and we have $\|v^i_t\|_\infty \le \frac{1}{1-\gamma}$ and $\|q^i_{t,0}\|_\infty \le \frac{1}{1-\gamma}$ for all $t \ge 0$.

(2) We first use induction to show that, given $t \ge 0$, if $\min_{s,a^i} \pi^i_{t,0}(a^i \mid s) \ge \ell_\tau$, then we have $\min_{s,a^i} \pi^i_{t,k}(a^i \mid s) \ge \ell_\tau$ for all $k \in \{0, 1, \cdots, K\}$. Since $\min_{s,a^i} \pi^i_{t,0}(a^i \mid s) \ge \ell_\tau$ by our assumption, we have the base case. Now suppose that $\min_{s \in \mathcal{S}, a^i \in \mathcal{A}^i} \pi^i_{t,k}(a^i \mid s) \ge \ell_\tau$ for some $k \ge 0$. Then we have by Algorithm 2 Line 4 that

$$\begin{aligned}
\pi^i_{t,k+1}(a^i \mid s) &= (1 - \beta_k) \pi^i_{t,k}(a^i \mid s) + \beta_k \sigma_\tau(q^i_{t,k}(s))(a^i) \\
&\ge (1 - \beta_k)\ell_\tau + \beta_k \ell_\tau \\
&= \ell_\tau,
\end{aligned}$$

where the inequality follows from (1) the induction hypothesis, and (2) $\sigma_\tau(q^i_{t,k}(s))(a^i) \ge \ell_\tau$, which follows from Lemma D.1 (1) and Lemma C.1. The induction is complete.

We next again use induction to show that $\min_{s,a^i} \pi^i_{t,0}(a^i \mid s) \ge \ell_\tau$ for all $t \in \{0, 1, \cdots, T\}$. Since $\min_{s,a^i} \pi^i_{0,0}(a^i \mid s)$ is initialized as a uniform policy, we have the base case. Now suppose that $\min_{s,a^i} \pi^i_{t,0}(a^i \mid s) \ge \ell_\tau$ for some $t \ge 0$. Since this implies that $\min_{s,a^i} \pi^i_{t,k}(a^i \mid s) \ge \ell_\tau$ for all $k \in \{0, 1, \cdots, K\}$, and in addition, $\pi^i_{t+1,0} = \pi^i_{t,K}$ according to Algorithm 2 Line 9, we have $\min_{s,a^i} \pi^i_{t+1,0}(a^i \mid s) \ge \ell_\tau$. The induction is complete.

$\square$

## D.3 Bounding the Nash Gap

Our ultimate goal is to bound the Nash gap

$$\mathrm{NG}(\pi^1_{T,K}, \pi^2_{T,K}) = \sum_{i=1,2} \left( \max_{\pi^i} U^i(\pi^i, \pi^{-i}_{T,K}) - U^i(\pi^i_{T,K}, \pi^{-i}_{T,K}) \right). \tag{28}$$

We first bound the Nash gap using the value functions of the output policies from Algorithm 2.

**Lemma D.2.** *It holds that*

$$\sum_{i=1,2} \left( \max_{\pi^i} U^i(\pi^i, \pi^{-i}_{T,K}) - U^i(\pi^i_{T,K}, \pi^{-i}_{T,K}) \right) \le \sum_{i=1,2} \left\| v^i_{*, \pi^{-i}_{T,K}} - v^i_{\pi^i_{T,K}, \pi^{-i}_{T,K}} \right\|_\infty. \tag{29}$$

*Proof of Lemma D.2.* Using the definition of the expected value functions, we have

$$\sum_{i=1,2} \left( \max_{\pi^i} U^i(\pi^i, \pi^{-i}_{T,K}) - U^i(\pi^i_{T,K}, \pi^{-i}_{T,K}) \right)$$

$$
\begin{aligned}
&= \sum_{i=1,2} \left( \max_{\pi^i} \mathbb{E}_{S \sim p_o} \left[ v^i_{\pi^i, \pi^{-i}_{T,K}}(S) - v^i_{\pi^i_{T,K}, \pi^{-i}_{T,K}}(S) \right] \right) \\
&\leq \sum_{i=1,2} \left( \mathbb{E}_{S \sim p_o} \left[ \max_{\pi^i} v^i_{\pi^i, \pi^{-i}_{T,K}}(S) - v^i_{\pi^i_{T,K}, \pi^{-i}_{T,K}}(S) \right] \right) \qquad \text{(Jensen's inequality)} \\
&= \sum_{i=1,2} \left( \mathbb{E}_{S \sim p_o} \left[ v^i_{*, \pi^{-i}_{T,K}}(S) - v^i_{\pi^i_{T,K}, \pi^{-i}_{T,K}}(S) \right] \right) \\
&\leq \sum_{i=1,2} \left\| v^i_{*, \pi^{-i}_{T,K}} - v^i_{\pi^i_{T,K}, \pi^{-i}_{T,K}} \right\|_{\infty}.
\end{aligned}
$$

$\qquad\qquad\qquad\qquad\qquad\qquad\qquad\qquad\qquad\qquad\qquad\qquad\qquad\qquad\qquad\qquad\qquad$ $\square$

The next lemma bounds the RHS of Eq. (29) using the actual iterates generated by Algorithm 2.

**Lemma D.3.** *It holds for $i \in \{1,2\}$ that*

$$
\left\| v^i_{*, \pi^{-i}_{t,K}} - v^i_{\pi^i_{t,K}, \pi^{-i}_{t,K}} \right\|_{\infty} \leq \frac{2}{1-\gamma} \left( 2\mathcal{L}_{sum}(T) + \mathcal{L}_v(T) + \mathcal{L}_\pi(T,K) + 2\tau \log(A_{\max}) \right).
$$

*Proof of Lemma D.3.* For any $s \in \mathcal{S}$ and $i \in \{1,2\}$, we have

$$
\begin{aligned}
0 &\leq \left| v^i_{*, \pi^{-i}_{T,K}}(s) - v^i_{\pi^i_{T,K}, \pi^{-i}_{T,K}}(s) \right| \\
&= v^i_{*, \pi^{-i}_{T,K}}(s) - v^i_{\pi^i_{T,K}, \pi^{-i}_{T,K}}(s) \\
&\leq v^i_{*, \pi^{-i}_{T,K}}(s) - v^i_{\pi^i_{T,K}, *}(s) \\
&= -v^{-i}_{\pi^{-i}_{T,K}, *}(s) - v^i_{\pi^i_{T,K}, *}(s) \\
&= v^i_*(s) - v^{-i}_{\pi^{-i}_{T,K}, *}(s) + v^{-i}_*(s) - v^i_{\pi^i_{T,K}, *}(s) \\
&\leq \sum_{j=1,2} \left\| v^{-j}_* - v^{-j}_{\pi^{-j}_{T,K}, *} \right\|_{\infty}.
\end{aligned}
$$

Since the RHS of the previous inequality does not depend on $s$, we have for $i \in \{1,2\}$ that

$$
\left\| v^i_{*, \pi^{-i}_{T,K}} - v^i_{\pi^i_{T,K}, \pi^{-i}_{T,K}} \right\|_{\infty} \leq \sum_{j=1,2} \left\| v^{-j}_* - v^{-j}_{\pi^{-j}_{T,K}, *} \right\|_{\infty}. \tag{30}
$$

It remains to bound the RHS of the previous inequality. Observe that for any $s \in \mathcal{S}$ and $i \in \{1,2\}$, we have

$$
\begin{aligned}
0 &\leq v^{-i}_*(s) - v^{-i}_{\pi^{-i}_{T,K}, *}(s) \\
&= v^i_{*, \pi^{-i}_{T,K}}(s) - v^i_*(s) \\
&= \max_{\mu^i \in \Delta(\mathcal{A}^i)} (\mu^i)^\top \mathcal{T}^i(v^i_{*, \pi^{-i}_{T,K}})(s)\pi^{-i}_{T,K}(s) - \max_{\mu^i \in \Delta(\mathcal{A}^i)} \min_{\mu^{-i} \in \Delta(\mathcal{A}^{-i})} (\mu^i)^\top \mathcal{T}^i(v^i_*)(s)\mu^{-i} \\
&\leq \left| \max_{\mu^i \in \Delta(\mathcal{A}^i)} (\mu^i)^\top \mathcal{T}^i(v^i_{*, \pi^{-i}_{T,K}})(s)\pi^{-i}_{T,K}(s) - \max_{\mu^i \in \Delta(\mathcal{A}^i)} (\mu^i)^\top \mathcal{T}^i(v^i_*)(s)\pi^{-i}_{T,K}(s) \right| \\
&\quad + \left| \max_{\mu^i \in \Delta(\mathcal{A}^i)} (\mu^i)^\top \mathcal{T}^i(v^i_*)(s)\pi^{-i}_{T,K}(s) - \max_{\mu^i \in \Delta(\mathcal{A}^i)} \min_{\mu^{-i} \in \Delta(\mathcal{A}^{-i})} (\mu^i)^\top \mathcal{T}^i(v^i_*)(s)\mu^{-i} \right| \\
&\leq \max_{\mu^i \in \Delta(\mathcal{A}^i)} \left| (\mu^i)^\top (\mathcal{T}^i(v^i_{*, \pi^{-i}_{T,K}})(s) - \mathcal{T}^i(v^i_*)(s))\pi^{-i}_{T,K}(s) \right| \\
&\quad + \left| \max_{\mu^i \in \Delta(\mathcal{A}^i)} (\mu^i)^\top \mathcal{T}^i(v^i_*)(s)\pi^{-i}_{T,K}(s) - \max_{\mu^i \in \Delta(\mathcal{A}^i)} (\mu^i)^\top \mathcal{T}^i(v^i_T)(s)\pi^{-i}_{T,K}(s) \right| \\
&\quad + \max_{\mu^i \in \Delta(\mathcal{A}^i)} (\mu^i)^\top \mathcal{T}^i(v^i_T)(s)\pi^{-i}_{T,K}(s) - \max_{\mu^i \in \Delta(\mathcal{A}^i)} \min_{\mu^{-i} \in \Delta(\mathcal{A}^{-i})} (\mu^i)^\top \mathcal{T}^i(v^i_T)(s)\mu^{-i}
\end{aligned}
$$

$$+\left|\max_{\mu^i\in\Delta(\mathcal{A}^i)}\min_{\mu^{-i}\in\Delta(\mathcal{A}^{-i})}(\mu^i)^\top\mathcal{T}^i(v_T^i)(s)\mu^{-i}-\max_{\mu^i\in\Delta(\mathcal{A}^i)}\min_{\mu^{-i}\in\Delta(\mathcal{A}^{-i})}(\mu^i)^\top\mathcal{T}^i(v_*^i)(s)\mu^{-i}\right|.$$
$$\tag{31}$$

We next bound each term on the RHS of the previous inequality.

**The $1$st Term on the RHS of Eq. (31).** Using the definition of $\mathcal{T}^i(\cdot)$, we have

$$\max_{\mu^i\in\Delta(\mathcal{A}^i)}\left|(\mu^i)^\top(\mathcal{T}^i(v^i_{*,\pi^{-i}_{T,K}})(s)-\mathcal{T}^i(v_*^i)(s))\pi^{-i}_{T,K}(s)\right|$$

$$\leq\max_{s,a^i,a^{-i}}\left|\mathcal{T}^i(v^i_{*,\pi^{-i}_{T,K}})(s,a^i,a^{-i})-\mathcal{T}^i(v_*^i)(s,a^i,a^{-i})\right|$$

$$=\gamma\max_{s,a^i,a^{-i}}\left|\mathbb{E}\left[v_*^i(S_1)-v^i_{*,\pi^{-i}_{T,K}}(S_1)\ \Big|\ S_0=s,A_0^i=a^i,A_0^{-i}=a^{-i}\right]\right|$$

$$\leq\gamma\max_{s,a^i,a^{-i}}\mathbb{E}\left[\left|v_*^i(S_1)-v^i_{*,\pi^{-i}_{T,K}}(S_1)\right|\ \Big|\ S_0=s,A_0^i=a^i,A_0^{-i}=a^{-i}\right]$$

$$\leq\gamma\left\|v_*^i-v^i_{*,\pi^{-i}_{T,K}}\right\|_\infty.$$

**The $2$nd Term on the RHS of Eq. (31).** Using the definition of $\mathcal{T}^i(\cdot)$, we have

$$\left|\max_{\mu^i\in\Delta(\mathcal{A}^i)}(\mu^i)^\top\mathcal{T}^i(v_*^i)(s)\pi^{-i}_{T,K}(s)-\max_{\mu^i\in\Delta(\mathcal{A}^i)}(\mu^i)^\top\mathcal{T}^i(v_T^i)(s)\pi^{-i}_{T,K}(s)\right|$$

$$\leq\max_{\mu^i\in\Delta(\mathcal{A}^i)}\left|(\mu^i)^\top(\mathcal{T}^i(v_*^i)(s)-\mathcal{T}^i(v_T^i)(s))\pi^{-i}_{T,K}(s)\right|$$

$$\leq\max_{s,a^i,a^{-i}}\left|\mathcal{T}^i(v_*^i)(s,a^i,a^{-i})-\mathcal{T}^i(v_T^i)(s,a^i,a^{-i})\right|$$

$$\leq\gamma\left\|v_*^i-v_T^i\right\|_\infty.$$

**The $3$rd Term on the RHS of Eq. (31).** Bounding the third term requires more effort. To begin with, we decompose it in the following way:

$$\max_{\mu^i\in\Delta(\mathcal{A}^i)}(\mu^i)^\top\mathcal{T}^i(v_T^i)(s)\pi^{-i}_{T,K}(s)-\max_{\mu^i\in\Delta(\mathcal{A}^i)}\min_{\mu^{-i}\in\Delta(\mathcal{A}^{-i})}(\mu^i)^\top\mathcal{T}^i(v_T^i)(s)\mu^{-i}$$

$$\leq\left|\max_{\mu^i\in\Delta(\mathcal{A}^i)}(\mu^i)^\top\mathcal{T}^i(v_T^i)(s)\pi^{-i}_{T,K}(s)-\min_{\mu^{-i}\in\Delta(\mathcal{A}^{-i})}\pi^i_{T,K}(s)\mathcal{T}^i(v_T^i)(s)\mu^{-i}\right|$$

$$\leq\left|\max_{\mu^{-i}\in\Delta(\mathcal{A}^{-i})}(\mu^{-i})^\top\mathcal{T}^{-i}(v_T^{-i})(s)\pi^i_{T,K}(s)+\min_{\mu^{-i}\in\Delta(\mathcal{A}^{-i})}(\mu^{-i})^\top\mathcal{T}^i(v_T^i)(s)^\top\pi^i_{T,K}(s)\right|$$

$$+\left|\sum_{i=1,2}\max_{\mu^i\in\Delta(\mathcal{A}^i)}(\mu^i)^\top\mathcal{T}^i(v_T^i)(s)\pi^{-i}_{T,K}(s)\right|.$$
$$\tag{32}$$

We next bound each term on the RHS of the previous inequality. For the first term, we have by definition of $\mathcal{T}^i(\cdot)$ that

$$\left|\max_{\mu^{-i}\in\Delta(\mathcal{A}^{-i})}(\mu^{-i})^\top\mathcal{T}^{-i}(v_T^{-i})(s)\pi^i_{T,K}(s)+\min_{\mu^{-i}\in\Delta(\mathcal{A}^{-i})}(\mu^{-i})^\top\mathcal{T}^i(v_T^i)(s)^\top\pi^i_{T,K}(s)\right|$$

$$=\left|\max_{\mu^{-i}\in\Delta(\mathcal{A}^{-i})}(\mu^{-i})^\top\mathcal{T}^{-i}(v_T^{-i})(s)\pi^i_{T,K}(s)-\max_{\mu^{-i}\in\Delta(\mathcal{A}^{-i})}(\mu^{-i})^\top[-\mathcal{T}^i(v_T^i)(s)]^\top\pi^i_{T,K}(s)\right|$$

$$\leq\max_{\mu^{-i}\in\Delta(\mathcal{A}^{-i})}\left|(\mu^{-i})^\top(\mathcal{T}^{-i}(v_T^{-i})(s)+\mathcal{T}^i(v_T^i)(s)^\top)\pi^i_{T,K}(s)\right|$$

$$\leq\max_{s,a^i,a^{-i}}\left|\mathcal{T}^{-i}(v_T^{-i})(s,a^i,a^{-i})+\mathcal{T}^i(v_T^i)(s,a^i,a^{-i})\right|$$

$$=\gamma\max_{s,a^i,a^{-i}}\left|\mathbb{E}\left[v_T^{-i}(S_1)+v_T^i(S_1)\ \Big|\ S_0=s,A_0^i=a^i,A_0^{-i}=a_0^{-i}\right]\right|$$

$$\leq\gamma\left\|v_T^{-i}+v_T^i\right\|_\infty.$$

For the second term on the RHS of Eq. (32), using the Lyapunov function $V_{v,s}(\cdot, \cdot)$ (defined in Appendix D.1), we have

$$\left| \sum_{i=1,2} \max_{\mu^i \in \Delta(\mathcal{A}^i)} (\mu^i)^\top \mathcal{T}^i(v_T^i)(s) \pi_{T,K}^{-i}(s) \right|$$

$$= \sum_{i=1,2} \max_{\mu^i \in \Delta(\mathcal{A}^i)} (\mu^i - \pi_{T,K}^i(s))^\top \mathcal{T}^i(v_T^i)(s) \pi_{T,K}^{-i}(s) + \left| \sum_{i=1,2} (\pi_{T,K}^i(s))^\top \mathcal{T}^i(v_T^i)(s) \pi_{T,K}^{-i}(s) \right|$$

$$\leq \sum_{i=1,2} \max_{\mu^i \in \Delta(\mathcal{A}^i)} \left\{ (\mu^i - \pi_{T,K}^i(s))^\top \mathcal{T}^i(v_T^i)(s) \pi_{T,K}^{-i}(s) + \tau \nu(\mu^i) - \tau \nu(\pi_{T,K}^i(s)) \right\}$$

$$+ 2\tau \log(A_{\max}) + \left| \sum_{i=1,2} (\pi_{T,K}^i(s))^\top \mathcal{T}^i(v_T^i)(s) \pi_{T,K}^{-i}(s) \right|$$

$$\leq V_{v_T,s}(\pi_{T,K}^i(s), \pi_{T,K}^{-i}(s)) + 2\tau \log(A_{\max})$$

$$+ \max_{s, a^i, a^{-i}} \left| \mathcal{T}^i(v_T^i)(s, a^i, a^{-i}) + \mathcal{T}^{-i}(v_T^{-i})(s, a^i, a^{-i}) \right|$$

$$\leq V_{v_T,s}(\pi_{T,K}^i(s), \pi_{T,K}^{-i}(s)) + 2\tau \log(A_{\max}) + \gamma \|v_T^i + v_T^{-i}\|_\infty.$$

Using the previous two inequalities together in Eq. (32), we obtain

$$\max_{\mu^i \in \Delta(\mathcal{A}^i)} (\mu^i)^\top \mathcal{T}^i(v_T^i)(s) \pi_{T,K}^{-i}(s) - \max_{\mu^i \in \Delta(\mathcal{A}^i)} \min_{\mu^{-i} \in \Delta(\mathcal{A}^{-i})} (\mu^i)^\top \mathcal{T}^i(v_T^i)(s) \mu^{-i}$$

$$\leq V_{v_T,s}(\pi_{T,K}^i(s), \pi_{T,K}^{-i}(s)) + 2\gamma \|v_T^i + v_T^{-i}\|_\infty + 2\tau \log(A_{\max}). \tag{33}$$

**The 4th Term on the RHS of Eq. (31).** Using the definition of $\mathcal{T}^i(\cdot)$, we have

$$\left| \max_{\mu^i \in \Delta(\mathcal{A}^i)} \min_{\mu^{-i} \in \Delta(\mathcal{A}^{-i})} (\mu^i)^\top \mathcal{T}^i(v_T^i)(s) \mu^{-i} - \max_{\mu^i \in \Delta(\mathcal{A}^i)} \min_{\mu^{-i} \in \Delta(\mathcal{A}^{-i})} (\mu^i)^\top \mathcal{T}^i(v_*^i)(s) \mu^{-i} \right|$$

$$\leq \max_{\mu^i \in \Delta(\mathcal{A}^i)} \min_{\mu^{-i} \in \Delta(\mathcal{A}^{-i})} \left| (\mu^i)^\top (\mathcal{T}^i(v_T^i)(s) - \mathcal{T}^i(v_*^i)(s)) \mu^{-i} \right|$$

$$\leq \max_{s, a^i, a^{-i}} \left| \mathcal{T}^i(v_T^i)(s, a^i, a^{-i}) - \mathcal{T}^i(v_*^i)(s, a^i, a^{-i}) \right|$$

$$\leq \gamma \|v_T^i - v_*^i\|_\infty.$$

Finally, using the upper bounds we obtained for all the terms on the RHS of Eq. (31), we have

$$\left\| v_*^{-i} - v_{\pi_{T,K}^{-i}, *}^{-i} \right\|_\infty \leq \gamma \left\| v_{*, \pi_{T,K}^{-i}}^i - v_*^i \right\|_\infty + 2\gamma \|v_T^i + v_T^{-i}\|_\infty + 2\gamma \|v_T^i - v_*^i\|_\infty$$

$$+ \max_{s \in \mathcal{S}} V_{v_T,s}(\pi_{T,K}^i(s), \pi_{T,K}^{-i}(s)) + 2\tau \log(A_{\max})$$

$$\leq \gamma \|v_*^{-i} - v_{\pi_{T,K}^{-i}, *}^{-i}\|_\infty + 2\|v_T^i + v_T^{-i}\|_\infty + 2\|v_T^i - v_*^i\|_\infty$$

$$+ \max_{s \in \mathcal{S}} V_{v_T,s}(\pi_{T,K}^i(s), \pi_{T,K}^{-i}(s)) + 2\tau \log(A_{\max}).$$

Rearranging terms and using $\mathcal{L}_{\text{sum}}(t)$ and $\mathcal{L}_\pi(t,k)$ to simplify the notation, we obtain

$$\left\| v_*^{-i} - v_{\pi_{T,K}^{-i}, *}^{-i} \right\|_\infty \leq \frac{1}{1-\gamma} \left( 2\mathcal{L}_{\text{sum}}(T) + 2\|v_T^i - v_*^i\|_\infty + \mathcal{L}_\pi(T, K) + 2\tau \log(A_{\max}) \right).$$

Summing up both sides of the previous inequality for $i \in \{1, 2\}$, we have

$$\sum_{i=1,2} \left\| v_*^{-i} - v_{\pi_{T,K}^{-i}, *}^{-i} \right\|_\infty \leq \frac{2}{1-\gamma} \left( 2\mathcal{L}_{\text{sum}}(T) + \mathcal{L}_v(T) + \mathcal{L}_\pi(T, K) + 2\tau \log(A_{\max}) \right).$$

Using the previous inequality in Eq. (30), we have the desired result. $\qquad \square$

Combining the results in Lemma D.2 and Lemma D.3 in Eq. (28), we have the following result, which bounds the Nash gap in terms of the Lyapunov functions defined in Appendix D.1.

**Lemma D.4.** *It holds that*

$$NG(\pi_{T,K}^1, \pi_{T,K}^2) \leq \frac{4}{1-\gamma} \left( 2\mathcal{L}_{sum}(T) + \mathcal{L}_v(T) + \mathcal{L}_\pi(T,K) + 2\tau \log(A_{\max}) \right).$$

The next step is to bound the Lyapunov functions, which require us to analyze the outer loop and inner loop of Algorithm 2.

### D.4 Analysis of the Outer Loop: $v$-Function Update

We first consider $\mathcal{L}_v(t)$ and establish a one-step Lyapunov drift inequality for it.

**Lemma D.5.** *It holds for all $t \geq 0$ that*

$$\mathcal{L}_v(t+1) \leq \gamma \mathcal{L}_v(t) + 4\mathcal{L}_{sum}(t) + 2\mathcal{L}_q^{1/2}(t,K) + 4\mathcal{L}_\pi(t,K) + 6\tau \log(A_{\max}). \tag{34}$$

*Proof of Lemma D.5.* For $i \in \{1,2\}$, using the outer-loop update equation (cf. Algorithm 2 Line 8) and the fact that $\mathcal{B}^i(v_*^i) = v_*^i$, we have for any $t \geq 0$ and $s \in \mathcal{S}$ that

$$v_{t+1}^i(s) - v_*^i(s) = \pi_{t,K}^i(s)^\top q_{t,K}^i(s) - v_*^i(s)$$
$$= \mathcal{B}^i(v_t^i)(s) - \mathcal{B}^i(v_*^i)(s) + \pi_{t,K}^i(s)^\top q_{t,K}^i(s) - \mathcal{B}^i(v_t^i)(s).$$

Since the minimax Bellman operator $\mathcal{B}^i(\cdot)$ is a $\gamma$-contraction mapping with respect to $\|\cdot\|_\infty$, we have from the previous inequality that

$$\left| v_{t+1}^i(s) - v_*^i(s) \right| \leq \left| \mathcal{B}^i(v_t^i)(s) - \mathcal{B}^i(v_*^i)(s) \right| + \left| \pi_{t,K}^i(s)^\top q_{t,K}^i(s) - \mathcal{B}^i(v_t^i)(s) \right|$$
$$\leq \left\| \mathcal{B}^i(v_t^i) - \mathcal{B}^i(v_*^i) \right\|_\infty + \left| \pi_{t,K}^i(s)^\top q_{t,K}^i(s) - \mathcal{B}^i(v_t^i)(s) \right|$$
$$\leq \gamma \left\| v_t^i - v_*^i \right\|_\infty + \left| \pi_{t,K}^i(s)^\top q_{t,K}^i(s) - \mathcal{B}^i(v_t^i)(s) \right|. \tag{35}$$

It remains to bound the second term on the RHS of Eq. (35). Using the definition of $\mathcal{B}^i(\cdot)$, we have

$$\left| \pi_{t,K}^i(s)^\top q_{t,K}^i(s) - \mathcal{B}^i(v_t^i)(s) \right|$$
$$= \left| \pi_{t,K}^i(s)^\top q_{t,K}^i(s) - \max_{\mu^i \in \Delta(\mathcal{A}^i)} \min_{\mu^{-i} \in \Delta(\mathcal{A}^{-i})} (\mu^i)^\top \mathcal{T}^i(v_t^i)(s) \mu^{-i} \right|$$
$$\leq \left| \max_{\mu^i \in \Delta(\mathcal{A}^i)} (\mu^i)^\top \mathcal{T}^i(v_t^i)(s) \pi_{t,K}^{-i}(s) - \pi_{t,K}^i(s)^\top q_{t,K}^i(s) \right|$$
$$\quad + \left| \max_{\mu^i \in \Delta(\mathcal{A}^i)} (\mu^i)^\top \mathcal{T}^i(v_t^i)(s) \pi_{t,K}^{-i}(s) - \max_{\mu^i \in \Delta(\mathcal{A}^i)} \min_{\mu^{-i} \in \Delta(\mathcal{A}^{-i})} (\mu^i)^\top \mathcal{T}^i(v_t^i)(s) \mu^{-i} \right|$$
$$\leq \max_{\mu^i \in \Delta(\mathcal{A}^i)} (\mu^i - \pi_{t,K}^i(s))^\top \mathcal{T}^i(v_t^i)(s) \pi_{t,K}^{-i}(s)$$
$$\quad + \left| (\pi_{t,K}^i(s))^\top (\mathcal{T}^i(v_t^i)(s) \pi_{t,K}^{-i}(s) - q_{t,K}^i(s)) \right|$$
$$\quad + \left| \max_{\mu^i \in \Delta(\mathcal{A}^i)} (\mu^i)^\top \mathcal{T}^i(v_t^i)(s) \pi_{t,K}^{-i}(s) - \max_{\mu^i \in \Delta(\mathcal{A}^i)} \min_{\mu^{-i} \in \Delta(\mathcal{A}^{-i})} (\mu^i)^\top \mathcal{T}^i(v_t^i)(s) \mu^{-i} \right|$$
$$\leq \left\| \mathcal{T}^i(v_t^i)(s) \pi_{t,K}^{-i}(s) - q_{t,K}^i(s) \right\|_\infty + 2V_{v_t,s}(\pi_{t,K}^1(s), \pi_{t,K}^2(s))$$
$$\quad + 2\gamma \| v_t^1 + v_t^2 \|_\infty + 3\tau \log(A_{\max}),$$

where the last line follows from Eq. (33). Using the previous inequality in Eq. (35), we obtain

$$\left\| v_{t+1}^i - v_*^i \right\|_\infty \leq \gamma \left\| v_t^i - v_*^i \right\|_\infty + \max_{s \in \mathcal{S}} \left\| \mathcal{T}^i(v_t^i)(s) \pi_{t,K}^{-i}(s) - q_{t,K}^i(s) \right\|_\infty$$
$$\quad + 2 \max_{s \in \mathcal{S}} V_{v_t,s}(\pi_{t,K}^1(s), \pi_{t,K}^2(s)) + 2\gamma \| v_t^1 + v_t^2 \|_\infty + 3\tau \log(A_{\max}).$$

Summing up both sides of the previous inequality for $i \in \{1,2\}$, we have

$$\mathcal{L}_v(t+1) \leq \gamma \mathcal{L}_v(t) + 4\mathcal{L}_{sum}(t) + 4\mathcal{L}_\pi(t,K) + 6\tau \log(A_{\max})$$

$$+ \sum_{i=1,2} \max_{s \in \mathcal{S}} \left\| \mathcal{T}^i(v_t^i)(s)\pi_{t,K}^{-i}(s) - q_{t,K}^i(s) \right\|_\infty.$$

To bound the last term on the RHS of the previous inequality, observe that

$$
\begin{aligned}
\sum_{i=1,2} \max_{s \in \mathcal{S}} \left\| \mathcal{T}^i(v_t^i)(s)\pi_{t,K}^{-i}(s) - q_{t,K}^i(s) \right\|_\infty &= \sum_{i=1,2} \left\| \bar{q}_k^i - q_{t,K}^i \right\|_\infty \\
&\le \sum_{i=1,2} \left\| \bar{q}_k^i - q_{t,K}^i \right\|_2 \\
&\le \left( 2 \sum_{i=1,2} \left\| \bar{q}_k^i - q_{t,K}^i \right\|_2^2 \right)^{1/2} \qquad (a^2 + b^2 \ge 2ab) \\
&\le 2\mathcal{L}_q^{1/2}(t, K). \qquad\qquad (36)
\end{aligned}
$$

Therefore, we have

$$\mathcal{L}_v(t+1) \le \gamma \mathcal{L}_v(t) + 4\mathcal{L}_{\text{sum}}(t) + 2\mathcal{L}_q^{1/2}(t,K) + 4\mathcal{L}_\pi(t,K) + 6\tau \log(A_{\max}).$$

The proof is complete. $\qquad\qquad\qquad\qquad\qquad\qquad\qquad\qquad\qquad\qquad\qquad\qquad\quad\square$

We next establish a one-step Lyapunov drift inequality for $\mathcal{L}_{\text{sum}}(t)$ in the following lemma.

**Lemma D.6.** *It holds for all $t \ge 0$ that $\mathcal{L}_{sum}(t+1) \le \gamma \mathcal{L}_{sum}(t) + 2\mathcal{L}_q^{1/2}(t,K)$.*

*Proof of Lemma D.6.* Using the outer-loop update equation (cf. Algorithm 2 Line 8), we have for any $t \ge 0$ and $s \in \mathcal{S}$ that

$$
\begin{aligned}
\left| v_{t+1}^1(s) + v_{t+1}^2(s) \right| &= \left| \sum_{i=1,2} \pi_{t,K}^i(s)^\top q_{t,K}^i(s) \right| \\
&\le \left| \sum_{i=1,2} \pi_{t,K}^i(s)^\top \left( q_{t,K}^i(s) - \mathcal{T}^i(v_t^i)(s)\pi_{t,K}^{-i}(s) \right) \right| \\
&\quad + \left| \sum_{i=1,2} \pi_{t,K}^i(s)\mathcal{T}^i(v_t^i)(s)\pi_{t,K}^{-i}(s) \right| \\
&\le \sum_{i=1,2} \max_{s \in \mathcal{S}} \| q_{t,K}^i(s) - \mathcal{T}^i(v_t^i)(s)\pi_{t,K}^{-i}(s) \|_\infty \\
&\quad + \sum_{i=1,2} \max_{(s,a^i,a^{-i})} \left| \mathcal{T}^i(v_t^i)(s,a^i,a^{-i}) + \mathcal{T}^{-i}(v_t^{-i})(s,a^i,a^{-i}) \right| \\
&\le \sum_{i=1,2} \max_{s \in \mathcal{S}} \| q_{t,K}^i(s) - \mathcal{T}^i(v_t^i)(s)\pi_{t,K}^{-i}(s) \|_\infty + \gamma \| v_t^1 + v_t^2 \|_\infty,
\end{aligned}
$$

where the last line follows from the definition of $\mathcal{T}^i(\cdot)$. Since the RHS of the previous inequality does not depend on $s$, we have

$$\| v_{t+1}^1 + v_{t+1}^2 \|_\infty \le \gamma \| v_t^1 + v_t^2 \|_\infty + \sum_{i=1,2} \max_{s \in \mathcal{S}} \| q_{t,K}^i(s) - \mathcal{T}^i(v_t^i)(s)\pi_{t,K}^{-i}(s) \|_\infty.$$

The result follows from using Eq. (36) to bound the last term on the RHS of the previous inequality and then using $\mathcal{L}_{\text{sum}}(t)$ and $\mathcal{L}_q(t,k)$ to simplify the notation. $\qquad\qquad\qquad\qquad\square$

## D.5   Analysis of the Inner Loop

In this section, we establish negative drift inequalities for the Lyapunov functions $\mathcal{L}_q(t,k)$ and $\mathcal{L}_\pi(t,k)$, which are defined in terms of the $q$-functions and the policies updated in the inner loop of Algorithm 2. For ease of presentation, we write down only the inner loop of Algorithm 2 in Algorithm 3, where we omit the subscript $t$, which is used as the index for the outer loop. Similarly, we will write $\mathcal{L}_q(k)$ for $\mathcal{L}_q(t,k)$ and $\mathcal{L}_\pi(k)$ for $\mathcal{L}_\pi(t,k)$. All results derived for the $q$-functions and policies of Algorithm 3 can be directly combined with the analysis of outer loop of Algorithm 2 using a conditioning argument together with the Markov property.

**Algorithm 3** Inner Loop of Algorithm 2: from Player $i$'s Perspective

---

1: **Input:** Integer $K$, initializations $q_0^i$ and $\pi_0^i$, and a $v$-function $v^i$ from the outer loop. Note that we have $\|q_0^i\|_\infty \leq \frac{1}{1-\gamma}$, $\|v^i\|_\infty \leq \frac{1}{1-\gamma}$, and $\min_{s,a^i} \pi_0^i(a^i \mid s) \geq \ell_\tau$ due to Lemma D.1.
2: **for** $k = 0, 1, \cdots, K - 1$ **do**
3:     $\pi_{k+1}^i(s) = \pi_k^i(s) + \beta_k(\sigma_\tau(q_k^i(s)) - \pi_k^i(s))$ for all $s \in \mathcal{S}$
4:     Sample $A_k^i \sim \pi_{k+1}^i(\cdot \mid S_k)$, and observe $S_{k+1} \sim p(\cdot \mid S_k, A_k^i, A_k^{-i})$
5:     $q_{k+1}^i(s, a^i) = q_k^i(s, a^i) + \alpha_k \mathbb{1}_{\{(S_k, A_k^i) = (s, a^i)\}} \left( R_i(S_k, A_k^i, A_k^{-i}) + \gamma v^i(S_{k+1}) - q_k^i(S_k, A_k^i) \right)$
    for all $(s, a^i) \in \mathcal{S} \times \mathcal{A}^i$
6: **end for**
7: **Output:** $q_K^i$ and $\pi_K^i$

---

### D.5.1 Analysis of the Policies

We consider $\{(\pi_k^1, \pi_k^2)\}_{k \geq 0}$ generated by Algorithm 3 and use $V_X(\cdot, \cdot)$ defined in Appendix D.1 as the Lyapunov function to study them. For simplicity of notation, we use $\nabla_1 V_X(\cdot, \cdot)$ (respectively, $\nabla_2 V_X(\cdot, \cdot)$) to denote the gradient with respect to the first argument (respectively, the second argument). Recall that Lemma D.1 implies that $\pi_k = (\pi_k^1, \pi_k^2) \in \Pi_\tau$ for all $k \geq 0$, where $\Pi_\tau = \{(\pi^1, \pi^2) \in \Delta(\mathcal{A}^1) \times \Delta(\mathcal{A}^2) \mid \min_{a^1 \in \mathcal{A}^1} \pi^1(a^1) \geq \ell_\tau, \min_{a^2 \in \mathcal{A}^2} \pi^2(a^2) \geq \ell_\tau\}$. The following lemma establishes the properties of $V_X(\cdot, \cdot)$.

**Lemma D.7.** *The function $V_X(\cdot, \cdot)$ has the following properties.*

*(1) For $i \in \{1, 2\}$, fix $\mu^{-i} \in \Delta(\mathcal{A}^{-i})$, the function $V_X(\mu^i, \mu^{-i})$ as a function of $\mu^i$ is $\tau$ – strongly convex with respect to $\|\cdot\|_2$.*

*(2) $V_X(\cdot, \cdot)$ is $\tilde{L}_\tau$ – smooth on $\Pi_\tau$, where $\tilde{L}_\tau = 2\left(\frac{\tau}{\ell_\tau} + \frac{\max(\|X_1\|_2^2, \|X_2\|_2^2)}{\tau} + \|X_1 + X_2^\top\|_2\right)$.*

*(3) It holds for any $(\mu^1, \mu^2) \in \Delta(\mathcal{A}^1) \times \Delta(\mathcal{A}^2)$ that*

$$\langle \nabla_1 V_X(\mu^1, \mu^2), \sigma_\tau(X_1 \mu^2) - \mu^1 \rangle + \langle \nabla_2 V_X(\mu^1, \mu^2), \sigma_\tau(X_2 \mu^1) - \mu^2 \rangle$$
$$\leq -\frac{7}{8} V_X(\mu^1, \mu^2) + \frac{16}{\tau} \|X_1 + X_2^\top\|_2^2.$$

*(4) For any $u^1 \in \mathbb{R}^{|\mathcal{A}^1|}$ and $u^2 \in \mathbb{R}^{|\mathcal{A}^2|}$, we have for all $(\mu^1, \mu^2) \in \Pi_\tau$ that*

$$\langle \nabla_1 V_X(\mu^1, \mu^2), \sigma_\tau(u^1) - \sigma_\tau(X_1 \mu^2) \rangle + \langle \nabla_2 V_X(\mu^1, \mu^2), \sigma_\tau(u^2) - \sigma_\tau(X_2 \mu^1) \rangle$$
$$\leq \frac{1}{8} V_X(\mu^1, \mu^2) + \frac{8}{\tau}\left(\frac{1}{\ell_\tau} + \frac{\max(\|X_1\|_2, \|X_2\|_2)}{\tau}\right)^2 \sum_{i=1,2} \|u^i - X_i \mu^{-i}\|_2^2.$$

*Proof of Lemma D.7.* To begin with, we have by Danskin's theorem [95] that

$$\nabla_1 V_X(\mu^1, \mu^2) = -(X_1 + X_2^\top)\mu^2 - \tau \nabla \nu(\mu^1) + X_2^\top \sigma_\tau(X_2 \mu^1). \tag{37}$$

Similar result holds for $\nabla_2 V_X(\mu^1, \mu^2)$.

(1) It is clear that the function $V_X(\cdot, \cdot)$ is non-negative. The strong convexity follows from the following two observations.

    (i) The negative entropy $-\nu(\cdot)$ is $1$ – strongly convex with respect to $\|\cdot\|_2$ [97, Example 5.27].

    (ii) Given $i \in \{1, 2\}$, the function $\max_{\hat{\mu}^{-i} \in \Delta(\mathcal{A}^{-i})} \left\{ (\hat{\mu}^{-i})^\top X_{-i} \mu^i + \tau \nu(\hat{\mu}^{-i}) \right\}$ as a function of $\mu^i$ is the maximum of linear functions in $\mu^i$, and therefore is convex.

It follows that, for any $i \in \{1, 2\}$, the function $V_X(\mu^1, \mu^2)$ is $\tau$ – strongly convex in $\mu^i$ with respect to $\|\cdot\|_2$ uniformly for all $\mu^{-i}$.

(2) For any $(\mu^1, \mu^2), (\bar{\mu}^1, \bar{\mu}^2) \in \Pi_\tau$, we have by Eq. (37) that

$$\left\|\nabla_1 V_X(\mu^1, \mu^2) - \nabla_1 V_X(\bar{\mu}^1, \bar{\mu}^2)\right\|_2$$
$$= \left\|(X_1 + X_2^\top)(\mu^2 - \bar{\mu}^2) + \tau(\nabla\nu(\mu^1) - \nabla\nu(\bar{\mu}^1)) + X_2^\top(\sigma_\tau(X_2\bar{\mu}^1) - \sigma_\tau(X_2\mu^1))\right\|_2$$
$$\leq \|X_1 + X_2^\top\|_2 \|\mu^2 - \bar{\mu}^2\|_2 + \left(\frac{\tau}{\ell_\tau} + \frac{\|X_2\|_2^2}{\tau}\right)\|\bar{\mu}^1 - \mu^1\|_2 \tag{38}$$

where Eq. (38) follows from Lemma C.5 and the Lipschitz continuity of $\sigma_\tau(\cdot)$ [96]. Similarly, we also have

$$\left\|\nabla_2 V_X(\mu^1, \mu^2) - \nabla_2 V_X(\bar{\mu}^1, \bar{\mu}^2)\right\|_2$$
$$\leq \|X_2 + X_1^\top\|_2 \|\mu^1 - \bar{\mu}^1\|_2 + \left(\frac{\tau}{\ell_\tau} + \frac{\|X_1\|_2^2}{\tau}\right)\|\bar{\mu}^2 - \mu^2\|_2.$$

Using the previous 2 inequalities, we have the following result for the full gradient of $V_X(\cdot, \cdot)$:

$$\left\|\nabla V_X(\mu^1, \mu^2) - \nabla V_X(\bar{\mu}^1, \bar{\mu}^2)\right\|_2^2$$
$$= \left\|\nabla_1 V_X(\mu^1, \mu^2) - \nabla_1 V_X(\bar{\mu}^1, \bar{\mu}^2)\right\|_2^2 + \left\|\nabla_2 V_X(\mu^1, \mu^2) - \nabla_2 V_X(\bar{\mu}^1, \bar{\mu}^2)\right\|_2^2$$
$$\leq \sum_{i=1,2}\left[2\left(\frac{\tau}{\ell_\tau} + \frac{\|X_{-i}\|_2^2}{\tau}\right)^2\|\bar{\mu}^i - \mu^i\|_2^2 + 2\|X_i + X_{-i}^\top\|_2^2\|\mu^{-i} - \bar{\mu}^{-i}\|_2^2\right]$$
$$\hspace{8cm} ((a+b)^2 \leq 2a^2 + 2b^2)$$
$$\leq 2\left[\left(\frac{\tau}{\ell_\tau} + \frac{\max(\|X_1\|_2^2, \|X_2\|_2^2)}{\tau}\right)^2 + \|X_1 + X_2^\top\|_2^2\right]\sum_{i=1,2}\|\bar{\mu}^i - \mu^i\|_2^2.$$

The previous inequality implies that $V_X(\cdot, \cdot)$ is an $\tilde{L}_\tau$-smooth function on $\Pi_\tau$ [97], where

$$\tilde{L}_\tau = 2\left(\frac{\tau}{\ell_\tau} + \frac{\max(\|X_1\|_2^2, \|X_2\|_2^2)}{\tau} + \|X_1 + X_2^\top\|_2\right).$$

(3) Using the formula for the gradient of $V_X(\cdot, \cdot)$ in Eq. (37), we have

$$\langle\nabla_1 V_X(\mu^1, \mu^2), \sigma_\tau(X_1\mu^2) - \mu^1\rangle$$
$$= \langle-(X_1 + X_2^\top)\mu^2 - \tau\nabla\nu(\mu^1) + X_2^\top\sigma_\tau(X_2\mu^1), \sigma_\tau(X_1\mu^2) - \mu^1\rangle$$
$$= \langle-(X_1 + X_2^\top)\mu^2 - \tau\nabla\nu(\mu^1) + X_2^\top\sigma_\tau(X_2\mu^1), \sigma_\tau(X_1\mu^2) - \mu^1\rangle$$
$$\quad + \langle X_1\mu^2 + \tau\nabla\nu(\sigma_\tau(X_1\mu^2)), \sigma_\tau(X_1\mu^2) - \mu^1\rangle \tag{39}$$
$$= \tau\langle\nabla\nu(\sigma_\tau(X_1\mu^2)) - \nabla\nu(\mu^1), \sigma_\tau(X_1\mu^2) - \mu^1\rangle$$
$$\quad + (\sigma_\tau(X_2\mu^1) - \mu^2)^\top X_2(\sigma_\tau(X_1\mu^2) - \mu^1),$$

where Eq. (39) is due to the first order optimality condition $X_1\mu^2 + \tau\nabla\nu(\sigma_\tau(X_1\mu^2)) = 0$. To proceed, observe that the concavity of $\nu(\cdot)$ and the optimality condition $X_1\mu^2 + \tau\nabla\nu(\sigma_\tau(X_1\mu^2)) = 0$ together imply that

$$\langle\nabla\nu(\sigma_\tau(X_1\mu^2)) - \nabla\nu(\mu^1), \sigma_\tau(X_1\mu^2) - \mu^1\rangle$$
$$= \langle\nabla\nu(\mu^1) - \nabla\nu(\sigma_\tau(X_1\mu^2)), \mu^1 - \sigma_\tau(X_1\mu^2)\rangle$$
$$= \langle\nabla\nu(\mu^1), \mu^1 - \sigma_\tau(X_1\mu^2)\rangle - \langle\nabla\nu(\sigma_\tau(X_1\mu^2)), \mu^1 - \sigma_\tau(X_1\mu^2)\rangle$$
$$\leq \nu(\mu^1) - \nu(\sigma_\tau(X_1\mu^2)) - \langle\nabla\nu(\sigma_\tau(X_1\mu^2)), \mu^1 - \sigma_\tau(X_1\mu^2)\rangle \quad \text{(Concavity of } \nu(\cdot))$$
$$= \nu(\mu^1) - \nu(\sigma_\tau(X_1\mu^2)) + \frac{1}{\tau}\langle X_1\mu^2, \mu^1 - \sigma_\tau(X_1\mu^2)\rangle$$
$$= \frac{1}{\tau}\left[(\mu^1)^\top X_1\mu^2 + \tau\nu(\mu^1) - \max_{\hat{\mu}^1 \in \Delta(\mathcal{A}^1)}\left\{(\hat{\mu}^1)^\top X_1\mu^2 + \tau\nu(\hat{\mu}^1)\right\}\right].$$

Therefore, we have from the previous 2 inequalities that

$$\langle\nabla_1 V_X(\mu^1, \mu^2), \sigma_\tau(X_1\mu^2) - \mu^1\rangle$$

$$\leq (\mu^1)^\top X_1 \mu^2 + \tau\nu(\mu^1) - \max_{\hat{\mu}^1 \in \Delta(\mathcal{A}^1)} \left\{ (\hat{\mu}^1)^\top X_1 \mu^2 + \tau\nu(\hat{\mu}^1) \right\}$$
$$+ (\sigma_\tau(X_2\mu^1) - \mu^2)^\top X_2(\sigma_\tau(X_i\mu^2) - \mu^1).$$

Similarly, we also have

$$\langle \nabla_2 V_X(\mu^1, \mu^2), \sigma_\tau(X_2\mu^1) - \mu^2 \rangle$$
$$\leq (\mu^2)^\top X_2\mu^1 + \tau\nu(\mu^2) - \max_{\hat{\mu}^2 \in \Delta(\mathcal{A}^2)} \left\{ (\hat{\mu}^2)^\top X_2\mu^1 + \tau\nu(\hat{\mu}^2) \right\}$$
$$+ (\sigma_\tau(X_1\mu^2) - \mu^1)^\top X_1(\sigma_\tau(X_2\mu^1) - \mu^2).$$

Adding up the previous 2 inequalities, we obtain

$$\langle \nabla_1 V_X(\mu^1, \mu^2), \sigma_\tau(X_1\mu^2) - \mu^1 \rangle + \langle \nabla_2 V_X(\mu^1, \mu^2), \sigma_\tau(X_2\mu^1) - \mu^2 \rangle$$
$$\leq -V_X(\mu^1, \mu^2) + (\sigma_\tau(X_1\mu^2) - \mu^1)^\top (X_1 + X_2^\top)(\sigma_\tau(X_2\mu^1) - \mu^2)$$
$$\leq -V_X(\mu^1, \mu^2) + \|\sigma_\tau(X_1\mu^2) - \mu^1\|_2 \|X_1 + X_2^\top\|_2 \|\sigma_\tau(X_2\mu^1) - \mu^2\|_2$$
$$\leq -V_X(\mu^1, \mu^2) + 2\|\sigma_\tau(X_1\mu^2) - \mu^1\|_2 \|X_1 + X_2^\top\|_2, \tag{40}$$

where the last line follows from $\|\sigma_\tau(X_2\mu^1) - \mu^2\|_2 \leq \|\sigma_\tau(X_2\mu^1)\|_1 + \|\mu^2\|_1 \leq 2$. Using Lemma D.7 (1) together with the quadratic growth property of strongly convex functions, we have

$$\|\sigma_\tau(X_1\mu^2) - \mu^1\|_2 \leq \frac{\sqrt{2}}{\sqrt{\tau}} V_X(\mu^1, \mu^2)^{1/2}.$$

It follows that

$$\langle \nabla_1 V_X(\mu^1, \mu^2), \sigma_\tau(X_1\mu^2) - \mu^1 \rangle + \langle \nabla_2 V_X(\mu^1, \mu^2), \sigma_\tau(X_2\mu^1) - \mu^2 \rangle$$
$$\leq -V_X(\mu^1, \mu^2) + 2\|\sigma_\tau(X_1\mu^2) - \mu^1)\|_2 \|X_1 + X_2^\top\|_2$$
$$\leq -V_X(\mu^1, \mu^2) + \frac{2\sqrt{2}}{\sqrt{\tau}} V_X(\mu^1, \mu^2)^{1/2} \|X_1 + X_2^\top\|_2$$
$$\leq -\frac{7}{8} V_X(\mu^1, \mu^2) + \frac{16}{\tau} \|X_1 + X_2^\top\|_2^2,$$

where the last line follows from $a^2 + b^2 \geq 2ab$.

(4) For any $u^1 \in \mathbb{R}^{|\mathcal{A}^1|}$, using the formula of the gradient of $V_X(\cdot, \cdot)$ from Eq. (37), we have

$$\langle \nabla_1 V_X(\mu^1, \mu^2), \sigma_\tau(u^1) - \sigma_\tau(X_1\mu^2) \rangle$$
$$= \langle -(X_1 + X_2^\top)\mu^2 - \tau\nabla\nu(\mu^1) + X_2^\top\sigma_\tau(X_2\mu^1), \sigma_\tau(u^1) - \sigma_\tau(X_1\mu^2) \rangle$$
$$= \langle -(X_1 + X_2^\top)\mu^2 - \tau\nabla\nu(\mu^1) + X_2^\top\sigma_\tau(X_2\mu^1), \sigma_\tau(u^1) - \sigma_\tau(X_1\mu^2) \rangle$$
$$\quad + \langle X_1\mu^2 + \tau\nabla\nu(\sigma_\tau(X_1\mu^2)), \sigma_\tau(u^1) - \sigma_\tau(X_1\mu^2) \rangle \quad \text{(First order optimality condition)}$$
$$= \tau\langle \nabla\nu(\sigma_\tau(X_1\mu^2)) - \nabla\nu(\mu^1), \sigma_\tau(u^1) - \sigma_\tau(X_1\mu^2) \rangle$$
$$\quad + (\sigma_\tau(X_2\mu^1) - \mu^2)^\top X_2(\sigma_\tau(u^1) - \sigma_\tau(X_1\mu^2))$$
$$\leq \left( \tau\|\nabla\nu(\sigma_\tau(X_1\mu^2)) - \nabla\nu(\mu^1)\|_2 + \|\sigma_\tau(X_2\mu^1) - \mu^2\|_2 \|X_2\|_2 \right) \|\sigma_\tau(u^1) - \sigma_\tau(X_1\mu^2)\|_2$$
$$\leq \left( \frac{\tau}{\ell_\tau} \|\sigma_\tau(X_1\mu^2) - \mu^1\|_2 + \|\sigma_\tau(X_2\mu^1) - \mu^2\|_2 \|X_2\|_2 \right) \frac{1}{\tau} \|u^1 - X_1\mu^2\|_2$$
$$\leq \frac{\sqrt{2}}{\sqrt{\tau}} \left( \frac{1}{\ell_\tau} + \frac{\|X_2\|_2}{\tau} \right) V_X(\mu^1, \mu^2)^{1/2} \|u^1 - X_1\mu^2\|_2$$
$$\leq \frac{1}{16} V_X(\mu^1, \mu^2) + \frac{8}{\tau} \left( \frac{1}{\ell_\tau} + \frac{\|X_2\|_2}{\tau} \right)^2 \|u^1 - X_1\mu^2\|_2^2,$$

where the third last inequality follows from the $\frac{1}{\ell_\tau}$-smoothness of $\nu(\cdot)$ on $\Pi_\tau$, the second last inequality follows from Lemma D.7 (1) together with the quadratic growth property of strongly

convex functions, and the last inequality follows from $a^2 + b^2 \geq 2ab$. Similarly, we also have for any $u^2 \in \mathbb{R}^{|\mathcal{A}^2|}$ that

$$
\langle \nabla_2 V_X(\mu^1, \mu^2), \sigma_\tau(u^2) - \sigma_\tau(X_2\mu^1) \rangle
$$

$$
\leq \frac{1}{16} V_X(\mu^1, \mu^2) + \frac{8}{\tau} \left( \frac{1}{\ell_\tau} + \frac{\|X_1\|_2}{\tau} \right)^2 \|u^2 - X_2\mu^1\|_2^2.
$$

Adding up the previous two inequalities, we obtain

$$
\langle \nabla_1 V_X(\mu^1, \mu^2), \sigma_\tau(u^1) - \sigma_\tau(X_1\mu^2) \rangle + \langle \nabla_2 V_X(\mu^1, \mu^2), \sigma_\tau(u^2) - \sigma_\tau(X_2\mu^1) \rangle
$$

$$
\leq \frac{1}{8} V_X(\mu^1, \mu^2) + \frac{8}{\tau} \left( \frac{1}{\ell_\tau} + \frac{\max(\|X_1\|_2, \|X_2\|_2)}{\tau} \right)^2 \sum_{i=1,2} \|u^i - X_i\mu^{-i}\|_2^2.
$$

The proof is complete.

$\square$

With the properties of $V_X(\cdot, \cdot)$ established in Lemma D.7, we now use it as a Lyapunov function to study $(\pi_k^1, \pi_k^2)$ generated by Algorithm 3. Specifically, using the smoothness of $V_X(\cdot, \cdot)$ (cf. Lemma D.7 (2)), the update equation in Algorithm 3 Line 3, and Lemma D.7 (3) and (4), we have the desired one-step Lyapunov drift inequality for $\mathcal{L}_\pi(k)$, which is presented in the following.

**Lemma D.8.** *The following inequality holds for all $k \geq 0$:*

$$
\mathcal{L}_\pi(k+1) \leq \left( 1 - \frac{3\beta_k}{4} \right) \mathcal{L}_\pi(k) + \frac{16 A_{\max}^2 \beta_k}{\tau} \|v^1 + v^2\|_\infty^2
$$

$$
+ \frac{32 A_{\max}^2 \beta_k}{\tau^3 \ell_\tau^2 (1-\gamma)^2} \mathcal{L}_q(k) + 2 L_\tau \beta_k^2.
$$

*Proof of Lemma D.8.* We will use $V_{v,s}(\cdot, \cdot)$ (see Lemma D.1) as the Lyapunov function to study the evolution of $(\pi_k^1(s), \pi_k^2(s))$. To begin with, we identify the smoothness parameter of $V_{v,s}(\cdot, \cdot)$. Using Lemma D.7 (1) and the definition of $V_{v,s}(\cdot, \cdot)$, we have

$$
\tilde{L}_\tau = 2 \left( \frac{\tau}{\ell_\tau} + \frac{\max(\|X_1\|_2^2, \|X_2\|_2^2)}{\tau} + \|X_1 + X_2^\top\|_2 \right)
$$

$$
= 2 \left( \frac{\tau}{\ell_\tau} + \frac{\max(\|\mathcal{T}^1(v^1)(s)\|_2^2, \|\mathcal{T}^2(v^2)(s)\|_2^2)}{\tau} + \|\mathcal{T}^1(v^1)(s) + \mathcal{T}^2(v^2)(s)^\top\|_2 \right)
$$

$$
\leq 2 \left( \frac{\tau}{\ell_\tau} + \frac{A_{\max}^2}{\tau(1-\gamma)^2} + \frac{2 A_{\max}}{1-\gamma} \right)
$$

$\qquad$ (This follows from $|\mathcal{T}^i(v^i)(s, a^i, a^{-i})| \leq \frac{1}{1-\gamma}, \forall (s, a^i, a^{-i})$ and $i \in \{1, 2\}$.)

$$
:= L_\tau.
$$

Therefore, $V_{v,s}(\cdot, \cdot)$ is an $L_\tau$ – smooth function on $\Pi_\tau$. Using the smoothness of $V_{v,s}(\cdot, \cdot)$, for any $s \in \mathcal{S}$, we have by the policy update equation (cf. Algorithm 3 Line 3) that

$$
V_{v,s}(\pi_{k+1}^1(s), \pi_{k+1}^2(s))
$$

$$
\leq V_{v,s}(\pi_k^1(s), \pi_k^2(s)) + \beta_k \langle \nabla_2 V_{v,s}(\pi_k^1(s), \pi_k^2(s)), \sigma_\tau(q_k^2(s)) - \pi_k^2(s) \rangle
$$

$$
+ \beta_k \langle \nabla_1 V_{v,s}(\pi_k^1(s), \pi_k^2(s)), \sigma_\tau(q_k^1(s)) - \pi_k^1(s) \rangle + \frac{L_\tau \beta_k^2}{2} \sum_{i=1,2} \|\sigma_\tau(q_k^1(s)) - \pi_k^1(s)\|_2^2
$$

$$
\leq V_{v,s}(\pi_k^1(s), \pi_k^2(s)) + \beta_k \langle \nabla_2 V_{v,s}(\pi_k^1(s), \pi_k^2(s)), \sigma_\tau(\mathcal{T}^2(v^2)(s)\pi_k^1(s)) - \pi_k^2(s) \rangle
$$

$$
+ \beta_k \langle \nabla_1 V_{v,s}(\pi_k^1(s), \pi_{k+1}^2(s)), \sigma_\tau(\mathcal{T}^1(v^1)(s)\pi_k^2(s)) - \pi_k^1(s) \rangle
$$

$$
+ \beta_k \langle \nabla_2 V_{v,s}(\pi_k^1(s), \pi_k^2(s)), \sigma_\tau(q_k^2(s)) - \sigma_\tau(\mathcal{T}^2(v^2)(s)\pi_k^1(s)) \rangle
$$

$$
+ \beta_k \langle \nabla_1 V_{v,s}(\pi_k^1(s), \pi_{k+1}^2(s)), \sigma_\tau(q_k^1(s)) - \sigma_\tau(\mathcal{T}^1(v^1)(s)\pi_k^2(s)) \rangle + 2 L_\tau \beta_k^2
$$

$$
\leq \left( 1 - \frac{3\beta_k}{4} \right) V_{v,s}(\pi_k^1(s), \pi_k^2(s)) + \frac{16 \beta_k}{\tau} \|\mathcal{T}^1(v^1)(s) + \mathcal{T}^2(v^2)(s)^\top\|_2^2
$$

$$+ \frac{8\beta_k}{\tau} \left( \frac{1}{\ell_\tau} + \frac{\max_{i \in \{1,2\}} \|\mathcal{T}^i(v^i)(s)\|_2}{\tau} \right)^2 \sum_{i=1,2} \|q_k^i(s) - \mathcal{T}^i(v^i)(s)\pi_k^2(s)\|_2^2 + 2L_\tau\beta_k^2.$$

where the last line follows from Lemma D.7 (3) and (4). Since $\max_{i \in \{1,2\}} \|\mathcal{T}^i(v^i)(s)\|_2 \leq \frac{A_{\max}}{1-\gamma}$ and

$$\|\mathcal{T}^1(v^1)(s) + \mathcal{T}^2(v^2)(s)^\top\|_2^2 \leq A_{\max}^2 \|v^1 + v^2\|_\infty^2,$$

we have

$$V_{v,s}(\pi_{k+1}^1(s), \pi_{k+1}^2(s))$$

$$\leq \left(1 - \frac{3\beta_k}{4}\right) V_{v,s}(\pi_k^1(s), \pi_k^2(s)) + \frac{16\beta_k A_{\max}^2}{\tau} \|v^1 + v^2\|_\infty^2$$

$$+ \frac{8\beta_k}{\tau} \left( \frac{1}{\ell_\tau} + \frac{A_{\max}}{\tau(1-\gamma)} \right)^2 \sum_{i=1,2} \|q_k^i(s) - \mathcal{T}^i(v^i)(s)\pi_k^2(s)\|_2^2 + 2L_\tau\beta_k^2$$

$$\leq \left(1 - \frac{3\beta_k}{4}\right) \max_{s \in \mathcal{S}} V_{v,s}(\pi_k^1(s), \pi_k^2(s)) + \frac{16\beta_k A_{\max}^2}{\tau} \|v^1 + v^2\|_\infty^2$$

$$+ \frac{8\beta_k}{\tau} \left( \frac{1}{\ell_\tau} + \frac{A_{\max}}{\tau(1-\gamma)} \right)^2 \sum_{i=1,2} \sum_{s \in \mathcal{S}} \|q_k^i(s) - \mathcal{T}^i(v^i)(s)\pi_k^2(s)\|_2^2 + 2L_\tau\beta_k^2$$

$$= \left(1 - \frac{3\beta_k}{4}\right) \mathcal{L}_\pi(k) + \frac{16\beta_k A_{\max}^2}{\tau} \|v^1 + v^2\|_\infty^2 + \frac{8\beta_k}{\tau} \left( \frac{1}{\ell_\tau} + \frac{A_{\max}}{\tau(1-\gamma)} \right)^2 \mathcal{L}_q(k) + 2L_\tau\beta_k^2.$$

Since the RHS of the previous inequality does not depend on $s$, we have

$$\mathcal{L}_\pi(k+1) \leq \left(1 - \frac{3\beta_k}{4}\right) \mathcal{L}_\pi(k) + \frac{16\beta_k A_{\max}^2}{\tau} \|v^1 + v^2\|_\infty^2$$

$$+ \frac{8\beta_k}{\tau} \left( \frac{1}{\ell_\tau} + \frac{A_{\max}}{\tau(1-\gamma)} \right)^2 \mathcal{L}_q(k) + 2L_\tau\beta_k^2$$

$$\leq \left(1 - \frac{3\beta_k}{4}\right) \mathcal{L}_\pi(k) + \frac{16\beta_k A_{\max}^2}{\tau} \|v^1 + v^2\|_\infty^2$$

$$+ \frac{32 A_{\max}^2 \beta_k}{\tau^3 \ell_\tau^2 (1-\gamma)^2} \mathcal{L}_q(k) + 2L_\tau\beta_k^2,$$

where the last line follows from $\tau \leq 1/(1-\gamma)$. $\qquad\square$

### D.5.2 Analysis of the $q$-Functions

In this section, we consider $q_k^i$ generated by Algorithm 3. We begin by reformulating the update of the $q$-function as a stochastic approximation algorithm for estimating a time-varying target. For $i \in \{1,2\}$, fixing $v^i \in \mathbb{R}^{|\mathcal{S}|}$, let $F^i : \mathbb{R}^{|\mathcal{S}||\mathcal{A}^i|} \times \mathcal{S} \times \mathcal{A}^i \times \mathcal{A}^{-i} \times \mathcal{S} \mapsto \mathbb{R}^{|\mathcal{S}||\mathcal{A}^i|}$ be defined as

$$[F^i(q^i, s_0, a_0^i, a_0^{-i}, s_1)](s, a^i) = \mathbb{1}_{\{(s,a^i)=(s_0, a_0^i)\}} \left( R_i(s_0, a_0^i, a_0^{-i}) + \gamma v^i(s_1) - q^i(s_0, a_0^i) \right)$$

for all $(q^i, s_0, a_0^i, a_0^{-i}, s_1)$ and $(s, a^i)$. Then Algorithm 3 Line 5 can be compactly written as

$$q_{k+1}^i = q_k^i + \alpha_k F^i(q_k^i, S_k, A_k^i, A_k^{-i}, S_{k+1}). \tag{41}$$

Denote the stationary distribution of the Markov chain $\{S_k\}$ induced by the joint policy $\pi_k = (\pi_k^1, \pi_k^2)$ by $\mu_k \in \Delta(\mathcal{S})$, the existence and uniqueness of which is guaranteed by Lemma D.1 and Lemma B.1 (1). Let $\bar{F}_k^i : \mathbb{R}^{|\mathcal{S}||\mathcal{A}^i|} \mapsto \mathbb{R}^{|\mathcal{S}||\mathcal{A}^i|}$ be defined as

$$\bar{F}_k^i(q^i) = \mathbb{E}_{S_0 \sim \mu_k(\cdot), A_0^i \sim \pi_k^i(\cdot|S_0), A_0^{-i} \sim \pi_k^{-i}(\cdot|S_0), S_1 \sim p(\cdot|S_0, A_0^i, A_0^{-i})} \left[ F^i(q^i, S_0, A_0^i, A_0^{-i}, S_1) \right]$$

for all $q^i \in \mathbb{R}^{|\mathcal{S}||\mathcal{A}^i|}$. Then, Eq. (41) can be viewed as a stochastic approximation algorithm for solving the (time-varying) equation $\bar{F}_k^i(q^i) = 0$ with time-inhomogeneous Markovian noise $\{(S_k, A_k^i, A_k^{-i}, S_{k+1})\}_{k \geq 0}$. We next establish the properties of the operators $F^i(\cdot)$ and $\bar{F}_k^i(\cdot)$ in the following lemma.

**Lemma D.9.** *The following properties hold for $i \in \{1, 2\}$.*

(1) *It holds that $\|F^i(q_1^i, s_0, a_0^i, a_0^{-i}, s_1) - F^i(q_2^i, s_0, a_0^i, a_0^{-i}, s_1)\|_2 \le \|q_1^i - q_2^i\|_2$ for any $(q_1^i, q_2^i)$ and $(s_0, a_0^i, a_0^{-i}, s_1)$.*

(2) *It holds that $\|F^i(0, s_0, a_0^i, a_0^{-i}, s_1)\|_2 \le \frac{1}{1-\gamma}$ for all $(s_0, a_0^i, a_0^{-i}, s_1)$.*

(3) *$\bar{F}_k^i(q^i) = 0$ has a unique solution $\bar{q}_k^i$, which is given as $\bar{q}_k^i(s) = \mathcal{T}^i(v^i)(s)\pi_k^{-i}(s)$ for all $s$.*

(4) *It holds that $\langle \bar{F}_k^i(q_1^i) - \bar{F}_k^i(q_2^i), q_1^i - q_2^i \rangle \le -c_\tau \|q_1^i - q_2^i\|_2^2$ for all $(q_1^i, q_2^i)$, where $c_\tau = \mu_{\min}\ell_\tau$. See Lemma B.1 for the definition of $\mu_{\min}$.*

*Proof of Lemma D.9.* (1) For any $(q_1^i, q_2^i)$ and $(s_0, a_0^i, a_0^{-i}, s_1)$, we have

$$
\|F^i(q_1^i, s_0, a_0^i, a_0^{-i}, s_1) - F^i(q_2^i, s_0, a_0^i, a_0^{-i}, s_1)\|_2^2
$$
$$
= \sum_{(s,a^i)} \left([F^i(q_1^i, s_0, a_0^i, a_0^{-i}, s_1)](s, a^i) - [F^i(q_2^i, s_0, a_0^i, a_0^{-i}, s_1)](s, a^i)\right)^2
$$
$$
= \left(q_1^i(s_0, a_0^i) - q_2^i(s_0, a_0^i)\right)^2
$$
$$
\le \|q_1^i - q_2^i\|_2^2.
$$

(2) For any $(s_0, a_0^i, a_0^{-i}, s_1)$, we have

$$
\|F^i(0, s_0, a_0^i, a_0^{-i}, s_1)\|_2^2 = \sum_{(s,a^i)} \left([F^i(0, s_0, a_0^i, a_0^{-i}, s_1)](s, a^i)\right)^2
$$
$$
= \left(R_i(s_0, a_0^i, a_0^{-i}) + \gamma v^i(s_1)\right)^2
$$
$$
\le \frac{1}{(1-\gamma)^2},
$$

where the last line follows from $\|v^i\|_\infty \le 1/(1-\gamma)$ and $|R_i(s_0, a_0^i, a_0^{-i})| \le 1$.

(3) We first write down the explicitly the operator $\bar{F}_k^i(\cdot)$. Using the definition of $\mathcal{T}^i(\cdot)$, we have

$$
\bar{F}_k^i(q^i)(s) = \mu_k(s)\mathrm{diag}(\pi_k^i(s)) \left(\mathcal{T}^i(v^i)(s)\pi_k^{-i}(s) - q^i(s)\right), \quad \forall\, s \in \mathcal{S},
$$

Since $\mu_k(s) \ge \mu_{\min} > 0$ (cf. Lemma B.1 (4)) and $\mathrm{diag}(\pi_k^i(s))$ has strictly positive diagonal entries (cf. Lemma D.1) for all $s \in \mathcal{S}$ and $k \ge 0$, the equation $\bar{F}_k^i(q^i) = 0$ has a unique solution $\bar{q}_k^i \in \mathbb{R}^{|\mathcal{S}||\mathcal{A}^i|}$, which is given as

$$
\bar{q}_k^i(s) = \mathcal{T}^i(v^i)(s)\pi_k^{-i}(s), \quad \forall\, s \in \mathcal{S}.
$$

(4) Using the expression of $\bar{F}_k^i(\cdot)$, we have for any $q_1^i, q_2^i \in \mathbb{R}^{|\mathcal{S}||\mathcal{A}^i|}$ that

$$
(q_1^i - q_2^i)^\top (\bar{F}_k^i(q_1^i) - \bar{F}_k^i(q_2^i)) = -\sum_{s,a^i} \mu_k(s)\pi_k^i(a^i|s)(q_1^i(s, a^i) - q_2^i(s, a^i))^2
$$
$$
\le -\min_{s,a^i} \mu_k(s)\pi_k^i(a^i|s)\|q_1^i - q_2^i\|_2^2
$$
$$
\le -\mu_{\min}\ell_\tau\|q_1^i - q_2^i\|_2^2 \qquad \text{(Lemma B.1 and Lemma D.1)}
$$
$$
= -c_\tau\|q_1^i - q_2^i\|_2^2.
$$

The proof is complete.

$\square$

Next, we establish a negative drift inequality for $\mathcal{L}_q(k)$. Using $\|\cdot\|_2^2$ as a Lyapunov function, we have by Eq. (41) that

$$
\mathbb{E}[\|q_{k+1}^i - \bar{q}_{k+1}^i\|_2^2] = \mathbb{E}[\|q_{k+1}^i - q_k^i + q_k^i - \bar{q}_k^i + \bar{q}_k^i - \bar{q}_{k+1}^i\|_2^2]
$$

$$
\begin{aligned}
&= \mathbb{E}[\|q_k^i - \bar{q}_k^i\|_2^2] + \mathbb{E}[\|q_{k+1}^i - q_k^i\|_2^2] + \mathbb{E}[\|\bar{q}_k^i - \bar{q}_{k+1}^i\|_2^2] \\
&\quad + 2\alpha_k \mathbb{E}[(q_k^i - \bar{q}_k^i)^\top \bar{F}_k^i(q_k^i)] \\
&\quad + 2\alpha_k \mathbb{E}[(F^i(q_k^i, S_k, A_k^i, A_k^{-i}, S_{k+1}) - \bar{F}_k^i(q_k^i))^\top (q_k^i - \bar{q}_k^i)] \\
&\quad + 2\mathbb{E}[(\bar{q}_k^i - \bar{q}_{k+1}^i)^\top (q_{k+1}^i - q_k^i)] \\
&\quad + 2\mathbb{E}[(q_k^i - \bar{q}_k^i)^\top (\bar{q}_k^i - \bar{q}_{k+1}^i)] \\
&\leq (1 - 2\alpha_k c_\tau)\mathbb{E}[\|q_k^i - \bar{q}_k^i\|_2^2] + \mathbb{E}[\|q_{k+1}^i - q_k^i\|_2^2] + \mathbb{E}[\|\bar{q}_k^i - \bar{q}_{k+1}^i\|_2^2] \\
&\quad + 2\mathbb{E}[(\bar{q}_k^i - \bar{q}_{k+1}^i)^\top (q_{k+1}^i - q_k^i)] \\
&\quad + 2\mathbb{E}[(q_k^i - \bar{q}_k^i)^\top (\bar{q}_k^i - \bar{q}_{k+1}^i)] \\
&\quad + 2\alpha_k \mathbb{E}[(F^i(q_k^i, S_k, A_k^i, A_k^{-i}, S_{k+1}) - \bar{F}_k^i(q_k^i))^\top (q_k^i - \bar{q}_k^i)] \qquad (42)
\end{aligned}
$$

where the last line follows from Lemma D.9 (4). The terms $\mathbb{E}[\|q_{k+1}^i - q_k^i\|_2^2]$, $\mathbb{E}[\|\bar{q}_k^i - \bar{q}_{k+1}^i\|_2^2]$, $\mathbb{E}[(\bar{q}_k^i - \bar{q}_{k+1}^i)^\top (q_{k+1}^i - q_k^i)]$, $\mathbb{E}[(q_k^i - \bar{q}_k^i)^\top (\bar{q}_k^i - \bar{q}_{k+1}^i)]$ on the RHS of Eq. (42) are bounded in the following lemma.

**Lemma D.10.** *The following inequalities hold for all $k \geq 0$.*

*(1)* $\mathbb{E}[\|q_{k+1}^i - q_k^i\|_2^2] \leq \frac{4|\mathcal{S}|A_{\max}\alpha_k^2}{(1-\gamma)^2}$.

*(2)* $\mathbb{E}[\|\bar{q}_k^i - \bar{q}_{k+1}^i\|_2^2] \leq \frac{4|\mathcal{S}|A_{\max}\beta_k^2}{(1-\gamma)^2}$.

*(3)* $\mathbb{E}[\langle q_{k+1}^i - q_k^i, \bar{q}_k^i - \bar{q}_{k+1}^i\rangle] \leq \frac{4|\mathcal{S}|A_{\max}\alpha_k\beta_k}{(1-\gamma)^2}$.

*(4)* $\mathbb{E}[\langle q_k^i - \bar{q}_k^i, \bar{q}_k^i - \bar{q}_{k+1}^i\rangle] \leq \frac{17|\mathcal{S}|A_{\max}^2\beta_k}{\tau(1-\gamma)^2}\mathbb{E}[\|q_k^i - \bar{q}_k^i\|_2^2] + \frac{\beta_k}{16}\mathbb{E}[\mathcal{L}_\pi(k)]$.

*Proof of Lemma D.10.* (1) For any $k \geq 0$, using Eq. (41) and Lemma D.9 (1), we have

$$
\begin{aligned}
\|q_{k+1}^i - q_k^i\|_2^2 &= \alpha_k^2 \|F^i(q_k^i, S_k, A_k^i, A_k^{-i}, S_{k+1})\|_2^2 \\
&= \alpha_k^2 \|F^i(q_k^i, S_k, A_k^i, A_k^{-i}, S_{k+1}) - F^i(0, S_k, A_k^i, A_k^{-i}, S_{k+1}) \\
&\quad + F^i(0, S_k, A_k^i, A_k^{-i}, S_{k+1})\|_2^2 \\
&\leq \alpha_k^2 \left(\|q_k^i\|_2 + \frac{1}{1-\gamma}\right)^2 \\
&\leq \alpha_k^2 \left(\frac{\sqrt{|\mathcal{S}|A_{\max}}}{1-\gamma} + \frac{1}{1-\gamma}\right)^2 \qquad (\|q_k^i\|_\infty \leq \tfrac{1}{1-\gamma} \text{ by Lemma D.1}) \\
&\leq \frac{4|\mathcal{S}|A_{\max}\alpha_k^2}{(1-\gamma)^2}.
\end{aligned}
$$

The result follows by taking expectation on both sides of the previous inequality.

(2) For any $k \geq 0$, using the definition of $\bar{q}_k$ in Appendix D.1, we have by Lemma D.9 that

$$
\begin{aligned}
\|\bar{q}_k^i - \bar{q}_{k+1}^i\|_2^2 &= \sum_s \|\mathcal{T}^i(v^i)(s)(\pi_{k+1}^{-i}(s) - \pi_k^{-i}(s))\|_2^2 \\
&= \beta_k^2 \sum_s \|\mathcal{T}^i(v^i)(s)(\sigma_\tau(q_k^{-i}(s)) - \pi_k^{-i}(s))\|_2^2 \\
&\leq \beta_k^2 \sum_s (\|\mathcal{T}^i(v^i)(s)\sigma_\tau(q_k^{-i}(s))\|_2 + \|\mathcal{T}^i(v^i)(s)\pi_k^{-i}(s)\|_2)^2 \\
&\leq \frac{4|\mathcal{S}|A_{\max}\beta_k^2}{(1-\gamma)^2}.
\end{aligned}
$$

The result follows by taking expectation on both sides of the previous inequality.

(3) For any $k \geq 0$, we have

$$\langle q^i_{k+1} - q^i_k, \bar{q}^i_k - \bar{q}^i_{k+1} \rangle \leq \|q^i_{k+1} - q^i_k\|_2 \|\bar{q}^i_k - \bar{q}^i_{k+1}\|_2 \leq \frac{4|\mathcal{S}|A_{\max}\alpha_k\beta_k}{(1-\gamma)^2},$$

where the last inequality follows from Part (1) and Part (2) of this lemma. The result follows by taking expectation on both sides of the previous inequality.

(4) For any $k \geq 0$, we have

$$\langle q^i_k - \bar{q}^i_k, \bar{q}^i_k - \bar{q}^i_{k+1} \rangle$$
$$= \beta_k \sum_s \langle q^i_k(s) - \bar{q}^i_k(s), \mathcal{T}^i(v^i)(s)(\sigma_\tau(q^{-i}_k(s)) - \pi^{-i}_k(s)) \rangle$$
$$\leq \frac{c_1\beta_k\|q^i_k - \bar{q}^i_k\|_2^2}{2} + \frac{\beta_k \sum_s \|\mathcal{T}^i(v^i)(s)(\sigma_\tau(q^{-i}_k(s)) - \pi^{-i}_k(s))\|_2^2}{2c_1}, \qquad (43)$$

where $c_1 > 0$ is an arbitrary positive real number. We next bound the second term on the RHS of the previous inequality. For any $s \in \mathcal{S}$, we have

$$\|\mathcal{T}^i(v^i)(s)(\sigma_\tau(q^{-i}_k(s)) - \pi^{-i}_k(s))\|_2$$
$$= \|\mathcal{T}^i(v^i)(s)(\sigma_\tau(q^{-i}_k(s)) - \sigma_\tau(\bar{q}^{-i}_k(s)) + \sigma_\tau(\mathcal{T}^{-i}(v^{-i})(s)\pi^i_k(s)) - \pi^{-i}_k(s))\|_2$$
$$\leq \underbrace{\|\mathcal{T}^i(v^i)(s)(\sigma_\tau(q^{-i}_k(s)) - \sigma_\tau(\bar{q}^{-i}_k(s)))\|_2}_{B_1}$$
$$+ \underbrace{\|\mathcal{T}^i(v^i)(s)(\sigma_\tau(\mathcal{T}^{-i}(v^{-i})(s)\pi^i_k(s)) - \pi^{-i}_k(s))\|_2}_{B_2}.$$

Since the softmax operator $\sigma_\tau(\cdot)$ is $\frac{1}{\tau}$ – Lipschitz continuous with respect to $\|\cdot\|_2$ [96, Proposition 4], we have

$$B_1 \leq \|\mathcal{T}^i(v^i)(s)\|_2 \|\sigma_\tau(q^{-i}_k(s)) - \sigma_\tau(\bar{q}^{-i}_k(s))\|_2$$
$$\leq \frac{A_{\max}}{\tau(1-\gamma)} \|q^{-i}_k(s) - \bar{q}^{-i}_k(s)\|_2.$$

We next analyze the term $B_2$. Using Lemma D.7 (1) and the quadratic growth property of strongly convex functions, we have

$$B_2 = \|\mathcal{T}^i(v^i)(s)(\sigma_\tau(\mathcal{T}^{-i}(v^{-i})(s)\pi^i_k(s)) - \pi^{-i}_k(s))\|_2$$
$$\leq \|\mathcal{T}^i(v^i)(s)\|_2 \|\sigma_\tau(\mathcal{T}^{-i}(v^{-i})(s)\pi^i_k(s)) - \pi^{-i}_k(s)\|_2$$
$$\leq \frac{\sqrt{2}A_{\max}}{\sqrt{\tau}(1-\gamma)} V_{v,s}(\pi^1_k(s), \pi^2_k(s))^{1/2}.$$

Combine the upper bounds we obtained for the terms $B_1$ and $B_2$, we obtain

$$\sum_s \|\mathcal{T}^i(v^i)(s)(\sigma_\tau(q^{-i}_k(s)) - \pi^{-i}_k(s))\|_2^2$$
$$\leq \sum_s (B_1 + B_2)^2$$
$$\leq 2\sum_s (B_1^2 + B_2^2)$$
$$\leq 2\sum_s \left( \frac{A_{\max}^2}{\tau^2(1-\gamma)^2} \|q^{-i}_k(s)) - \bar{q}^{-i}_k(s)\|_2^2 + \frac{2A_{\max}^2}{\tau(1-\gamma)^2} V_{v,s}(\pi^1_k(s), \pi^2_k(s)) \right)$$
$$\leq \frac{2A_{\max}^2}{\tau^2(1-\gamma)^2} \|q^{-i}_k - \bar{q}^{-i}_k\|_2^2 + \frac{4|\mathcal{S}|A_{\max}^2}{\tau(1-\gamma)^2} \mathcal{L}_\pi(k).$$

Coming back to Eq. (43), using the previous inequality, we have

$$\langle q^i_k - \bar{q}^i_k, \bar{q}^i_k - \bar{q}^i_{k+1} \rangle$$

$$\leq \frac{c_1\beta_k\|q_k^i - \bar{q}_k^i\|_2^2}{2} + \frac{\beta_k \sum_s \|\mathcal{T}^i(v^i)(s)(\sigma_\tau(q_k^{-i}(s)) - \pi_k^{-i}(s))\|_2^2}{2c_1}$$

$$\leq \frac{c_1\beta_k\|q_k^i - \bar{q}_k^i\|_2^2}{2} + \frac{A_{\max}^2\beta_k}{c_1\tau^2(1-\gamma)^2}\|q_k^{-i} - \bar{q}_k^{-i}\|_2^2 + \frac{2|\mathcal{S}|A_{\max}^2\beta_k}{c_1\tau(1-\gamma)^2}\mathcal{L}_\pi(k).$$

Choosing $c_1 = \frac{32|\mathcal{S}|A_{\max}^2}{\tau(1-\gamma)^2}$ in the previous inequality and then taking expectation on both sides, we obtain

$$\mathbb{E}[\langle q_k^i - \bar{q}_k^i, \bar{q}_k^i - \bar{q}_{k+1}^i\rangle] \leq \frac{17|\mathcal{S}|A_{\max}^2\beta_k}{\tau(1-\gamma)^2}\mathbb{E}[\|q_k^i - \bar{q}_k^i\|_2^2] + \frac{\beta_k}{16}\mathbb{E}[\mathcal{L}_\pi(k)].$$

The proof is complete.

$\square$

We next consider the last term on the RHS of Eq. (42), which involves the difference between the operator $F^i(q_k^i, S_k, A_k^i, A_k^{-i}, S_{k+1})$ and its expected version $\bar{F}_k^i(q_k^i)$, and hence can be viewed as the stochastic error due to sampling. The fact that the Markov chain $\{(S_k, A_k^i, A_k^{-i}, S_{k+1})\}$ is time-inhomogeneous presents a challenge in our analysis. To overcome this challenge, observe that: (1) the policy (hence the transition probability matrix of the induced Markov chain) is changing slowly compared to the $q$-function; see Algorithm 3 Line 3, and (2) the stationary distribution as a function of the policy is Lipschitz (cf. Lemma B.1 (3)). These two observations together enable us to develop a refined conditioning argument to handle the time-inhomogeneous Markovian noise. The result is presented in following. Similar ideas were previous used in [23, 24, 65, 41, 76] for finite-sample analysis of single-agent RL algorithms. Recall that we use $\alpha_{k_1,k_2} = \sum_{k=k_1}^{k_2} \alpha_k$ to simplify the notation.

**Lemma D.11** (Proof in Appendix D.7.2). *The following inequality holds for all $k \geq z_k$ :*

$$\mathbb{E}[(F^i(q_k^i, S_k, A_k^i, A_k^{-i}, S_{k+1}) - \bar{F}_k^i(q_k^i))^\top (q_k^i - \bar{q}_k^i)] \leq \frac{17z_k\alpha_{k-z_k,k-1}}{(1-\gamma)^2},$$

*where $z_k$ is the mixing time of the Markov chain $\{S_n\}_{n\geq 0}$ induced by the joint policy $\pi_k = (\pi_k^1, \pi_k^2)$ with accuracy $\beta_k$; see Eq. (11).*

When using constant stepsize, we have $z_k\alpha_{k-z_k,k-1} = z_\beta^2\alpha = \mathcal{O}(\alpha\log^2(1/\beta))$. Since the two stepsizes $\alpha$ and $\beta$ differ only by a multiplicative constant $c_{\alpha,\beta}$, we have $\lim_{\alpha\to 0} z_\beta^2\alpha = 0$. Similarly, we also have $\lim_{k\to\infty} z_k\alpha_{k-z_k,k-1} = 0$ when using diminishing stepsizes.

Using the upper bounds we obtained for all the terms on the RHS of Eq. (42), we have the one-step Lyapunov drift inequality for $q_k^i$. Following the same line of analysis, we also obtain the one-step inequality for $q_k^{-i}$. Adding up the two Lyapunov drift inequalities, we arrive at the following lemma.

**Lemma D.12.** *The following inequality holds for all $k \geq z_k$ and $i \in \{1,2\}$:*

$$\mathbb{E}[\mathcal{L}_q(k+1)] \leq (1 - \alpha_k c_\tau)\mathbb{E}[\mathcal{L}_q(k)] + \frac{\beta_k}{4}\mathbb{E}[\mathcal{L}_\pi(k)] + \frac{100|\mathcal{S}|A_{\max}}{(1-\gamma)^2}z_k\alpha_k\alpha_{k-z_k,k-1}.$$

*Proof of Lemma D.12.* For $i \in \{1,2\}$, we have from Eq. (42), Lemma D.10, and Lemma D.11 that

$$\mathbb{E}[\|q_{k+1}^i - \bar{q}_{k+1}^i\|_2^2] \leq (1 - 2\alpha_k c_\tau)\mathbb{E}[\|q_k^i - \bar{q}_k^i\|_2^2] + \frac{4|\mathcal{S}|A_{\max}}{(1-\gamma)^2}(\alpha_k^2 + 2\alpha_k\beta_k + \beta_k^2)$$

$$+ \frac{34|\mathcal{S}|A_{\max}^2\beta_k}{\tau(1-\gamma)^2}\mathbb{E}[\|q_k^i - \bar{q}_k^i\|_2^2] + \frac{\beta_k}{8}\mathbb{E}[\mathcal{L}_\pi(k)] + \frac{34z_k\alpha_k\alpha_{k-z_k,k-1}}{(1-\gamma)^2}$$

$$\leq \left(1 - 2\alpha_k c_\tau + \frac{34|\mathcal{S}|A_{\max}^2\beta_k}{\tau(1-\gamma)^2}\right)\mathbb{E}[\|q_k^i - \bar{q}_k^i\|_2^2] + \frac{\beta_k}{8}\mathbb{E}[\mathcal{L}_\pi(k)]$$

$$+ \frac{50|\mathcal{S}|A_{\max}}{(1-\gamma)^2}z_k\alpha_k\alpha_{k-z_k,k-1},$$

where the second inequality follows from $\beta_k = c_{\alpha,\beta}\alpha_k$ with $c_{\alpha,\beta} \leq 1$. Since

$$c_{\alpha,\beta} \leq \frac{c_\tau \tau (1-\gamma)^2}{34|\mathcal{S}|A_{\max}^2}, \qquad \text{(Condition 3.1)}$$

we have

$$\mathbb{E}[\|q_{k+1}^i - \bar{q}_{k+1}^i\|_2^2] \leq (1 - \alpha_k c_\tau)\,\mathbb{E}[\|q_k^i - \bar{q}_k^i\|_2^2] + \frac{\beta_k}{8}\mathbb{E}[\mathcal{L}_\pi(k)] + \frac{50|\mathcal{S}|A_{\max}}{(1-\gamma)^2}z_k \alpha_k \alpha_{k-z_k,k-1}.$$

Summing up the previous inequality for $i = 1, 2$, we have

$$\mathbb{E}[\mathcal{L}_q(k+1)] \leq (1 - \alpha_k c_\tau)\,\mathbb{E}[\mathcal{L}_q(k)] + \frac{\beta_k}{4}\mathbb{E}[\mathcal{L}_\pi(k)] + \frac{100|\mathcal{S}|A_{\max}}{(1-\gamma)^2}z_k \alpha_k \alpha_{k-z_k,k-1}.$$

$\square$

## D.6 Solving Coupled Lyapunov Drift Inequalities

We first restate the Lyapunov drift inequalities from previous sections. Recall our notation $\mathcal{L}_q(t,k) = \sum_{i=1,2}\|q_{t,k}^i - \bar{q}_{t,k}^i\|_2^2$, $\mathcal{L}_\pi(t,k) = \max_{s\in\mathcal{S}} V_{v_t,s}(\pi_{t,k}^1(s), \pi_{t,k}^2(s))$, $\mathcal{L}_{\text{sum}}(t) = \|v_t^1 + v_t^2\|_\infty$, and $\mathcal{L}_v(t) = \sum_{i=1,2}\|v_t^i - v_*^i\|_\infty$. Let $\mathcal{F}_t$ be the history of Algorithm 2 right before the $t$-th outer-loop iteration. Note that $v_t^1$ and $v_t^2$ are both measurable with respect to $\mathcal{F}_t$. In what follows, for ease of presentation, we write $\mathbb{E}_t[\,\cdot\,]$ for $\mathbb{E}[\,\cdot\mid\mathcal{F}_t]$.

- **Lemma D.5:** It holds for all $t \geq 0$ that

$$\mathcal{L}_v(t+1) \leq \gamma\mathcal{L}_v(t) + 4\mathcal{L}_{\text{sum}}(t) + 2\mathcal{L}_q^{1/2}(t,K) + 4\mathcal{L}_\pi(t,K) + 6\tau\log(A_{\max}). \qquad (44)$$

- **Lemma D.6:** It holds for all $t \geq 0$ that

$$\mathcal{L}_{\text{sum}}(t+1) \leq \gamma\mathcal{L}_{\text{sum}}(t) + 2\mathcal{L}_q^{1/2}(t,K). \qquad (45)$$

- **Lemma D.8:** It holds for all $t, k \geq 0$ that

$$\mathbb{E}_t[\mathcal{L}_\pi(t,k+1)] \leq \left(1 - \frac{3\beta_k}{4}\right)\mathbb{E}_t[\mathcal{L}_\pi(t,k)] + \frac{16A_{\max}^2\beta_k}{\tau}\mathcal{L}_{\text{sum}}(t)^2$$
$$+ \frac{32A_{\max}^2\beta_k}{\tau^3\ell_\tau^2(1-\gamma)^2}\mathcal{L}_q(k) + 2L_\tau\beta_k^2. \qquad (46)$$

- **Lemma D.12:** It holds for all $t \geq 0$ and $k \geq z_k$ that

$$\mathbb{E}_t[\mathcal{L}_q(t,k+1)] \leq (1 - \alpha_k c_\tau)\,\mathbb{E}_t[\mathcal{L}_q(t,k)] + \frac{\beta_k}{4}\mathbb{E}_t[\mathcal{L}_\pi(t,k)] + \frac{100|\mathcal{S}|A_{\max}}{(1-\gamma)^2}z_k \alpha_k \alpha_{k-z_k,k-1}. \qquad (47)$$

Adding up Eqs. (46) and (47), we have by $c_{\alpha,\beta} \leq \min\left(\frac{1}{L_\tau^{1/2}}, \frac{c_\tau \tau^3 \ell_\tau^2 (1-\gamma)^2}{128A_{\max}^2}, c_\tau\right)$ (cf. Condition 3.1) that

$$\mathbb{E}_t[\mathcal{L}_\pi(t,k+1) + \mathcal{L}_q(t,k+1)] \leq \left(1 - \frac{\beta_k}{2}\right)\mathbb{E}_t[\mathcal{L}_\pi(t,k) + \mathcal{L}_q(t,k)]$$
$$\frac{16A_{\max}^2\beta_k}{\tau}\mathcal{L}_{\text{sum}}(t)^2 + \frac{102|\mathcal{S}|A_{\max}}{(1-\gamma)^2}z_k \alpha_k \alpha_{k-z_k,k-1}. \qquad (48)$$

### D.6.1 Constant Stepsize

When using constant stepsizes, i.e., $\alpha_k \equiv \alpha$, $\beta_k \equiv \beta = c_{\alpha,\beta}\alpha$, repeatedly using Eq. (48) from $z_\beta$ to $k$, we have

$$\mathbb{E}_t[\mathcal{L}_\pi(t,k) + \mathcal{L}_q(t,k)] \leq \left(1 - \frac{\beta}{2}\right)^{k-z_\beta}(\mathcal{L}_\pi(t,0) + \mathcal{L}_q(t,0))$$

$$+ \frac{16A_{\max}^2}{\tau}\mathcal{L}_{\text{sum}}(t)^2 + \frac{204|\mathcal{S}|A_{\max}}{(1-\gamma)^2 c_{\alpha,\beta}}z_\beta^2\alpha. \tag{49}$$

We next bound $\mathcal{L}_\pi(t,0) + \mathcal{L}_q(t,0)$. For $i \in \{1,2\}$, we have

$$\begin{aligned}
\mathcal{L}_\pi(t,0) &= \max_s V_{v_t,s}(\pi_{t,0}^1(s), \pi_{t,0}^2(s)) \\
&= \max_s \sum_{i=1,2}\max_{\mu^i}\{(\mu^i - \pi_{t,0}^i(s))^\top \mathcal{T}^i(v_t^i)(s)\pi_{t,0}^{-i}(s) + \tau\nu(\mu^i) - \tau\nu(\pi_{t,0}^i(s))\} \\
&\leq 2\sum_{i=1,2}\max_{s,a^i,a^{-i}}|\mathcal{T}^i(v_t^i)(s,a^i,a^{-i})| + 2\tau\log(A_{\max}) \\
&\leq \frac{4}{(1-\gamma)} + 2\tau\log(A_{\max}),
\end{aligned}$$

and

$$\mathcal{L}_q(t,0) = \sum_{i=1,2}\|q_{t,0}^i - \bar{q}_{t,0}^i\|_2^2 \leq \frac{8|\mathcal{S}|A_{\max}}{(1-\gamma)^2}. \tag{Lemma D.1}$$

It follows that

$$\mathcal{L}_\pi(t,0) + \mathcal{L}_q(t,0) \leq \frac{4}{(1-\gamma)} + 2\tau\log(A_{\max}) + \frac{8|\mathcal{S}|A_{\max}}{(1-\gamma)^2} = L_{\text{in}}.$$

Using the previous inequality in Eq. (49), we have

$$\mathbb{E}_t[\mathcal{L}_\pi(t,k) + \mathcal{L}_q(t,k)] \leq L_{\text{in}}\left(1 - \frac{\beta}{2}\right)^{k-z_\beta} + \frac{16A_{\max}^2}{\tau}\mathcal{L}_{\text{sum}}(t)^2 + \frac{204|\mathcal{S}|A_{\max}}{(1-\gamma)^2 c_{\alpha,\beta}}z_\beta^2\alpha, \tag{50}$$

which implies

$$\mathbb{E}_t[\mathcal{L}_\pi(t,k)] \leq L_{\text{in}}\left(1 - \frac{\beta}{2}\right)^{k-z_\beta} + \frac{16A_{\max}^2}{\tau}\mathcal{L}_{\text{sum}}(t)^2 + \frac{204|\mathcal{S}|A_{\max}}{(1-\gamma)^2 c_{\alpha,\beta}}z_\beta^2\alpha.$$

Substituting the previous inequality on $\mathbb{E}_t[\mathcal{L}_\pi(t,k)]$ into Eq. (47), we have

$$\begin{aligned}
\mathbb{E}_t[\mathcal{L}_q(t,k+1)] \leq &(1-\alpha c_\tau)\mathbb{E}_t[\mathcal{L}_q(t,k)] + \frac{151|\mathcal{S}|A_{\max}}{(1-\gamma)^2}z_\beta^2\alpha^2 + \frac{\beta L_{\text{in}}}{4}\left(1-\frac{\beta}{2}\right)^{k-z_\beta} \\
&+ \frac{4A_{\max}^2\beta}{\tau}\mathcal{L}_{\text{sum}}(t)^2.
\end{aligned}$$

Repeatedly using the previous inequality, since $c_{\alpha,\beta} \leq c_\tau$ (cf. Condition 3.1), we have

$$\begin{aligned}
\mathbb{E}_t[\mathcal{L}_q(t,k)] \leq &L_{\text{in}}(1-c_\tau\alpha)^{k-z_\beta} + \frac{\beta L_{\text{in}}(k-z_\beta)}{4}\left(1-\frac{\beta}{2}\right)^{k-z_\beta-1} \\
&+ \frac{4A_{\max}^2 c_{\alpha,\beta}}{c_\tau\tau}\mathcal{L}_{\text{sum}}(t)^2 + \frac{151|\mathcal{S}|A_{\max}}{(1-\gamma)^2 c_\tau}z_\beta^2\alpha,
\end{aligned}$$

which implies (by using Jensen's inequality) that

$$\begin{aligned}
\mathbb{E}_t[\mathcal{L}_q(t,k)^{1/2}] \leq &L_{\text{in}}^{1/2}(1-c_\tau\alpha)^{\frac{k-z_\beta}{2}} + \frac{\beta^{1/2}L_{\text{in}}^{1/2}(k-z_\beta)^{1/2}}{2}\left(1-\frac{\beta}{2}\right)^{\frac{k-z_\beta-1}{2}} \\
&+ \frac{2A_{\max}c_{\alpha,\beta}^{1/2}}{c_\tau^{1/2}\tau^{1/2}}\mathcal{L}_{\text{sum}}(t) + \frac{13|\mathcal{S}|^{1/2}A_{\max}^{1/2}}{(1-\gamma)c_\tau^{1/2}}z_\beta\alpha^{1/2}.
\end{aligned}$$

Substituting the previous bound on $\mathbb{E}_t[\mathcal{L}_q(t,k)^{1/2}]$ into Eq. (45) and then taking total expectation, we have

$$\mathbb{E}[\mathcal{L}_{\text{sum}}(t+1)] \leq \gamma\mathcal{L}_{\text{sum}}(t) + 2L_{\text{in}}^{1/2}(1-c_\tau\alpha)^{\frac{K-z_\beta}{2}} + \beta^{1/2}L_{\text{in}}^{1/2}(K-z_\beta)^{1/2}\left(1-\frac{\beta}{2}\right)^{\frac{K-z_\beta-1}{2}}$$

$$+ \frac{4A_{\max}c_{\alpha,\beta}^{1/2}}{c_\tau^{1/2}\tau^{1/2}}\mathcal{L}_{\text{sum}}(t) + \frac{26|\mathcal{S}|^{1/2}A_{\max}^{1/2}}{(1-\gamma)c_\tau^{1/2}}z_\beta\alpha^{1/2}$$

$$\leq \left(\frac{1+\gamma}{2}\right)\mathcal{L}_{\text{sum}}(t) + 2L_{\text{in}}^{1/2}\left(1 - c_\tau\alpha\right)^{\frac{K-z_\beta}{2}}$$

$$+ \beta^{1/2}L_{\text{in}}^{1/2}(K-z_\beta)^{1/2}\left(1-\frac{\beta}{2}\right)^{\frac{K-z_\beta-1}{2}} + \frac{26|\mathcal{S}|^{1/2}A_{\max}^{1/2}}{(1-\gamma)c_\tau^{1/2}}z_\beta\alpha^{1/2},$$

where the last line follows from $c_{\alpha,\beta} \leq \frac{c_\tau\tau(1-\gamma)^2}{64A_{\max}^2}$ (cf. Condition 3.1). Since $\|v_0^1 + v_0^2\|_\infty \leq \frac{2}{1-\gamma}$, repeatedly using the previous inequality, we have for all $k \geq 0$ that

$$\mathbb{E}[\mathcal{L}_{\text{sum}}(t)] \leq \frac{2}{1-\gamma}\left(\frac{1+\gamma}{2}\right)^t + \frac{4L_{\text{in}}^{1/2}\left(1 - c_\tau\alpha\right)^{\frac{K-z_\beta}{2}}}{1-\gamma}$$

$$+ \frac{2\beta^{1/2}L_{\text{in}}^{1/2}(K-z_\beta)^{1/2}}{1-\gamma}\left(1-\frac{\beta}{2}\right)^{\frac{K-z_\beta-1}{2}} + \frac{52|\mathcal{S}|^{1/2}A_{\max}^{1/2}}{(1-\gamma)^2c_\tau^{1/2}}z_\beta\alpha^{1/2}$$

$$\leq \frac{2}{1-\gamma}\left(\frac{1+\gamma}{2}\right)^t + \frac{6L_{\text{in}}^{1/2}(K-z_\beta)^{1/2}}{1-\gamma}\left(1-\frac{\beta}{2}\right)^{\frac{K-z_\beta-1}{2}}$$

$$+ \frac{52|\mathcal{S}|^{1/2}A_{\max}^{1/2}}{(1-\gamma)^2c_\tau^{1/2}}z_\beta\alpha^{1/2}. \tag{51}$$

Now we have obtained finite-sample bounds for $\mathcal{L}_q(t,k)$, $\mathcal{L}_\pi(t,k)$, and $\mathcal{L}_{\text{sum}}(t)$. The next step is to use them in Eq. (44) to obtain finite-sample bounds for $\mathcal{L}_v(t)$. Specifically, using Eq. (44), Eq. (50), and Eq. (51), we have

$$\mathbb{E}[\mathcal{L}_v(t+1)] \leq \gamma\mathbb{E}[\mathcal{L}_v(t)] + 4\mathbb{E}[\mathcal{L}_{\text{sum}}(t)] + 2\mathbb{E}[\mathcal{L}_q^{1/2}(t,K)] + 4\mathbb{E}[\mathcal{L}_\pi(t,K)] + 6\tau\log(A_{\max})$$

$$\leq \gamma\mathbb{E}[\mathcal{L}_v(t)] + 2L_{\text{in}}^{1/2}\left(1 - c_\tau\alpha\right)^{\frac{K-z_\beta}{2}}$$

$$+ \beta^{1/2}L_{\text{in}}^{1/2}(K-z_\beta)^{1/2}\left(1-\frac{\beta}{2}\right)^{\frac{K-z_\beta-1}{2}} + \frac{26|\mathcal{S}|^{1/2}A_{\max}^{1/2}}{(1-\gamma)c_\tau^{1/2}}z_\beta\alpha^{1/2}$$

$$+ 4L_{\text{in}}\left(1-\frac{\beta}{2}\right)^{K-z_\beta} + \frac{816|\mathcal{S}|A_{\max}}{(1-\gamma)^2c_{\alpha,\beta}}z_\beta^2\alpha + 6\tau\log(A_{\max})$$

$$+ \frac{266A_{\max}^2}{\tau(1-\gamma)^2}\left(\frac{1+\gamma}{2}\right)^t + \frac{798A_{\max}^2L_{\text{in}}^{1/2}(K-z_\beta)^{1/2}}{(1-\gamma)^2\tau}\left(1-\frac{\beta}{2}\right)^{\frac{K-z_\beta-1}{2}}$$

$$+ \frac{6916|\mathcal{S}|^{1/2}A_{\max}^{5/2}}{(1-\gamma)^3\tau c_\tau^{1/2}}z_\beta\alpha^{1/2}$$

$$\leq \gamma\mathbb{E}[\mathcal{L}_v(t)] + \frac{1223|\mathcal{S}|A_{\max}}{(1-\gamma)^2c_{\alpha,\beta}}z_\beta^2\alpha^{1/2} + 6\tau\log(A_{\max})$$

$$+ \frac{266A_{\max}^2}{\tau(1-\gamma)^2}\left(\frac{1+\gamma}{2}\right)^t + \frac{805A_{\max}^2L_{\text{in}}(K-z_\beta)^{1/2}}{(1-\gamma)^2\tau}\left(1-\frac{\beta}{2}\right)^{\frac{K-z_\beta-1}{2}}.$$

Repeatedly using the previous inequality from 0 to $T-1$ and then using $\mathcal{L}_v(0) \leq \frac{4}{1-\gamma}$, we have

$$\mathcal{L}_v(T) \leq \frac{270A_{\max}^2T}{\tau(1-\gamma)^2}\left(\frac{1+\gamma}{2}\right)^{T-1}$$

$$+ \frac{805A_{\max}^2L_{\text{in}}(K-z_\beta)^{1/2}}{\tau(1-\gamma)^3}\left(1-\frac{\beta}{2}\right)^{\frac{K-z_\beta-1}{2}}$$

$$+ \frac{1223|\mathcal{S}|A_{\max}}{(1-\gamma)^3c_{\alpha,\beta}}z_\beta^2\alpha^{1/2} + \frac{6\tau\log(A_{\max})}{1-\gamma}.$$

Our next step is to use the bounds we obtained for $\mathcal{L}_q(t,k)$, $\mathcal{L}_\pi(t,k)$, $\mathcal{L}_v(t)$, and $\mathcal{L}_{\text{sum}}(t)$ in Lemma D.4. For simplicity of presentation, we use $a \lesssim b$ to mean that there exists a *numerical* constant $c$ such that $a \leq cb$. Using the previous inequality, Eq. (50), and Eq. (51), we have

$$\mathbb{E}[\text{NG}(\pi_{T,K}^1, \pi_{T,K}^2)] \leq \frac{8}{1-\gamma}\mathcal{L}_{\text{sum}}(T) + \frac{4}{1-\gamma}\mathcal{L}_v(T) + \frac{4}{1-\gamma}\mathcal{L}_\pi(T,K) + \frac{8\tau\log(A_{\max})}{1-\gamma}$$

$$\lesssim \frac{A_{\max}^2 T}{\tau(1-\gamma)^3}\left(\frac{1+\gamma}{2}\right)^{T-1} + \frac{A_{\max}^2 L_{\text{in}}(K-z_\beta)^{1/2}}{\tau(1-\gamma)^4}\left(1-\frac{\beta}{2}\right)^{\frac{K-z_\beta-1}{2}}$$

$$+ \frac{|\mathcal{S}|A_{\max}}{(1-\gamma)^4 c_{\alpha,\beta}}z_\beta^2\alpha^{1/2} + \frac{\tau\log(A_{\max})}{(1-\gamma)^2}.$$

The proof of Theorem 3.1 (1) is complete.

### D.6.2 Diminishing Stepsizes

Consider using linearly diminishing stepsizes, i.e., $\alpha_k = \frac{\alpha}{k+h}$, $\beta_k = \frac{\beta}{k+h}$, and $\beta = c_{\alpha,\beta}\alpha$. Repeatedly using Eq. (48), we have for all $k \geq k_0 := \min\{k' \mid k' \geq z_{k'}\}$ that

$$\mathbb{E}_t[\mathcal{L}_\pi(t,k) + \mathcal{L}_q(t,k)] \leq L_{\text{in}}\underbrace{\prod_{m=k_0}^{k-1}\left(1 - \frac{\beta_m}{2}\right)}_{\hat{\mathcal{E}}_1} + \frac{204|\mathcal{S}|A_{\max}}{(1-\gamma)^2}\underbrace{\sum_{n=k_0}^{k-1}z_n^2\alpha_n^2\prod_{m=n+1}^{k-1}\left(1 - \frac{\beta_m}{2}\right)}_{\hat{\mathcal{E}}_2}$$

$$+ \frac{16A_{\max}^2}{\tau}\mathcal{L}_{\text{sum}}(t)^2\underbrace{\sum_{n=k_0}^{k-1}\beta_n\prod_{m=n+1}^{k-1}\left(1 - \frac{\beta_m}{2}\right)}_{\hat{\mathcal{E}}_3}.$$

Next, we evaluate the terms $\{\hat{\mathcal{E}}_j\}_{1 \leq j \leq 3}$. Terms like $\{\hat{\mathcal{E}}_j\}_{1 \leq j \leq 3}$ have been well studied in the existing literature [24, 44, 65]. Specifically, we have from [65, Appendix A.2.] and $\beta = 4$ that

$$\hat{\mathcal{E}}_1 \leq \frac{k_0 + h}{k + h}, \quad \hat{\mathcal{E}}_2 \leq \frac{64ez_k^2}{(k+h)c_{\alpha,\beta}^2}, \quad \text{and} \quad \hat{\mathcal{E}}_3 \leq 2.$$

It follows that

$$\mathbb{E}_t[\mathcal{L}_\pi(t,k) + \mathcal{L}_q(t,k)] \leq L_{\text{in}}\frac{k_0 + h}{k + h} + \frac{3264e|\mathcal{S}|A_{\max}}{(1-\gamma)^2 c_{\alpha,\beta}}z_k^2\alpha_k + \frac{32A_{\max}^2}{\tau}\mathcal{L}_{\text{sum}}(t)^2,$$

which implies

$$\mathbb{E}_t[\mathcal{L}_\pi(t,k)] \leq L_{\text{in}}\frac{k_0 + h}{k + h} + \frac{3264e|\mathcal{S}|A_{\max}}{(1-\gamma)^2 c_{\alpha,\beta}}z_k^2\alpha_k + \frac{32A_{\max}^2}{\tau}\mathcal{L}_{\text{sum}}(t)^2. \tag{52}$$

Using the previous inequality on $\mathbb{E}_t[\mathcal{L}_\pi(t,k)]$ in Eq. (47), we have

$$\mathbb{E}_t[\mathcal{L}_q(t,k+1)] \leq (1 - \alpha_k c_\tau)\mathbb{E}_t[\mathcal{L}_q(t,k)] + \frac{100|\mathcal{S}|A_{\max}}{(1-\gamma)^2}z_k\alpha_k\alpha_{k-z_k,k-1}$$

$$+ \frac{L_{\text{in}}c_{\alpha,\beta}\alpha_k^2}{4\alpha_{k_0}} + \frac{816e|\mathcal{S}|A_{\max}}{(1-\gamma)^2}z_k^2\alpha_k^2 + \frac{8A_{\max}^2\beta_k}{\tau}\mathcal{L}_{\text{sum}}(t)^2$$

$$\leq (1 - \alpha_k c_\tau)\mathbb{E}_t[\mathcal{L}_q(t,k)] + \frac{1017eL_{\text{in}}|\mathcal{S}|A_{\max}}{(1-\gamma)^2\alpha_{k_0}}z_k^2\alpha_k^2$$

$$+ \frac{8A_{\max}^2\beta_k}{\tau}\mathcal{L}_{\text{sum}}(t)^2.$$

Repeatedly using the previous inequality starting from $k_0$, since $\alpha c_\tau \geq 1$ (cf. Condition 3.1), we have

$$\mathbb{E}_t[\mathcal{L}_q(t,k)] \leq L_{\text{in}}\frac{k_0 + h}{k + h} + \frac{4068e^2 L_{\text{in}}|\mathcal{S}|A_{\max}}{(1-\gamma)^2 c_\tau\alpha_{k_0}}z_k^2\alpha_k + \frac{8A_{\max}^2 c_{\alpha,\beta}}{c_\tau\tau}\mathcal{L}_{\text{sum}}(t)^2,$$

which implies (by using Jensen's inequality) that

$$\mathbb{E}_t[\mathcal{L}_q(t,k)^{1/2}] \le L_{\text{in}}^{1/2}\left(\frac{k_0+h}{k+h}\right)^{1/2} + \frac{64eL_{\text{in}}^{1/2}|\mathcal{S}|^{1/2}A_{\max}^{1/2}}{(1-\gamma)c_\tau^{1/2}\alpha_{k_0}^{1/2}}z_k\alpha_k^{1/2} + \frac{3A_{\max}c_{\alpha,\beta}^{1/2}}{c_\tau^{1/2}\tau^{1/2}}\mathcal{L}_{\text{sum}}(t). \quad (53)$$

Taking total expectation on both sides of the previous inequality and then using the result in Eq. (45), we have

$$\mathbb{E}[\mathcal{L}_{\text{sum}}(t+1)] \le \gamma\mathbb{E}[\mathcal{L}_{\text{sum}}(t)] + 2L_{\text{in}}^{1/2}\left(\frac{k_0+h}{K+h}\right)^{1/2} + \frac{128e|\mathcal{S}|^{1/2}A_{\max}^{1/2}}{(1-\gamma)c_\tau^{1/2}\alpha_{k_0}^{1/2}}z_K\alpha_K^{1/2}$$

$$+ \frac{6A_{\max}c_{\alpha,\beta}^{1/2}}{c_\tau^{1/2}\tau^{1/2}}\mathcal{L}_{\text{sum}}(t)$$

$$\le \left(\frac{\gamma+1}{2}\right)\mathbb{E}[\mathcal{L}_{\text{sum}}(t)] + \frac{130eL_{\text{in}}^{1/2}|\mathcal{S}|^{1/2}A_{\max}^{1/2}}{(1-\gamma)c_\tau^{1/2}\alpha_{k_0}^{1/2}}z_K\alpha_K^{1/2},$$

where the last line follows from $c_{\alpha,\beta} \le \frac{c_\tau\tau(1-\gamma)^2}{144A_{\max}^2}$ (cf. Condition 3.1). Repeatedly using the previous inequality starting from 0, we have

$$\mathbb{E}[\mathcal{L}_{\text{sum}}(t)] \le \frac{2}{1-\gamma}\left(\frac{1+\gamma}{2}\right)^t + \frac{260eL_{\text{in}}^{1/2}|\mathcal{S}|^{1/2}A_{\max}^{1/2}}{(1-\gamma)^2c_\tau^{1/2}\alpha_{k_0}^{1/2}}z_K\alpha_K^{1/2}. \quad (54)$$

The next step is to bound $\mathcal{L}_v(t)$. Recall from Eq. (44) that

$$\mathbb{E}_t[\mathcal{L}_v(t+1)] \le \gamma\mathcal{L}_v(t) + 4\mathcal{L}_{\text{sum}}(t) + 2\mathbb{E}_t[\mathcal{L}_q^{1/2}(t,K)] + 4\mathbb{E}_t[\mathcal{L}_\pi(t,K)] + 6\tau\log(A_{\max}).$$

Using Eqs. (52), (53), and (54) in the previous inequality, we have

$$\mathbb{E}[\mathcal{L}_v(t+1)] \le \gamma\mathbb{E}[\mathcal{L}_v(t)] + 4\mathbb{E}[\mathcal{L}_{\text{sum}}(t)] + 2\mathbb{E}[\mathcal{L}_q^{1/2}(t,K)] + 4\mathbb{E}[\mathcal{L}_\pi(t,K)] + 6\tau\log(A_{\max})$$

$$\le \gamma\mathbb{E}[\mathcal{L}_v(t)] + \frac{130eL_{\text{in}}^{1/2}|\mathcal{S}|^{1/2}A_{\max}^{1/2}}{(1-\gamma)c_\tau^{1/2}\alpha_{k_0}^{1/2}}z_K\alpha_K^{1/2}$$

$$+ \frac{4L_{\text{in}}\alpha_K}{\alpha_{k_0}} + \frac{13056e|\mathcal{S}|A_{\max}}{(1-\gamma)^2c_{\alpha,\beta}}z_K^2\alpha_K + 6\tau\log(A_{\max})$$

$$+ \frac{522A_{\max}^2}{\tau(1-\gamma)^2}\left(\frac{1+\gamma}{2}\right)^t + \frac{67860eL_{\text{in}}^{1/2}|\mathcal{S}|^{1/2}A_{\max}^{5/2}}{(1-\gamma)^3\tau c_\tau^{1/2}\alpha_{k_0}^{1/2}}z_K\alpha_K^{1/2}$$

$$\le \gamma\mathbb{E}[\mathcal{L}_v(t)] + \frac{522A_{\max}^2}{\tau(1-\gamma)^2}\left(\frac{1+\gamma}{2}\right)^t + \frac{15056eL_{\text{in}}|\mathcal{S}|A_{\max}}{(1-\gamma)^2\alpha_{k_0}^{1/2}c_{\alpha,\beta}}z_K^2\alpha_K^{1/2}$$

$$+ 6\tau\log(A_{\max}).$$

Repeatedly using the previous inequality starting from 0 to $T-1$, we have

$$\mathcal{L}_v(T) \le \frac{526A_{\max}^2 T}{\tau(1-\gamma)^2}\left(\frac{1+\gamma}{2}\right)^{T-1} + \frac{15056eL_{\text{in}}|\mathcal{S}|A_{\max}}{(1-\gamma)^3\alpha_{k_0}^{1/2}c_{\alpha,\beta}}z_K^2\alpha_K^{1/2} + \frac{6\tau\log(A_{\max})}{1-\gamma}$$

Finally, using the previous inequality, Eq. (52), and Eq. (54) in Lemma D.4, we obtain

$$\mathbb{E}[\text{NG}(\pi_{T,K}^1, \pi_{T,K}^2)] \lesssim \frac{A_{\max}^2 T}{\tau(1-\gamma)^3}\left(\frac{1+\gamma}{2}\right)^{T-1} + \frac{L_{\text{in}}|\mathcal{S}|A_{\max}}{(1-\gamma)^4\alpha_{k_0}^{1/2}c_{\alpha,\beta}}z_K^2\alpha_K^{1/2} + \frac{\tau\log(A_{\max})}{(1-\gamma)^2}.$$

The proof of Theorem 3.1 (2) is complete.

## D.7  Proof of All Supporting Lemmas

### D.7.1  Proof of Lemma B.1

Lemma B.1 (1), (3), and (4) are identical to [10, Proposition 3]. We here only prove Lemma B.1 (2). Consider the Markov chain $\{S_k\}$ induced by $\pi_b$. Since $\{S_k\}$ is irreducible and aperiodic, there exists

a positive integer $r_b$ such that $P_{\pi_b}^{r_b}$ has strictly positive entries [93, Proposition 1.7]. Therefore, there exists $\delta_b \in (0,1)$ such that
$$P_{\pi_b}^{r_b}(s,s') \geq \delta_b \mu_b(s')$$
for all $(s,s')$. In addition, the constant $\rho_b$ introduced after Assumption 3.1 is explicitly given as $\rho_b = \exp(-\delta_b/r_b)$. The previous two equations are from the proof of the Markov chain convergence theorem presented in [93, Section 4.3]. Next, we consider the Markov chain $\{S_k\}$ induced by an arbitrary $\pi \in \Pi_\tau$. Since
$$\frac{\pi_b(a|s)}{\pi(a|s)} = \frac{\pi_b^i(a^i|s)\pi_b^{-i}(a^i|s)}{\pi^i(a^i|s)\pi^{-i}(a^i|s)} \leq \frac{1}{\ell_\tau^2}, \quad \forall\, a = (a^i, a^{-i}) \text{ and } s,$$
we have for any $s, s' \in \mathcal{S}$ and $k \geq 1$ that
$$
\begin{aligned}
P_{\pi_b}^k(s,s') &= \sum_{s_0} P_{\pi_b}^{k-1}(s,s_0) P_{\pi_b}(s_0,s') \\
&= \sum_{s_0} P_{\pi_b}^{k-1}(s,s_0) \sum_{a \in \mathcal{A}} \pi_b(a|s_0) P_a(s_0,s') \\
&= \sum_{s_0} P_{\pi_b}^{k-1}(s,s_0) \sum_{a \in \mathcal{A}} \frac{\pi_b(a|s_0)}{\pi(a|s_0)} \pi(a|s_0) P_a(s_0,s') \\
&\leq \frac{1}{\ell_\tau^2} \sum_{s_0} P_{\pi_b}^{k-1}(s,s_0) \sum_{a \in \mathcal{A}} \pi(a|s_0) P_a(s_0,s') \\
&\leq \frac{1}{\ell_\tau^2} \sum_{s_0} P_{\pi_b}^{k-1}(s,s_0) P_\pi(s_0,s') \\
&= \frac{1}{\ell_\tau^2} [P_{\pi_b}^{k-1} P_\pi](s,s').
\end{aligned}
$$
Since the previous inequality holds for all $s$ and $s'$, we in fact have $\ell_\tau^2 P_{\pi_b}^k \leq P_{\pi_b}^{k-1} P_\pi$ (which is an entry-wise inequality). Repeatedly using the previous inequality, we obtain
$$\ell_\tau^{2k} P_{\pi_b}^k \leq P_\pi^k,$$
which implies
$$
\begin{aligned}
P_\pi^{r_b}(s,s') &\geq \ell_\tau^{2r_b} P_{\pi_b}^{r_b}(s,s') \\
&\geq \delta_b \ell_\tau^{2r_b} \mu_b(s') \\
&\geq \delta_b \ell_\tau^{2r_b} \frac{\mu_b(s')}{\mu_\pi(s')} \mu_\pi(s') \\
&\geq \delta_b \ell_\tau^{2r_b} \mu_{b,\min} \mu_\pi(s').
\end{aligned}
$$
Following the proof of the Markov chain convergence theorem in [93, Section 4.3], we have
$$\|P_\pi^k(s,\cdot) - \mu_\pi(\cdot)\|_{\text{TV}} \leq (1 - \delta_b \ell_\tau^{2r_b} \mu_{b,\min})^{k/r_b - 1}, \quad \forall\, s \in \mathcal{S},\ \pi \in \Pi_\tau. \tag{55}$$
Since $A_{\max} \geq 2$ (otherwise there is no decision to make in this stochastic game), we have $\ell_\tau^2 \leq \frac{1}{2}$. It follows that $1 - \delta_b \ell_\tau^{2r_b} \mu_{b,\min} > 1/2$. Using the previous inequality in Eq. (55), we have
$$
\begin{aligned}
\sup_{\pi \in \Pi_\tau} \max_{s \in \mathcal{S}} \|P_\pi^k(s,\cdot) - \mu_\pi(\cdot)\|_{\text{TV}} &\leq 2(1 - \delta_b \ell_\tau^{2r_b} \mu_{b,\min})^{k/r_b} \\
&\leq 2\exp\left(-\delta_b \ell_\tau^{2r_b} \mu_{b,\min} k/r_b\right) \\
&= 2\rho_b^{\ell_\tau^{2r_b} \mu_{b,\min} k} \qquad \text{(Recall that } \rho_b = \exp(-\delta_b/r_b)) \\
&= 2\rho_\tau^k.
\end{aligned}
$$
We next compute the mixing time. Using the previous inequality and the definition of the total variation distance, we have
$$\sup_{\pi \in \Pi_\tau} \max_{s \in \mathcal{S}} \|P_\pi^k(s,\cdot) - \mu_\pi(\cdot)\|_{\text{TV}} \leq \eta$$
as long as
$$k \geq \frac{\log(2/\eta)}{\log(1/\rho_\delta)} = \frac{1}{\ell_\tau^{2r_b} \mu_{b,\min}} \frac{\log(2/\eta)}{\log(1/\rho_b)} \geq \frac{t_{\pi_b,\eta}}{\ell_\tau^{2r_b} \mu_{b,\min}}.$$

### D.7.2 Proof of Lemma D.11

For any $k \geq z_k$, we have

$$
\begin{aligned}
& \mathbb{E}[(F^i(q_k^i, S_k, A_k^i, A_k^{-i}, S_{k+1}) - \bar{F}_k^i(q_k^i)^\top (q_k^i - \bar{q}_k^i)] \\
& = \underbrace{\mathbb{E}[(F^i(q_{k-z_k}^i, S_k, A_k^i, A_k^{-i}, S_{k+1}) - \bar{F}_{k-z_k}^i(q_{k-z_k}^i))^\top (q_{k-z_k}^i - \bar{q}_{k-z_k}^i)]}_{N_1} \\
& \quad + \underbrace{\mathbb{E}[(F^i(q_{k-z_k}^i, S_k, A_k^i, A_k^{-i}, S_{k+1}) - \bar{F}_{k-z_k}^i(q_{k-z_k}^i))^\top (q_k^i - q_{k-z_k}^i)]}_{N_2} \\
& \quad + \underbrace{\mathbb{E}[(F^i(q_{k-z_k}^i, S_k, A_k^i, A_k^{-i}, S_{k+1}) - \bar{F}_{k-z_k}^i(q_{k-z_k}^i))^\top (\bar{q}_{k-z_k}^i - \bar{q}_k^i)]}_{N_3} \\
& \quad + \underbrace{\mathbb{E}[(F^i(q_k^i, S_k, A_k^i, A_k^{-i}, S_{k+1}) - F^i(q_{k-z_k}^i, S_k, A_k^i, A_k^{-i}, S_{k+1}))^\top (q_k^i - \bar{q}_k^i)]}_{N_4} \\
& \quad + \underbrace{\mathbb{E}[(\bar{F}_{k-z_k}^i(q_{k-z_k}^i) - \bar{F}_k^i(q_k^i))^\top (q_k^i - \bar{q}_k^i)]}_{N_5}.
\end{aligned}
\tag{56}
$$

To bound the terms $N_1$ to $N_5$ on the RHS of the previous inequality, the following lemma is needed.

**Lemma D.13.** *For any positive integers $k_1 \leq k_2$, we have (1)* $\|q_{k_2}^i - q_{k_1}^i\|_\infty \leq \frac{2\alpha_{k_1, k_2-1}}{1-\gamma}$, *and (2)* $\max_{s \in \mathcal{S}} \|\pi_{k_2}^i(s) - \pi_{k_1}^i(s)\|_1 \leq 2\beta_{k_1, k_2-1}$.

*Proof of Lemma D.13.* For any $k \in [k_1, k_2 - 1]$, we have by Eq. (41) that

$$
\|q_{k+1}^i - q_k^i\|_\infty = \alpha_k \|F^i(q_k^i, S_k, A_k^i, A_k^{-i}, S_{k+1})\|_\infty \leq \frac{2\alpha_k}{1-\gamma}.
$$

It follows that $\|q_{k_2}^i - q_{k_1}^i\|_\infty \leq \frac{2\alpha_{k_1, k_2-1}}{1-\gamma}$. Similarly, for any $k \in [k_1, k_2 - 1]$ and $s \in \mathcal{S}$, we have

$$
\|\pi_{k+1}^i(s) - \pi_k^i(s)\|_1 = \beta_k \|\sigma_\tau(q_k^i(s)) - \pi_k^i(s)\|_1 \leq 2\beta_k,
$$

which implies $\max_{s \in \mathcal{S}} \|\pi_{k_2}^i(s) - \pi_{k_1}^i(s)\|_1 \leq 2\beta_{k_1, k_2-1}$. $\qquad \square$

We next bound the terms $N_1$ to $N_5$. Let $\mathcal{F}_k$ be the $\sigma$-algebra generated the sequence of random variables $\{S_0, A_0^i, A_0^{-i}, \cdots, S_{k-1}, A_{k-1}^i, A_{k-1}^{-i}, S_k\}$.

**The Term $N_1$.** Using the tower property of conditional expectations, we have

$$
\begin{aligned}
N_1 & = \mathbb{E}[(F^i(q_{k-z_k}^i, S_k, A_k^i, A_k^{-i}, S_{k+1}) - \bar{F}_{k-z_k}^i(q_{k-z_k}^i))^\top (q_{k-z_k}^i - \bar{q}_{k-z_k}^i)] \\
& = \mathbb{E}[(\mathbb{E}[F^i(q_{k-z_k}^i, S_k, A_k^i, A_k^{-i}, S_{k+1}) \mid \mathcal{F}_{k-z_k}] - \bar{F}_{k-z_k}^i(q_{k-z_k}^i))^\top (q_{k-z_k}^i - \bar{q}_{k-z_k}^i)] \\
& \leq \mathbb{E}[\|\mathbb{E}[F^i(q_{k-z_k}^i, S_k, A_k^i, A_k^{-i}, S_{k+1}) \mid \mathcal{F}_{k-z_k}] - \bar{F}_{k-z_k}^i(q_{k-z_k}^i)\|_1 \|q_{k-z_k}^i - \bar{q}_{k-z_k}^i\|_\infty] \quad (57) \\
& \leq \frac{2}{1-\gamma} \mathbb{E}[\|\mathbb{E}[F^i(q_{k-z_k}^i, S_k, A_k^i, A_k^{-i}, S_{k+1}) \mid \mathcal{F}_{k-z_k}] - \bar{F}_{k-z_k}^i(q_{k-z_k}^i)\|_1] \quad \text{(Lemma D.1)} \\
& \leq \frac{2}{1-\gamma} \mathbb{E}[\|\bar{F}_k^i(q_{k-z_k}^i) - \bar{F}_{k-z_k}^i(q_{k-z_k}^i)\|_1] \\
& \quad + \frac{2}{1-\gamma} \mathbb{E}[\|\mathbb{E}[F^i(q_{k-z_k}^i, S_k, A_k^i, A_k^{-i}, S_{k+1}) \mid \mathcal{F}_{k-z_k}] - \bar{F}_k^i(q_{k-z_k}^i)\|_1]. \quad (58)
\end{aligned}
$$

We next bound the two terms on the RHS of the previous inequality. Observe that

$$
\begin{aligned}
& \|\bar{F}_k^i(q_{k-z_k}^i) - \bar{F}_{k-z_k}^i(q_{k-z_k}^i)\|_1 \\
& = \sum_{s, a^i} |[\bar{F}_k^i(q_{k-z_k}^i)](s, a^i) - [\bar{F}_{k-z_k}^i(q_{k-z_k}^i)](s, a^i)| \\
& = \sum_{s, a^i} \left| [\mathbb{E}_k[F^i(q_{k-z_k}^i, S, A^i, A^{-i}, S')](s, a^i) - [\mathbb{E}_{k-z_k}[F^i(q_{k-z_k}^i, S, A^i, A^{-i}, S')](s, a^i) \right|,
\end{aligned}
$$

where we use $\mathbb{E}_k[\cdot]$ to denote $\mathbb{E}_{S \sim \mu_k(\cdot), A^i \sim \pi_k^i(\cdot|S_0), A^{-i} \sim \pi_k^{-i}(\cdot|S_0), S' \sim p(\cdot|S_0, A_0^i, A_0^{-i})}[\cdot]$ for ease of presentation. To proceed, recall the following equivalent definition of total variation distance between probability measures $p_1, p_2$:

$$\|p_1 - p_2\|_{\text{TV}} = \frac{1}{2} \sup_{f : \|f\|_\infty \leq 1} |\mathbb{E}_{p_1}[f] - \mathbb{E}_{p_2}[f]| .$$

It follows that

$$\left| \left[ \mathbb{E}_k[F^i(q_{k-z_k}^i, S, A^i, A^{-i}, S')](s, a^i) - [\mathbb{E}_{k-z_k}[F^i(q_{k-z_k}^i, S, A^i, A^{-i}, S')](s, a^i) \right|$$

$$\leq \max_{\bar{s}, \bar{a}^i, \bar{a}^{-i}, \bar{s}'} \left| [F^i(q_{k-z_k}^i, \bar{s}, \bar{a}^i, \bar{a}^{-i}, \bar{s}')](s, a^i) \right|$$

$$\times \sum_{\tilde{s}, \tilde{a}^i, \tilde{a}^{-i}} \left| \mu_k(\tilde{s}) \pi_k^i(\tilde{a}^i|\tilde{s}) \pi_k^{-i}(\tilde{a}^{-i}|\tilde{s}) - \mu_{k-z_k}(\tilde{s}) \pi_{k-z_k}^i(\tilde{a}^i|\tilde{s}) \pi_{k-z_k}^{-i}(\tilde{a}^{-i}|\tilde{s}) \right|$$

$$\leq \frac{1}{1-\gamma} \left( \|\mu_k - \mu_{k-z_k}\|_1 + \max_s \|\pi_k^i(s) - \pi_{k-z_k}^i(s)\|_1 + \max_s \|\pi_k^{-i}(s) - \pi_{k-z_k}^{-i}(s)\|_1 \right)$$

$$\leq \frac{2L_p}{1-\gamma} \left( \max_s \|\pi_k^i(s) - \pi_{k-z_k}^i(s)\|_1 + \max_s \|\pi_k^{-i}(s) - \pi_{k-z_k}^{-i}(s)\|_1 \right) \qquad \text{(Lemma B.1)}$$

$$\leq \frac{8L_p \beta_{k-z_k, k-1}}{1-\gamma}. \qquad \text{(Lemma D.13)}$$

Therefore, we have

$$\|\bar{F}_k^i(q_{k-z_k}^i) - \bar{F}_{k-z_k}^i(q_{k-z_k}^i)\|_1 = \sum_{s, a^i} |[\bar{F}_k^i(q_{k-z_k}^i)](s, a^i) - [\bar{F}_{k-z_k}^i(q_{k-z_k}^i)](s, a^i)|$$

$$\leq \frac{8|\mathcal{S}| A_{\max} L_p \beta_{k-z_k, k-1}}{1-\gamma}. \qquad (59)$$

It remains to bound the second term on the RHS of Eq. (58). Recall that we denote $P_\pi \in \mathbb{R}^{|\mathcal{S}| \times |\mathcal{S}|}$ as the transition probability matrix of the Markov chain $\{S_k\}$ induced by a joint policy $\pi$. Using the definition of conditional expectations, we have

$$\|\mathbb{E}[F^i(q_{k-z_k}^i, S_k, A_k^i, A_k^{-i}, S_{k+1}) \mid \mathcal{F}_{k-z_k}] - \bar{F}_k^i(q_{k-z_k}^i)\|_1$$

$$= \left\| \sum_{s \in \mathcal{S}} \left[ \left( \prod_{j=k+1}^{k+z_k} P_{\pi_{j-z_k}} \right)(S_{k-z_k}, s) - \mu_k(s) \right] \sum_{a^i} \pi_k^i(a^i|s) \sum_{a^{-i}} \pi_k^{-i}(a^{-i}|s) \right.$$

$$\left. \times \sum_{s'} p(s'|s, a^i, a^{-i}) F^i(q_{k-z_k}^i, s, a^i, a^{-i}, s') \right\|_1$$

$$\leq \frac{2}{1-\gamma} \sum_{s \in \mathcal{S}} \left| \left( \prod_{j=k+1}^{k+z_k} P_{\pi_{j-z_k}} \right)(S_{k-z_k}, s) - \mu_k(s) \right| \qquad \text{(Lemma D.9)}$$

$$\leq \frac{2}{1-\gamma} \left\{ \sum_{s \in \mathcal{S}} \left| \left( \prod_{j=k+1}^{k+z_k} P_{\pi_{j-z_k}} \right)(S_{k-z_k}, s) - P_{\pi_k}^{z_k}(S_{k-z_k}, s) \right| \right.$$

$$\left. + \sum_{s \in \mathcal{S}} \left| P_{\pi_k}^{z_k}(S_{k-z_k}, s) - \mu_k(s) \right| \right\}$$

$$\leq \frac{2}{1-\gamma} \left\{ \left\| \prod_{j=k+1}^{k+z_k} P_{\pi_{j-z_k}} - P_{\pi_k}^{z_k} \right\|_\infty + 2\rho_\tau^{z_k} \right\}, \qquad (60)$$

where the last line follows from Lemma B.1 (2). Observe that

$$\left\| \prod_{j=k+1}^{k+z_k} P_{\pi_{j-z_k}} - P_{\pi_k}^{z_k} \right\|_\infty = \left\| \sum_{\ell=1}^{z_k} \left( \prod_{j=k+1}^{k-\ell+1+z_k} P_{\pi_{j-z_k}} P_{\pi_k}^{\ell-1} - \prod_{j=k+1}^{k-\ell+z_k} P_{\pi_{j-z_k}} P_{\pi_k}^\ell \right) \right\|_\infty$$

$$= \left\| \sum_{\ell=1}^{z_k} \left( \prod_{j=k+1}^{k-\ell+z_k} P_{\pi_{j-z_k}} (P_{\pi_{k-\ell+1}} - P_{\pi_k}) P_{\pi_k}^{\ell-1} \right) \right\|_\infty$$

$$\leq \sum_{\ell=1}^{z_k} \left\| \prod_{j=k+1}^{k-\ell+z_k} P_{\pi_{j-z_k}} \right\|_\infty \| P_{\pi_{k-\ell+1}} - P_{\pi_k} \|_\infty \| P_{\pi_k}^{\ell-1} \|_\infty$$

$$\leq \sum_{\ell=1}^{z_k} \| P_{\pi_{k-\ell+1}} - P_{\pi_k} \|_\infty.$$

Since $P_\pi$ as a function of $\pi$ is 1-Lipschitz continuous with respect to the $\ell_\infty$-norm, we have

$$\left\| \prod_{j=k+1}^{k+z_k} P_{\pi_{j-z_k}} - P_{\pi_k}^{z_k} \right\|_\infty \leq \sum_{\ell=1}^{z_k} \max_{s \in \mathcal{S}} \| \pi_{k-\ell+1}(s) - \pi_k(s) \|_1$$

$$= \sum_{\ell=1}^{z_k} \max_{s \in \mathcal{S}} \left( \| \pi_{k-\ell+1}^{-i}(s) - \pi_k^{-i}(s) \|_1 + \| \pi_{k-\ell+1}^{i}(s) - \pi_k^{i}(s) \|_1 \right)$$

$$\leq 4 z_k \beta_{k-z_k, k-1}. \qquad \text{(Lemma D.13)}$$

Using the previous inequality in Eq. (60), we have

$$\| \mathbb{E}[F^i(q_{k-z_k}^i, S_k, A_k^i, A_k^{-i}, S_{k+1}) \mid \mathcal{F}_{k-z_k}] - \bar{F}_k^i(q_{k-z_k}^i) \|_1$$

$$\leq \frac{2}{1-\gamma} \left( 4 z_k \beta_{k-z_k, k-1} + 2 \rho_\tau^{z_k} \right)$$

$$\leq \frac{2}{1-\gamma} \left( 4 z_k \beta_{k-z_k, k-1} + \beta_k \right) \qquad \text{(Definition of } z_k)$$

$$\leq \frac{10 z_k \beta_{k-z_k, k-1}}{1-\gamma}. \qquad (z_k \geq 1)$$

Using the previous inequality and Eq. (59) together in Eq. (58), we obtain

$$N_1 \leq \frac{16 L_p |\mathcal{S}| A_{\max} \beta_{k-z_k, k-1}}{(1-\gamma)^2} + \frac{20 z_k \beta_{k-z_k, k-1}}{(1-\gamma)^2} \leq \frac{36 L_p |\mathcal{S}| A_{\max} z_k \beta_{k-z_k, k-1}}{(1-\gamma)^2}.$$

**The Term $N_2$.** For any $k \geq z_k$, we have by Lemma D.13 that

$$N_2 \leq \mathbb{E}[\| F^i(q_{k-z_k}^i, S_k, A_k^i, A_k^{-i}, S_{k+1}) - \bar{F}_{k-z_k}^i(q_{k-z_k}^i) \|_1 \| q_k^i - q_{k-z_k}^i \|_\infty]$$

$$\leq \frac{2 \alpha_{k-z_k, k-1}}{1-\gamma} \mathbb{E}[\| F^i(q_{k-z_k}^i, S_k, A_k^i, A_k^{-i}, S_{k+1}) \|_1 + \| \bar{F}_{k-z_k}^i(q_{k-z_k}^i) \|_1]. \qquad (61)$$

Using the definition of $F^i(\cdot)$, we have

$$\| F^i(q_{k-z_k}^i, S_k, A_k^i, A_k^{-i}, S_{k+1}) \|_1$$

$$= \sum_{s, a^i} \mathbb{1}_{\{(s, a^i) = (S_k, A_k^i)\}} \left| R_i(S_k, A_k^i, A_k^{-i}) + \gamma v^i(S_{k+1}) - q_k^i(S_k, A_k^i) \right|$$

$$= \left| R_i(S_k, A_k^i, A_k^{-i}) + \gamma v^i(S_{k+1}) - q_k^i(S_k, A_k^i) \right|$$

$$\leq 1 + \frac{\gamma}{1-\gamma} + \frac{1}{1-\gamma}$$

$$= \frac{2}{1-\gamma}. \qquad (62)$$

Moreover, we have by Jensen's inequality that

$$\| \bar{F}_{k-z_k}^i(q_{k-z_k}^i) \|_1 \leq \frac{2}{1-\gamma}. \qquad (63)$$

Using Eqs. (62) and (63) together in Eq. (61), we have

$$N_2 \leq \frac{8 \alpha_{k-z_k, k-1}}{(1-\gamma)^2}.$$

**The Term $N_3$.** For any $k \geq z_k$, we have

$$
\begin{aligned}
N_3 &\leq \mathbb{E}[\|F^i(q^i_{k-z_k}, S_k, A^i_k, A^{-i}_k, S_{k+1}) - \bar{F}^i_{k-z_k}(q^i_{k-z_k})\|_1 \|\bar{q}^i_{k-z_k} - \bar{q}^i_k\|_\infty] \\
&\leq \mathbb{E}[(\|F^i(q^i_{k-z_k}, S_k, A^i_k, A^{-i}_k, S_{k+1})\|_1 + \|\bar{F}^i_{k-z_k}(q^i_{k-z_k})\|_1)\|\bar{q}^i_{k-z_k} - \bar{q}^i_k\|_\infty] \\
&\leq \frac{4}{1-\gamma}\mathbb{E}[\|\bar{q}^i_{k-z_k} - \bar{q}^i_k\|_\infty]. \qquad\qquad\qquad\qquad \text{(Eqs. (62) and (63))}
\end{aligned}
$$

Observe that

$$
\begin{aligned}
\|\bar{q}^i_{k-z_k} - \bar{q}^i_k\|_\infty &= \max_{s \in \mathcal{S}} \|\mathcal{T}^i(v^i)(s)(\pi^{-i}_k(s) - \pi^{-i}_{k-z_k}(s))\|_\infty \\
&\leq \max_{s \in \mathcal{S}} \|\mathcal{T}^i(v^i)(s)\|_{1,\infty} \|\pi^{-i}_k(s) - \pi^{-i}_{k-z_k}(s)\|_1 \\
&\leq \frac{2\beta_{k-z_k, k-1}}{1-\gamma},
\end{aligned}
$$

where the last line follows from Lemma D.13 and

$$
\|\mathcal{T}^i(v^i)(s)\|_{1,\infty} \leq \max_{s,a^i,a^{-i}} |\mathcal{T}^i(v^i)(s, a^i, a^{-i})| \leq \frac{1}{1-\gamma}.
$$

Therefore, we have

$$
N_3 \leq \frac{8\beta_{k-z_k, k-1}}{(1-\gamma)^2}.
$$

**The Term $N_4$.** For any $k \geq z_k$, we have

$$
\begin{aligned}
N_4 &\leq \mathbb{E}[\|F^i(q^i_k, S_k, A^i_k, A^{-i}_k, S_{k+1}) - F^i(q^i_{k-z_k}, S_k, A^i_k, A^{-i}_k, S_{k+1})\|_1 \|q^i_k - \bar{q}^i_k\|_\infty] \\
&\leq \frac{2}{1-\gamma}\mathbb{E}[\|F^i(q^i_k, S_k, A^i_k, A^{-i}_k, S_{k+1}) - F^i(q^i_{k-z_k}, S_k, A^i_k, A^{-i}_k, S_{k+1})\|_1],
\end{aligned}
$$

where the last line follows from Lemma D.1. Using the definition of $F^i(\cdot)$, we have

$$
\begin{aligned}
&\|F^i(q^i_k, S_k, A^i_k, A^{-i}_k, S_{k+1}) - F^i(q^i_{k-z_k}, S_k, A^i_k, A^{-i}_k, S_{k+1})\|_1 \\
&= \sum_{s,a^i} \mathbb{1}_{\{(s,a^i)=(S_k, A^i_k)\}} \left|q^i_{k-z_k}(S_k, A^i_k) - q^i_k(S_k, A^i_k)\right| \\
&= \left|q^i_{k-z_k}(S_k, A^i_k) - q^i_k(S_k, A^i_k)\right| \\
&\leq \|q^i_{k-z_k} - q^i_k\|_\infty \\
&\leq \frac{2\alpha_{k-z_k, k-1}}{1-\gamma}. \qquad\qquad\qquad\qquad\qquad\qquad\qquad\qquad\qquad \text{(Lemma D.13)}
\end{aligned}
$$

It follows that

$$
N_4 \leq \frac{4\alpha_{k-z_k, k-1}}{(1-\gamma)^2}.
$$

**The Term $N_5$.** For any $k \geq z_k$, we have

$$
\begin{aligned}
N_5 &\leq \mathbb{E}[\|\bar{F}^i_k(q^i_k) - \bar{F}^i_{k-z_k}(q^i_{k-z_k})\|_1 \|q^i_k - \bar{q}^i_k\|_\infty] \\
&\leq \frac{2}{1-\gamma}\mathbb{E}[\|\bar{F}^i_k(q^i_k) - \bar{F}^i_{k-z_k}(q^i_{k-z_k})\|_1] \qquad\qquad\qquad\qquad \text{(Lemma D.1)} \\
&\leq \frac{2}{1-\gamma}\mathbb{E}[\|\bar{F}^i_k(q^i_k) - \bar{F}^i_{k-z_k}(q^i_k)\|_1 + \|\bar{F}^i_{k-z_k}(q^i_k) - \bar{F}^i_{k-z_k}(q^i_{k-z_k})\|_1] \\
&\leq \frac{16 L_p |\mathcal{S}| A_{\max}\beta_{k-z_k, k-1}}{(1-\gamma)^2} + \frac{2}{1-\gamma}\mathbb{E}[\|\bar{F}^i_{k-z_k}(q^i_k) - \bar{F}^i_{k-z_k}(q^i_{k-z_k})\|_1], \qquad (64)
\end{aligned}
$$

where the last line follows from the same analysis as we obtain Eq. (59). As for the second term on the RHS of Eq. (64), using the definition of $\bar{F}^i_k(\cdot)$, we have

$$
\|\bar{F}^i_{k-z_k}(q^i_k) - \bar{F}^i_{k-z_k}(q^i_{k-z_k})\|_1 = \sum_{s \in \mathcal{S}} \mu_{k-z_k}(s) \sum_{a^i} \pi^i_{k-z_k}(a^i \mid s)|q^i_k(s, a^i) - q^i_{k-z_k}(s, a^i)|
$$

$$\leq \|q_k^i - q_{k-z_k}^i\|_\infty$$
$$\leq \frac{2\alpha_{k-z_k,k-1}}{1-\gamma}. \qquad \text{(Lemma D.13)}$$

Using the previous inequality in Eq. (64), we obtain

$$N_5 \leq \frac{16 L_p |\mathcal{S}| A_{\max} \beta_{k-z_k,k-1}}{(1-\gamma)^2} + \frac{4\alpha_{k-z_k,k-1}}{(1-\gamma)^2}.$$

Combining the upper bounds we derived for the terms $N_1$ to $N_5$ in Eq. (56), we have

$$\mathbb{E}[(F^i(q_k^i, S_k, A_k^i, A_k^{-i}, S_{k+1}) - \bar{F}_k^i(q_k^i)^\top (q_k^i - \bar{q}_k^i)]$$
$$\leq \frac{36 L_p |\mathcal{S}| A_{\max} z_k \beta_{k-z_k,k-1}}{(1-\gamma)^2} + \frac{8\alpha_{k-z_k,k-1}}{(1-\gamma)^2} + \frac{8\beta_{k-z_k,k-1}}{(1-\gamma)^2} + \frac{4\alpha_{k-z_k,k-1}}{(1-\gamma)^2}$$
$$+ \frac{16 L_p |\mathcal{S}| A_{\max} \beta_{k-z_k,k-1}}{(1-\gamma)^2} + \frac{4\alpha_{k-z_k,k-1}}{(1-\gamma)^2}$$
$$\leq \frac{60 L_p |\mathcal{S}| A_{\max} z_k \beta_{k-z_k,k-1}}{(1-\gamma)^2} + \frac{16\alpha_{k-z_k,k-1}}{(1-\gamma)^2}$$
$$\leq \frac{17 z_k \alpha_{k-z_k,k-1}}{(1-\gamma)^2},$$

where the last line follows from $\beta_k/\alpha_k = c_{\alpha,\beta} \leq \frac{1}{60 L_p |\mathcal{S}| A_{\max}}$ (cf. Condition 3.1).

## D.8 Proof of Corollary 3.1.1

The following proof idea was previous used in [15] to show the rationality of their decentralized $Q$-learning algorithm.

Observe that Theorem 3.1 can be easily generalized to the case where the reward is corrupted by noise. Specifically, suppose that player $i$ takes action $a^i$ and player $-i$ takes action $a^{-i}$. Instead of assuming player $i$ receives a deterministic reward $R_i(s, a^i, a^{-i})$, we assume that player $i$ receives a random reward $r^i(s, a^i, a^{-i}, \xi)$, where $\xi \in \Xi$ ($\Xi$ is a finite set) is a random variable with distribution $\mu_\xi(s)$, and is independent of everything else. The proof is identical as long as $r^i + r^{-i} = 0$, and the reward is uniformly bounded, i.e., $\max_{s,a^i,a^{-i},\xi} |r^i(s, a^i, a^{-i}, \xi)| < \infty$. Now consider the case where player $i$'s opponent follows a stationary policy $\pi^{-i}$. We incorporate the randomness of player $-i$'s action into the model and introduce a fictitious opponent with only one action $a^*$. In particular, let the random reward function be defined as $\hat{r}^i(s, a^i, a^*, A^{-i}) = R_i(s, a^i, A^{-i})$ for all $(s, a^i)$, where $A^{-i} \sim \pi^{-i}(\cdot|s)$, and let $\hat{p}(s' \mid s, a^i, a^*) = \sum_{\pi^{-i}(a^{-i}|s)} p(s' \mid a^i, a^{-i}, s)$. Now the problem can be reformulated as player $i$ playing against the fictitious player with a single action $a^*$, with reward function $\hat{r}^i$ ($i \in \{1,2\}$) and transition probabilities $\hat{p}$. Using the same proof for Theorem 3.1, we have the desired finite-sample bound.

# E On the Mixing Time of MDPs with Almost Deterministic Policies

Consider an MDP with two states $s_1, s_2$ and two actions $a_1, a_2$. The transition probability matrix $P_1$ of taking action $a_1$ is the identity matrix $I_2$, and the transition probability matrix $P_2$ of taking action $a_2$ is $P_2 = [0, 1; 1, 0]$. Given $\alpha \in (1/2, 1)$, let $\pi_\alpha$ be a policy such that $\pi(a_1|s) = \alpha$ and $\pi(a_2|s) = 1 - \alpha$ for any $s \in \{s_1, s_2\}$. Denote $P_\alpha$ as the transition probability matrix under $\pi_\alpha$. It is easy to see that

$$P_\alpha = \begin{bmatrix} \alpha & 1-\alpha \\ 1-\alpha & \alpha \end{bmatrix}.$$

Since $P_\alpha$ is a doubly stochastic matrix, and has strictly positive entries, it has a unique stationary distribution $\mu = \mathbf{1}^\top/2$.

We next compute a lower bound of the mixing time of the $\pi_\alpha$-induced Markov chain. Let $e_1 = [1, 0]^\top$ be the initial distribution of the states, and denote $[x_k, 1 - x_k]^\top$ as the distribution of the states at time step $k$. Then we have

$$x_{k+1} = x_k \alpha + (1 - x_k)(1 - \alpha)$$

$$= (2\alpha - 1)x_k + 1 - \alpha$$

$$= (2\alpha - 1)^{k+1}x_0 + \sum_{i=0}^{k}(1 - \alpha)(2\alpha - 1)^{k-i}$$

$$= \frac{1}{2} + \frac{(2\alpha - 1)^{k+1}}{2}.$$

It follows that

$$
\begin{aligned}
t_{\pi_\alpha, \eta} &= \min_{k \geq 0}\left\{\max_{\mu_0 \in \Delta^2}\left\|\mu_0^\top P_\alpha^k - \mathbf{1}^\top/2\right\|_{\mathrm{TV}} \leq \eta\right\} \\
&\geq \min_{k \geq 0}\left\{\left\|e_1^\top P_\alpha^k - \mathbf{1}^\top/2\right\|_{\mathrm{TV}} \leq \eta\right\} \\
&= \min_{k \geq 0}\left\{(2\alpha - 1)^k \leq 2\eta\right\} \\
&\geq \frac{\log(1/2\eta)}{\log(1/(2\alpha - 1))} - 1,
\end{aligned}
$$

which implies $\lim_{\alpha \to 1} t_{\alpha, \eta} = \infty$. Therefore, as the policies become deterministic, the mixing time of the associated Markov chain can approach infinity.

# F   Numerical Simulations

We first conduct numerical simulations to investigate the impact of choosing different $\tau$, which is used to define the softmax operator in Algorithms 1 and 2. Our theoretical results indicate that there is an asymptotically non-vanishing bias due to using a positive $\tau$. Intuitively, since a softmax policy always has strictly positive entries while a Nash equilibrium policy can have zero entries, we cannot, in general, expect the Nash gap to converge to zero.

To demonstrate this phenomenon, consider the following example of a zero-sum matrix game. Let

$$
R_1 = \begin{bmatrix} N & 1 & -1 \\ -1 & 0 & 1 \\ 1 & -1 & 0 \end{bmatrix}
$$

be the payoff matrix for player 1, and let $R_2 = -(R_1)^\top$, where $N > 0$ is a tunable parameter. Note that this matrix game has a unique Nash equilibrium, which goes to the joint policy $\pi^1 = (1/3, 2/3, 0)$, $\pi^2 = (0, 2/3, 1/3)$ as $N \to \infty$. In our simulations, we use constant stepsizes $\alpha_k \equiv 0, 5$ and $\beta_k \equiv 0.01$ and run Algorithm 1 for 100 trajectories (each has $K = 2000$ iterations). Then, we plot the average Nash gap (averaged over the 100 trajectories) as a function of the number of iterations $k$ in Figure 1 for different temperatures $\tau$. To enable a fair comparison, we use the normalized $q$-function to compute the softmax, that is, instead of directly using $\sigma_\tau(q_k^i)$ in Algorithm 1, we use $\sigma_\tau(q_k^i/\|q_k^i\|_2)$. As we can see in Figure 1, as $\tau$ increases, the asymptotic error also increases, which is consistent with our theoretical results.

## F.1   Comparison with the Optimistic Multiplicative-Weights Update

The Optimistic Multiplicative-Weights Update (OMWU) was recognized as a popular learning algorithm for zero-sum matrix games [98]. Since OMWU in the payoff-based setting (or noisy-feedback setting) may not have last-iterate convergence [99], to enable a fair comparison, we will compare OMWU in the noiseless setting (which does enjoy last-iterate convergence [98]) with smoothed-best response dynamics. We start by writting down the algorithm.

**OMWU:** With initializations $\pi_0^1, \pi_1^1$ (respectively, $\pi_0^2, \pi_1^2$) that live in the interior of the probability simplex $\Delta(\mathcal{A}^1)$ (respectively, $\Delta(\mathcal{A}^2)$), OMWU updates $(\pi_k^1, \pi_k^2)$ iteratively according to

$$
\pi_{k+1}^i(a^i) = \frac{\pi_k^i(a^i)\exp(2\eta[R_i\pi_k^{-i}](a^i) - \eta[R_i\pi_{k-1}^{-i}(a^i)])}{\sum_{\tilde{a}^i \in \mathcal{A}^i}\pi_k^i(\tilde{a}^i)\exp(2\eta[R_i\pi_k^{-i}](\tilde{a}^i) - \eta[R_i\pi_{k-1}^{-i}(\tilde{a}^i)])}, \quad \forall a^i \in \mathcal{A}^i, i \in \{1, 2\},
$$

where $\eta \in (0, 1)$ is the stepsize.

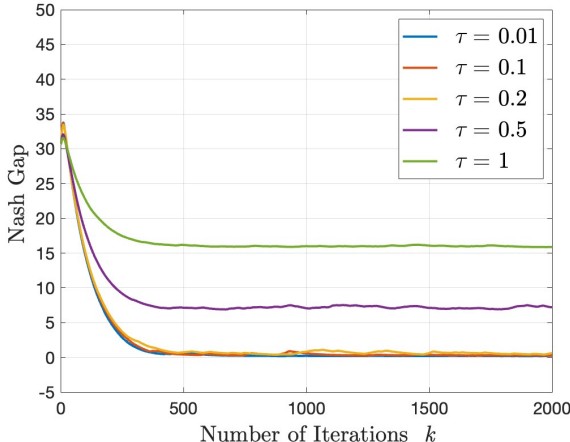

Figure 1: The Nash Gap for Different Temperatures $\tau$

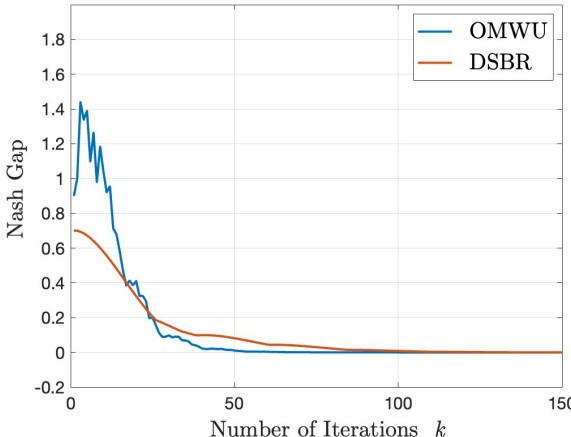

Figure 2: The Nash Gap as a Function of the Number of Iterations $k$

**The Discrete Smoothed Best-Response Dynamics (DSBR):** With arbitrary initializations $\pi_0^1 \in \Delta(\mathcal{A}^1)$ and $\pi_0^2 \in \Delta(\mathcal{A}^2)$, the discrete smoothed best-response dynamics update $(\pi_k^1, \pi_k^2)$ iteratively according to

$$\pi_{k+1}^i = (1 - \beta_k)\pi_k^i + \beta_k \sigma_\tau(R_i \pi_k^{-i}), \quad \forall i \in \{1, 2\},$$

where $\beta_k$ is the stepsize.

We perform two sets of numerical simulations to compare OMWU and DSBR. Our first experiment is implemented on the rock-paper-scissor game, where the payoff matrix for player 1 is

$$R_1 = \begin{bmatrix} 0 & 1 & -1 \\ -1 & 0 & 1 \\ 1 & -1 & 0 \end{bmatrix},$$

and $R_2 = -(R_1)^\top$. As we see in Figure 2, the convergence rates of OMWU and DSBR are comparable. However, DSBR seems to be more stable compared with OMWU. Note that while we use softmax policies in DSBR, since the rock-paper-scissor game has a unique Nash equilibrium, which is also the unique Nash equilibrium of the entropy-regularized matrix game for any temperature $\tau > 0$, there is no smoothing bias and the Nash gap under DSBR does converge to zero.

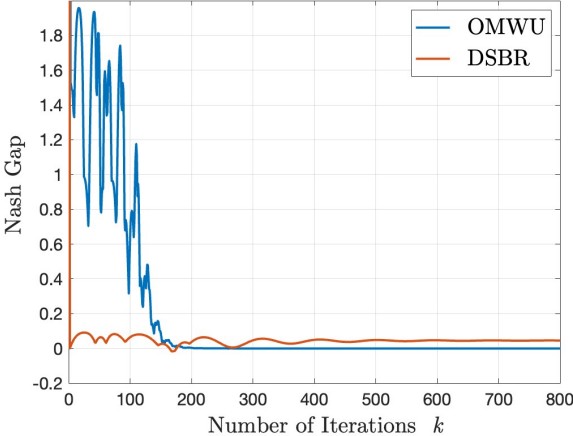

Figure 3: The Nash Gap as a Function of the Number of Iterations $k$

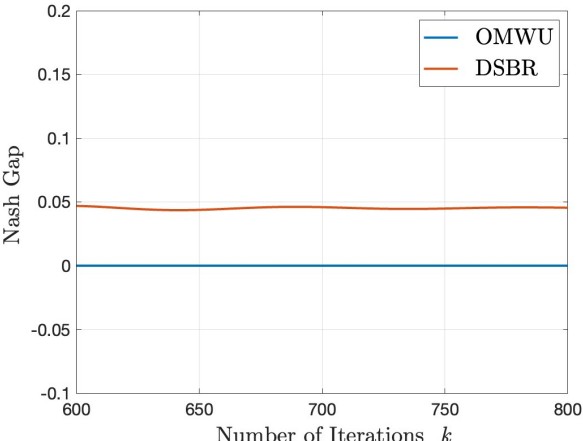

Figure 4: The Asymptotic Behavior of Figure 3

In our second numerical simulation, we set the payoff matrix of player 1 to be

$$R_1 = \begin{bmatrix} N & 1 & -1 \\ -1 & 0 & 1 \\ 1 & -1 & 0 \end{bmatrix},$$

and $R_2 = -(R_1)^\top$, where we choose $N = 100$. Note that as $N \to \infty$, the unique Nash equilibrium goes to $\pi^1 = (1/3, 2/3, 0)$, $\pi^2 = (0, 2/3, 1/3)$. In this case, we also see from Figure 3 that DSBR is more stable compared with OMWU. However, since in this case, the Nash equilibrium has zero entries, due to the use of softmax policies, DSBR suffers from an asymptotically non-vanishing bias. This is clear from Figure 4, which plots the asymptotic behavior of Figure 3. We see that OMWU converges to zero while the Nash gap from DSBR converges to a positive real number.

