# OpenReview forum: "A Finite-Sample Analysis of Payoff-Based Independent Learning in Zero-Sum Stochastic Games"
_NeurIPS.cc/2023/Conference — NeurIPS 2023 poster_

### Official Review · Reviewer_bFVC · 2023-07-05

**Soundness:** 3 good
**Presentation:** 3 good
**Contribution:** 2 fair
**Rating:** 5
**Confidence:** 4

**Summary:**

The authors propose a doubly smoothed best-response dynamics for two-player zero-sum Markov games, with matrix games as the degenerate case. Upper bounds of Nash Gap with bias and regularized Nash Gap without bias are presented.

**Strengths:**

The technical sections are concrete and solid. The idea of simultaneously smoothly updating in the value (q) space and the policy (simplex) space appears novel. In such a way, each player is facing a much more stable environment than simply smoothing in the value space. As a consequence, the convergence results are natural outcomes of such an algorithmic design. Moreover, the Assumption 3.1 makes the results more appealing in a sense that it deviates from typically made assumptions in RL analysis.

**Weaknesses:**

Some major concerns:

(1) The "first form of smoothing" in the policy space is very similar to the mixed strategy in fictitious self-play but not exactly the same. In fictitious self-play, the policies are mixed in the way of having $1-\alpha_k$ vs $\alpha_k$ probabilities to select the old policy vs the new best response. In comparison, here the authors simply averaged two stochastic strategies. Even with the explanations provided in the paper, it is still not very clear to me why the authors would want to deviate from the classical practice in learning Markov games (or extensive form games), which is backed by enormous amount of practical implementations.

(2) While the authors claim that the learning dynamics are independent for the two players, the learning rates of them are tied through a constant, which makes the claim somewhat questionable. Please elaborate more about your definition of being "independent".

(3) The bound for Nash gap is somewhat weak. Apart from the term $\ell_\tau$ that is exponential in $\tau$, the constant $c_2$ also has both $O(\tau)$ and  $O(\tau^{-1})$ terms, which requires balancing. Overall, I think a corollary with the best choice of $\tau$ is highly desirable to at least let readers to directly evaluate the strength of the bound.

(4) The bound of regularized Nash gap is not at all surprising -- it appears rather standard in minimax (saddle point) optimization with regularized geometry. So the real emphasis should be the Nash gap itself. Although I see the authors discuss the natural difficulty for softmax policy algorithms to converge without bad dependencies on the temperature $\tau$, it still feels quite vague why this is the case -- I believe a deeper illustration with math will greatly strengthen the paper -- it won't make the Nash gap look as unsatisfying as it appears now.

**Questions:**

Please see the Weakness section.

**Limitations:**

Yes, there is no potential negative societal impact.

---

> ### Author Rebuttal · Authors · 2023-08-07
>
> We thank the reviewer for the nice comments about our learning dynamics and the presentation of our technical sections. We next provide a point-by-point response to the reviewer's comments.
>
> **Comment:** The "first form of smoothing" in the policy space ...
>
> **Response:** To our knowledge, fictitious play refers to the learning dynamics where each player estimates its opponent's policy by taking an empirical average of the opponent's historical actions and then best responds to that estimated policy. It is not entirely clear to us whether the fictitious self-play (the reviewer refers to) is the same as the fictitious play we illustrated above. We would greatly appreciate it if the reviewer can provide more information or pointers to the references.
>
> On the other hand, it seems that the learning dynamics the reviewer describes ("the policies are mixed in the way of having  $(1-\alpha_k)$ vs $\alpha_k$  probabilities to select the old policy vs the new best response") is exactly what we are doing here. Using zero-sum matrix games for illustration, the new policy from our policy update equation is a convex combination of the old policy and the new best response (estimated through the $q$-function):
>
> $\pi_{k+1}^i=\pi_k^i+\beta_k(\sigma_\tau(q_k^i)-\pi_k^i)=(1-\beta_k)\pi_k^i+\beta_k\sigma_\tau(q_k^i)$,
>
> which can exactly be interpreted as w.p. $1-\beta_k$ taking actions according to the old policy and w.p. $\beta_k$ taking actions according to the new best response (up to a smoothing using softmax). The reason that we choose the convex combination parameter to be $\beta_k\ll \alpha_k$ (which is the stepsize for the $q$-function) is to ensure that the policies are evolving in a slower time scale.
>
> **Comment:** While the authors claim that ...
>
> **Response:** Our learning dynamics is independent in the sense that to carry out the algorithm, each player only needs to use its local information, i.e., its own actions and the realized payoffs. We agree with the reviewer that for our algorithm to achieve provable guarantees, the stepsizes used by the two players should be of the same order. We will clarify this in our revised manuscript. That being said, this type of conditions on stepsizes are actually common in the existing literature studying independent learning dynamics. For example, even for asymptotic convergence, the results in Leslie and Collins (2005), Sayin et al (2021) require the stepsizes used by the two players to be of the same order.
>
> Leslie, D. S., & Collins, E. J. (2005). Individual Q-learning in normal form games. SIAM Journal on Control and Optimization, 44(2), 495-514.
>
> Sayin, M., Zhang, K., Leslie, D., Basar, T., & Ozdaglar, A. (2021). Decentralized Q-learning in zero-sum Markov games. Advances in Neural Information Processing Systems, 34, 18320-18334.
>
> **Comment:** The bound for Nash gap is somewhat weak ...
>
> **Response:** Please see the response to the common comments from the reviewers in the beginning of this page.
>
> **Comment:** The bound of regularized Nash gap is not at all surprising ...
>
> **Response:** We agree with the reviewer that the finite-sample bound is qualitatively similar to existing work studying saddle point optimization problems. However, there are two major differences of this work in terms of the algorithm and the sampling. One is that almost all existing results studying saddle point problems in optimization use gradient-based method (including mirror descent), while our learning dynamics is best-response type, which is more natural in the game setting. Second, the sample collection process in learning is significantly different from that in optimization. In particular, to estimate the marginalized payoff to the opponent, our learning dynamics introduces the $q$-function, which is updated using an asynchronous stochastic approximation algorithm based on the realized payoffs. The fact that the update for the $q$-function is asynchronous is the main reason for us to have a constant that is exponential in $\tau$ in the bound, which we will illustrate in more detail in the next paragraph. To our knowledge, for payoff-based best-response learning dynamics, there are no existing results that provide last-iterate finite-sample analysis.
>
> We next provide a detailed elaboration on why we have an exponential dependence on $\tau$ in the bound. For simplicity of illustration, consider the zero-sum matrix game setting. The update equation for the $q$-functions in Algorithm 1 performs asynchronous update as only one component (which corresponds to the action taken at time step $k$) of the $q$-function is updated in the $k$-th iteration. Therefore, suppose an action is never taken in the algorithm trajectory, we cannot hope for convergence as the specific component of the $q$-function is never updated during learning. Similarly, suppose an action is rarely taken in the learning dynamics, we would expect the overall convergence rate to be slow. Therefore, the finite-sample bound should depend on the minimum frequency of taking actions in the learning process. This is captured by the quantity $\min_{1\leq k\leq K}\min_{a^i}\pi_k^i(a^i)$, which is lowered bounded in Lemma D.1. Due to the exponential nature of softmax functions, the lower bound is also exponential in $\tau$, which eventually leads to the exponential dependence in $\tau$ in the finite-sample bound.
>
> We will add this discussion with more mathematical details to the next version of this work.

---

> > ### Comment · Reviewer_bFVC · 2023-08-14
> > **Re: Rebuttal by Authors**
> >
> > Thanks for your responses and I am generally glad with the changes the authors promise to make in the next version. Since some weaknesses still stand, I'm raising my score to 5.

---

> > > ### Author Response · Authors · 2023-08-14
> > >
> > > We thank the reviewer for the feedback and if you have any other questions, please let us know.

---

### Official Review · Reviewer_zACq · 2023-07-07

**Soundness:** 3 good
**Presentation:** 4 excellent
**Contribution:** 3 good
**Rating:** 6
**Confidence:** 3

**Summary:**

This paper studied a best-response type learning dynamics in two-player zero-sum stochastic games called doubly smoothed best-response dynamics. The dynamics uses smoothed value function updates with softmax smoothed best-response, and combines minimax value iteration. Ths dynamics is payoff-based, convergent, rational, and symmetric. The author also provided the first finite-sample analysis of these type of dynamics, showing convergence to Nash equilibrium up to smoothing bias.

**Strengths:**

1. Finite-sample analysis (or convergence rates analysis) for payoff-based (or noisy bandit feedback-based) learning dynamics in stochasitc games is an interesing question with practical importance. This paper contributes to this area by the first finite-sample analysis of a payoff-based best-response-type independent learning dynamics for zero-sum stochastic games.
2. This paper is well-written and easy to follow. I appreciated it that the authors provided detailed discussion and high-level ideas of the algorithm and proof sketch for the main results.

**Weaknesses:**

The claim that the proposed dynamics is *convergent* and *rational* seems not rigorous since the smoothing bias $\tau$ persists over time and no convergence to exact Nash equilirium or best response is really proved. One possible fix might be introducing a decreasing parameter $\tau_t$ so that exact convergence can be shown. This might significantly slow down the convergence of other bias terms because of the exponetionally dependence on $\tau$. The proposed dynamics is thus very slow in terms of convergence to Nash equilirium and this is a weakness of the current work.

Minor comments:

1. Line 277, $Q(s,a^{i}, a^{-i})$ has not been defined.
2. Under Line 284: one term should be $R^i(s, a^{i}, a^{-i})$.

**Questions:**

1. Do Algorithm 1 and 2 provide sublinear regret guarantee for one player when the other player might be an adversary?

---

> ### Author Rebuttal · Authors · 2023-08-07
>
> We thank the reviewer for the nice comments about our presentation. We next provide a point-by-point response to the reviewer's comments.
>
> **Comment:** The claim that the proposed dynamics is convergent and rational seems not rigorous since the smoothing bias  persists over time and no convergence to exact Nash equilirium or best response is really proved. One possible fix might be introducing a decreasing parameter  so that exact convergence can be shown. This might significantly slow down the convergence of other bias terms because of the exponential dependence on $\tau$. The proposed dynamics is thus very slow in terms of convergence to Nash equilirium and this is a weakness of the current work.
>
> **Response:** Please see the response to the common comments from the reviewers in the beginning of this page.
>
> **Comment:** Minor comments: Line 277,  $Q(s,a^i,a^{-i})$ has not been defined. Under Line 284: one term should be $R(s,a^i,a^{-i})$.
>
> **Response:** In Line 277, we use $Q$ as a dummy variable to introduce the notation. We will change the notation to avoid confusion in the next version. $R(a,a^i,a^{-i})$ is a typo, we will correct it in our next version.
>
> **Comment:** Do Algorithm 1 and 2 provide sublinear regret guarantee for one player when the other player might be an adversary?
>
> **Response:** For simplicity of illustration, consider the matrix-game setting. Our learning dynamics is closely related to the celebrated smoothed fictitious play for zero-sum games in the sense that their corresponding ODE is the same, which is the continuous smoothed best-response dynamics $\dot{\pi}^i=\sigma_\tau(R^i\pi^{-i})-\pi^i$, $i\in \{1,2\}$. It was shown in the existing literature
> (Benaïm and Faure 2013) that smoothed fictitious play (with a diminishing sequence of temperatures $\{\tau_k\}$) is consistent, which implies that the average regret goes to zero asymptotically. See (Benaïm and Faure 2013) Definition 1.1 for the definition of consistency and (Benaïm and Faure 2013) Theorem 1.8 for the result. Since our learning dynamics is a discrete and stochastic variant of the smoothed best-response dynamics, it is conceivable that our learning dynamics can also be consistent. Rigorously proving the result and explicitly characterizing the rate at which the regret goes to zero are interesting future directions.
>
> Benaïm, M., \& Faure, M. (2013). Consistency of vanishingly smooth fictitious play. Mathematics of Operations Research, 38(3), 437-450.

---

> > ### Comment · Reviewer_zACq · 2023-08-11
> > **Acknowledgment of Rebuttal**
> >
> > Thank you for the detailed reply! I have no further questions.

---

> > > ### Author Response · Authors · 2023-08-14
> > >
> > > We thank the reviewer for the feedback.

---

### Official Review · Reviewer_r3LS · 2023-07-07

**Soundness:** 3 good
**Presentation:** 3 good
**Contribution:** 3 good
**Rating:** 6
**Confidence:** 3

**Summary:**

Authors propose algorithms for learning the Nash equilibrium in two-player games and two-player stochastic games. The algorithm for games is effectively a single-time scale algorithm (Doubly smoothed best response dynamics). While , the algorithm for stochastic games (Double smoothed best response dynamics with value iteration) is a two-time scale approach. They show that these algorithm is independently implemented by each agent without any communication requirement and the rewards are based on actual payoffs obtained after each each action, rather than the full information setting. Also, they also show that the players reach a policy that is a best-response of the other players' (stationary) policies.



**Strengths:**

The aim to reach convergence in last-iterate in zero-sum games and stochastic zero-sum games through independent learning is a challenging problem and the authors provide not only a finite sample guarantee, but also in addition, that the learned policies are best-responses to the stationary policies of other agents and achieve "rational" learning (Bowling and Veloso ' 2001). Also, the assumption about existence of a joint policy that induces an irreducible and aperiodic Markov chain is relatively weaker than other existing assumptions, that assume any policy always induces such an irreducible and aperiodic Markov chain or any policy created by the algorithm's trajectory is uniformly geometrically ergodic. They use coupled Lyapunov drift inequalities to guarantee convergence and finite sample guarantees.

**Weaknesses:**

1) If the game has a non-interior Nash, do you still get convergence? Lemma D.1 seems to say that all strategies are played with a probability lower bounded by a certain value.
2) Is there any hope of getting a stronger probabilistic guarantee for at least the two-player zero-sum games setting? Essentially to understand the convergence of the iterates (in distribution or w.h.p)? instead of an expected sense as derived in this paper).
3) Authors state that prior two-time scale approaches might require implicit coordination amongst players, however, their algorithm for the stochastic games is also two-time scale and it would help if they could clarify how they are able to avoid this in the algorithm and how they make up for this in the analysis.

**Questions:**

See weaknesses.

**Limitations:**

They have mentioned the limitations in the paper.

---

> ### Author Rebuttal · Authors · 2023-08-07
>
> We thank the reviewer for the encouraging comments about our work. We next provide a point-by-point response to the reviewer's comments.
>
> **Comment:** If the game has a non-interior Nash, do you still get convergence? Lemma D.1 seems to say that all strategies are played with a probability lower bounded by a certain value.
>
>  **Response:** We want to clarify that, regardless of whether a Nash equilibrium is interior or not, our finite-sample bound always holds. However, the finite-sample bound in general does not imply asymptotic convergence to *zero*. Specifically, observe that in either Theorem 2.1 or Theorem 3.1, the last term on the right-hand side of the bound (which we call the smoothing bias in our paper) is asymptotically nonvanishing, and is proportional to $\tau$, which is the temperature used in defining the softmax operator. This term captures the error between the output of the algorithm and a Nash equilibrium because a Nash equilibrium can potentially be a pure strategy (i.e., a non-interior Nash) while our learned policies are always stochastic.
>
> **Comment:** Is there any hope of getting a stronger probabilistic guarantee for at least the two-player zero-sum games setting? Essentially to understand the convergence of the iterates (in distribution or w.h.p)? instead of an expected sense as derived in this paper).
>
>  **Response:** In zero-sum matrix games, the policies produced from our algorithm do enjoy mean-square convergence to the Nash distribution, denoted by $(\pi_\tau^1,\pi_\tau^2)$. Recall that the Nash distribution is the unique minimizer of the regularized Nash gap defined right before Corollary 2.2.1; see also (Leslie and Collins 2005) for the definition of the Nash distribution. The uniqueness part follows from the regularized Nash gap being a strongly convex function (thanks to the regularizer). Therefore, by the quadratic growth property of strongly convex functions, we have $\sum_{i=1,2} \lVert  \pi^{i}-\pi_{\tau}^{i} \rVert^2\leq c RNG(\pi^1,\pi^2) $ for some constant $c>0$, which, after combined with Corollary 2.1.1., provides the mean-square convergence of $(\pi_k^1,\pi_k^2)$.
>
> A mean-square bound implies a high probability bound via the Markov inequality (while the tail is polynomial instead of sub-Gaussian). Investigating whether a high probability bound (with sub-Gaussian or sub-exponential tail) is achievable or not is a future direction. Also by Markov inequality, mean-square convergence implies convergence in probability, which in turn implies convergence in distribution because the limit point $(\pi_\tau^1,\pi_\tau^2)$ is deterministic. We will include this result as a corollary in our next version.
>
> Leslie, D. S., & Collins, E. J. (2005). Individual Q-learning in normal form games. SIAM Journal on Control and Optimization, 44(2), 495-514.
>
> **Comment:** Authors state that prior two-time scale approaches might require implicit coordination amongst players, however, their algorithm for the stochastic games is also two-time scale and it would help if they could clarify how they are able to avoid this in the algorithm and how they make up for this in the analysis.
>
> **Response:** To clarify, in Lines 620 - 628, we say that the proposed learning dynamics in these results are two time-scale in the sense that there is a time-scale separation between the two players. Specifically, one player is updating much faster than the other, making the learning dynamics *asymmetric* between the two, which also indicates implicit coordination. In contrast, our learning dynamics only require the update of each player's policy to be slower than the update of their $q$-functions, but crucially we do not assume a time-scale separation between the players, making our learning dynamics *symmetric*. We will clarify this point in our next version.

---

> > ### Comment · Reviewer_r3LS · 2023-08-14
> > **Thank you for the rebuttal**
> >
> > I thank the authors for their rebuttal and I understand this is mainly a theoretical study, but did the authors try their algorithms against well known algorithms (MWU, OMWU) in simple games such as Rock, Paper, Scissors etc? I think it might be useful to shed light on the smoothing bias, temperature and how this affects the convergence. Other than that my initial questions have been clarified.

---

> > > ### Author Response · Authors · 2023-08-14
> > >
> > > We thank the reviewer for the great suggestion. We will include numerical simulations on Benchmark examples (such as the RPS game suggested by the reviewer) in our next version and compare our learning dynamics with existing algorithms such as MWU and OMWU. Moreover, we will also investigate the dependence on the temperature $\tau$ (which determines the smoothing bias) in our numerical simulations.

---

### Official Review · Reviewer_Tp2P · 2023-07-13

**Soundness:** 4 excellent
**Presentation:** 3 good
**Contribution:** 4 excellent
**Rating:** 7
**Confidence:** 3

**Summary:**

The authors focus on the problem of finite-sample convergence analysis of independent best-response-type learning dynamics in two-player zero-sum stochastic games. The dynamics are payoff based and stimulate the Shapley operator (that is known to be a contractive). Under the assumption that for any pair of policies, the induced Markov chain is irreducible and aperiodic, it is shown that in expectation the Nash gap is at most epsilon after 1/eps steps of the algorithm. Finally the authors provide results rationality type results (i.e., on the regret when the one player follows the dynamics and the other plays a stationary policy).

**Strengths:**

I think that the result of 1/eps rate is quite interesting and surprising (I would expect a rate of 1/eps^2). Moreover the techniques seem highly non-trivial (design of a novel Lyapunov type function).

**Weaknesses:**

Sometimes the write-up is not self-contained, but this is because of the limited space.

**Questions:**

Can you please explain if your results carry over for other settings like Markov potential Games?

---

> ### Author Rebuttal · Authors · 2023-08-07
>
> We thank the reviewer for the encouraging comments about our work.
>
> **Question:** Can you please explain if your results carry over for other settings like Markov potential Games?
>
> **Response:** For potential games, it was shown in Swenson et al. (2018) that continuous best-response dynamics provably converges to a pure-strategy Nash equilibrium with an exponential rate. Since our learning dynamics is a discrete, smoothed, and stochastic variant of the best-response dynamics, it should converge for potential games when the stepsizes are appropriately chosen, while the analysis and the rate of convergence might be largely different. Rigorously proving this result could be an interesting future direction.
>
> In the Markovian setting, recall that the outer loop of our learning dynamics is an approximation of minimax value iteration, which works because minimax value iteration converges due to the contraction property of the minimax Bellman operator. For Markov potential games (MPGs), it is unclear if value iteration leads to convergence. However, since there exists a potential function for MPGs, existing results mostly use the gradient-based method, which works because the potential function naturally serves as a Lyapunov function. See for example Ding et al. (2022); Zhang et al. (2022).
>
> Swenson, B., Murray, R., \& Kar, S. (2018). On best-response dynamics in potential games. SIAM Journal on Control and Optimization, 56(4), 2734-2767.
>
> Ding, D., Wei, C. Y., Zhang, K., \& Jovanovic, M. (2022, June). Independent policy gradient for large-scale markov potential games: Sharper rates, function approximation, and game-agnostic convergence. In International Conference on Machine Learning (pp. 5166-5220). PMLR.
>
> Zhang, R., Mei, J., Dai, B., Schuurmans, D., \& Li, N. (2022). On the global convergence rates of decentralized softmax gradient play in markov potential games. Advances in Neural Information Processing Systems, 35, 1923-1935.

---

> > ### Comment · Reviewer_Tp2P · 2023-08-18
> >
> > Thank you for your response. I have no further questions.

---

> > > ### Author Response · Authors · 2023-08-19
> > >
> > > We thank the reviewer for the feedback.

---

### Author Rebuttal · Authors · 2023-08-08

We thank all reviewers for their time and effort in reviewing this work. We here provide the response to the common comments raised by the reviewers. The point-by-point response to each reviewer's individual comments is provided under the corresponding review.

**Common Comments:**

*Major comment from Reviewer zACq:* The claim that the proposed dynamics is convergent and rational seems not rigorous since ...

*The (3)rd comment from Reviewer bFVC:* The bound for Nash gap is somewhat weak ...

**Our Response:** We agree with the reviewers that the convergence bound is not asymptotically vanishing because of the presence of the smoothing bias, which will be made clear in the next version. In the next version, we will also explicitly choose the temperature $\tau$ to provide an overall (possibly slower) rate of convergence to a Nash equilibrium. The consequence of having slow convergence to a Nash equilibrium is due to having a constant that is exponential in $\tau$ in the bound, which dominates other terms that are polynomial in $\tau$. We have acknowledged this as a weakness of this work in the conclusion section.

In general, we believe that achieving a sharp (polynomial) last-iterate convergence to a Nash equilibrium (without regularization) with best-response learning dynamics is a much more challenging problem. Recall that for fictitious play (which is the simplest best-response dynamics, and inspires our algorithm design), while Samuel Karlin conjectured an $\mathcal{O}(1/k^{1/2})$ rate of convergence in 1959, the state-of-the-art provable rate of convergence is $\mathcal{O}(1/k^{1/(m+n-2)})$ (where $n=|\mathcal{A}^1|$ and $m=|\mathcal{A}^2|$) provided in Shapiro, H. N. (1958). Note that this provable rate, while being polynomial, is *dimension* dependent, and hence can be arbitrarily slow when $m$ and $n$ are large. Therefore, establishing dimension-independent polynomial rate of convergence for fictitious play remains as an *open problem* for more than $70$ years. Compared with fictitious play, our algorithm is payoff-based (more challenging than fictitious play where the opponent's actions can be observed), and achieves $1/K$ rate of convergence to the regularized Nash, while achieving a worse rate of convergence to the true Nash. This, to some extent, also reflects the challenge in establishing dimension-independent polynomial rate of convergence for best-response type learning dynamics.

That being said, this work provides **the first last-iterate finite-sample analysis of best-response independent learning dynamics** in the literature. Investigating the possibility of achieving an improved rate of convergence is a compelling future direction.

Shapiro, H. N. (1958). Note on a computation method in the theory of games. Communications on Pure and Applied Mathematics, 11(4), 587-593

---

### Comment · Area_Chair_rHGt · 2023-08-21
**Quick question**

Dear authors,

Could you clarify what is the difference between Algorithm 1 in your paper and the 2003 algorithm of Leslie and Collins, "Convergent multiple-times-scales reinforcement learning algorithms in normal form games", The Annals of Applied Probability 13 (2003), no. 4, 1231–1251 [Eq. (6) in p. 8]?

I understand that you use a different tuning for $\alpha_k$ and $\beta_k$ but, given that Leslie and Collins establish convergence with probability $1$ in zero-sum games (not just in expectation), this raises the question why the standard choice $\beta_k/\alpha_k \to 0$ would not be more appropriate in your setting as well.

Regards,

The AC

---

> ### Author Response · Authors · 2023-08-21
> **Thank You for the Comments**
>
> Dear AC,
>
> **The Difference:** We thank the AC for pointing out Leslie and Collins (2003), which we will include in our literature review. The main difference between Algorithm 1 of this work and the one in Leslie and Collins (2003) is that our learning dynamics is *single time-scale* while the one in Leslie and Collins (2003) is *two time-scale* as they require $\lambda_n/\mu_n\rightarrow 0$. Technically, the fact that we do not have a time-scale separation between the two sets of iterates presents major technical challenges in establishing the finite-sample bound, which we overcome with our coupled Lyapunov approach. In terms of results, Leslie and Collins (2003) consider only the asymptotic convergence in matrix games while we establish the rate of convergence (*which is more informative and challenging compared with asymptotic convergence*) in both matrix and stochastic games.
>
> **Our Results Imply the Mean-Square Convergence of the Last Policy Iterate** Thanks to reviewer r3LS’s second comment, we realized that algorithm 1 does enjoy a stronger convergence guarantee (not just in expectation for the regularized Nash gap). Specifically, recall that the Nash distribution (denoted by $(\pi_\tau^1,\pi_\tau^2)$) is the unique minimizer of the regularized Nash gap defined right before Corollary 2.2.1, where the uniqueness part follows from the regularized Nash gap being a strongly convex function. Therefore, by the quadratic growth property of strongly convex functions, we have $\sum_{i=1,2}\lVert \pi_k^i-\pi_\tau^i \rVert_2^2\leq c RNG(\pi_k^1,\pi_k^2)$ for some constant $c>0$, which, after combining with Corollary 2.1.1., provides the mean-square convergence of the last iterate $(\pi_k^1,\pi_k^2)$. We will include this result as a corollary in our next version.
>
> **Single Time-Scale vs Two Time-Scale** The stepsize condition imposed in Leslie and Collins (2003) is standard for establishing the almost sure convergence (Borkar 2009) while our goal is to characterize the rate of convergence, where the stepsize choice might be different to obtain the best possible rate. While it is theoretically unclear if there is a fundamental gap between the convergence rates of two time-scale stochastic iterative algorithms (of which the algorithm in Leslie and Collins (2003) is a special case) and their single time-scale counterparts, in the existing literature for finite-sample analysis, the established upper bounds for two time-scale algorithms are mostly **sub-optimal**. For example, the convergence rate of two time-scale SGD established in Zeng et al (2022) was $k^{-1/4}$ while single time-scale SGD achieves $k^{-1/2}$. The convergence rate of the two time-scale natural actor-critic algorithm established in Khodadadian et al. (2022) was $k^{-1/6}$ while the lower bound for RL is $k^{-1/2}$, which is achieved with single time-scale algorithms such as $Q$-learning (Li et al. 2020). To obtain the best possible rate of convergence (cf. the $1/k$ rate convergence in Corollary 2.1.1 of Algorithm 1), we adopt a single time-scale framework.
>
> **Almost Sure Convergence vs. Finite-Sample Analysis** Asymptotic convergence in general does *not* imply any rate of convergence. In addition, the techniques and conditions to establish asymptotic convergence and finite-sample bound are fundamentally different. In particular, consider the requirements on choosing the stepsizes. Asymptotic convergence typically requires the stepsizes to be squared summable (Robbins and Monro 1951), recent results in Bhandari et al.( 2018); Chen et al (2022) show that mean-square finite-sample bound can be established with stepsizes of the form $1/k^z$ for all $z\in (0,1]$. Note that when $z\in (0,1/2)$, the stepsizes are not squared summable.
>
>
> Borkar, V. S. (2009). Stochastic approximation: a dynamical systems viewpoint (Vol. 48). Springer.
>
> Zeng, S., Doan, T. T., & Romberg, J. (2021). A two-time-scale stochastic optimization framework with applications in control and reinforcement learning. arXiv preprint arXiv:2109.14756.
>
> Bhandari, J., Russo, D., & Singal, R. (2018, July). A finite time analysis of temporal difference learning with linear function approximation. In Conference on learning theory (pp. 1691-1692). PMLR.
>
> Chen, Z., Zhang, S., Doan, T. T., Clarke, J. P., & Maguluri, S. T. (2022). Finite-sample analysis of nonlinear stochastic approximation with applications in reinforcement learning. Automatica, 146, 110623.
>
> Khodadadian, S., Doan, T. T., Romberg, J., & Maguluri, S. T. (2022). Finite sample analysis of two-time-scale natural actor-critic algorithm. IEEE Transactions on Automatic Control.
>
> Li, G., Wei, Y., Chi, Y., Gu, Y., & Chen, Y. (2020). Sample complexity of asynchronous Q-learning: Sharper analysis and variance reduction. Advances in neural information processing systems, 33, 7031-7043.
>
> Best,
>
> Authors

---

> > ### Comment · Area_Chair_rHGt · 2023-08-21
> >
> > Dear authors,
> >
> > Thanks for confirming that the algorithm is the same as that of Leslie and Collins (LC in the sequel), only with a different schedule for $\alpha_k$ and $\beta_k$.
> >
> > For the rest, my question still stands: the LC schedule guarantees almost sure convergence, but its convergence rate is not known; your choice of schedule comes with rates, but it does not guarantee almost sure convergence (only in expectation). Given that the LC algorithm is quite classical by now, it is natural to ask whether you can get a rate of convergence under the LC schedule - or, rather, for general $\alpha_k$, $\beta_k$ - before settling on a particular schedule.
> >
> > On this point, the papers by Zeng et al. and Khodadadian et al. are not particularly relevant because they concern completely different settings and algorithms - for example, the time-scales in Khodadadian involve an explicit exploration step, which is well-known to slow down convergence. There are other papers in the game-theoretic literature where different time-scales *do* lead to better rates - for example, Hsieh et al., "*Explore aggressively, update conservatively: Stochastic extragradient methods with variable stepsize scaling*", NeurIPS 2020 - but it is impossible to make a meaningful comparison across different settings, so let us focus on the precise question at hand.
> >
> > Regards,
> >
> > The AC
> >
> > PS: This point is not particularly relevant in the discussion (so I'm not including it above), but square summability is not required to establish asymptotic convergence, see e.g., Propositions 4.2 and 4.4 in Benaïm, "*Dynamics of Stochastic Approximation Algorithms*" (1999). In the setting of your paper, you could have step-sizes of the form $1/k^z$ for all $z\in(0,1]$, or even of the form $1/(\log k)^{1+\epsilon}$ for some $\epsilon > 0$.

---

> > > ### Author Response · Authors · 2023-08-21
> > >
> > > Dear AC,
> > >
> > > Thank the AC for the prompt feedback.
> > >
> > > We want to clarify that the algorithm is the same as the one in LC up to a time-scale separation *only* in the matrix-game setting. The fact that we only require the stepsizes to be different up to a multiplicative constant enables us to run our algorithm with a single set of stepsizes. Furthermore, our main focus is on the *stochastic-game* case as emphasized in our title. This requires the additional value iteration component in the algorithm, which is not present in the matrix-game setting, and presents technical challenges in handling the time-inhomogenous Markovian noise.
> > >
> > > As for the rate of convergence, in our proof of Theorem 2.1 and Corollary 2.1.1 (See Appendix D), we are, in fact, keeping the stepsizes as general as possible (i.e., using $\alpha_k$ and $\beta_k$ without specifying the decay rate) up until the last inequality of Line 1294, at which point imposing a time-scale separation clearly results in a **slower** rate of convergence. Specifically, in the last inequality of Line 1294, we establish the overall negative drift inequality of the combined Lyapunov function. Suppose that we impose a time-scale separation, which means that the stepsize ratio $c_{\alpha,\beta}$ is not constant but decays with $k$. Since $c_{\alpha,\beta}$ appears in the denominator in the last term on the right-hand side of the last inequality in Line 1294, the resulting rate from our proof technique will be slower compared with our current result without time-scale separation.
> > >
> > > Regards,
> > >
> > > Authors

---

> > > > ### Comment · Area_Chair_rHGt · 2023-08-21
> > > >
> > > > Dear authors,
> > > >
> > > > > Furthermore, our main focus is on the stochastic-game case as emphasized in our title.
> > > >
> > > > There is no confusion on that - I am focusing on the normal form setting because this is where the overlap with LC lies.
> > > >
> > > > > Imposing a time-scale separation [in the last inequality of Line 1294] clearly results in a slower rate of convergence
> > > >
> > > > The trade-off in L1294 is clear: the two coefficients in the parentheses have to scale at the same rate in order to get sufficient decrease for your choice of Lyapunov function $V_R + \|\mathcal{R}\pi - q\|^2$. However, the definition of $V_R$ treats the Nash gap for the underlying game and the entropic smoothing differently: the standard Lyapunov function for this type of (smoothed) best reply algorithms would be $\sum_{i=1,2} \max_{\hat\mu^i}(\hat\mu^i - \mu^i) [ R^i \mu^{-i} - \tau \nabla\nu(\mu^i) ]$, i.e., the Nash gap for the modified game with payoffs $\mathcal{R} - \tau\nu$ (note that the Nash equilibria of this modified game are precisely the Nash distributions / logit equilibria of the original game). [Caveat: the sign of $\nu$ might be off depending on whether you include a "$-$" in the definition of the entropy or not; I'm not sure which convention you use in the appendix] This choice does not involve the two terms that you control separately in Lemmas D.3 and D.4, so it is not clear if the common scaling of $\alpha_k$ and $\beta_k$ is really necessary or if it is an artifact of the analysis.
> > > >
> > > > At any rate, I thank you for your replies, and I will be sure to reach out if any further points requiring clarification come up during the committee discussion stage.
> > > >
> > > > Regards,
> > > >
> > > > The AC

---

> > > > > ### Author Response · Authors · 2023-08-22
> > > > >
> > > > > Dear AC,
> > > > >
> > > > > We thank the AC for pointing us to the new Lyapunov function. It is interesting to investigate whether using this Lyapunov function would result in an improved rate of convergence. On the other hand, our regularized Nash gap, i.e., $V_R$, is also a natural choice of the Lyapunov function, which was previously used to study the smoothed best-response dynamics in Hofbauer and Hopkins (2005) (see their Theorem 3.2).
> > > > >
> > > > > We will include a clear discussion (in words and in math) about the single time-scale vs two time-scale comparison and our choice of the Lyapunov functions in our next version, which will certainly improve this work.
> > > > >
> > > > > Hofbauer, J., & Hopkins, E. (2005). Learning in perturbed asymmetric games. Games and Economic Behavior, 52(1), 133-152.
> > > > >
> > > > > Best,
> > > > >
> > > > > The Authors

---

### Decision · Program_Chairs · 2023-09-21

**Decision:**

Accept (poster)

**Comment:**

This paper examines examines the problem of payoff-based equilibrium learning in two-player zero-sum games. The authors analyze two settings: finite games (which is intended as a "warm-up"), and then, stochastic games. In both settings, they provide a finite-sample convergence rate analysis assuming that players only observe the rewards that they observed in-game.

This paper was discussed extensively by the reviewers, area chairs and senior area chairs. Initially, the reviewers were reluctant and raised a number of concerns, several of which were addressed by the authors' rebuttal, ultimately leading to a positive reassessment.

At the same time, during the discussion phase, the following issues emerged:
- Contrary to what the authors state, the idea of what they call the "doubly smoothed best response" (DSBR) dynamics has been around in the literature for some twenty years now (from the work of Leslie and Collins, 2003). In fact, the basic idea of this algorithm is well-known in the field (see e.g., the classical treatments of Bertsekas & Tsitsiklis, 1996, and Sutton & Barto, 1998), so this cannot be claimed as a contribution of the paper.
- Regarding the novelty of the analysis, the authors stated in the discussion phase that they employed the Lyapunov function of Hofbauer and Hopkins (2005). However, they make no mention of this in the paper, and instead suggest that "there is no off-the-shelf Lyapunov function" so they had to design it from scratch.

Since the base algorithm used by the authors has been around for so long (and the same is true for the core elements of the authors' Lyapunov analysis), the paper has to be thoroughly revised to clarify that these algorithmic developments are not due to the authors (this includes the abstract, the introduction, the "design" paragraphs in Section 1.1, the relevant parts of Section 1.2 (where no mention is made of the work of Hofbauer & Hopkins), Section 1.3 (the algorithm they use has been used extensively, in both games and RL), the motivation they provide in Section 2.1, the name of the algorithm, etc. The extension of the algorithm to the stochastic game is likewise natural and straightforward, so the corresponding parts should be toned down as well.

These concerns would require extensive revision work to be addressed, definitely more than what would be considered a "minor revision" (i.e., a revision not requiring a second round of reviewing). At the same time, the authors' finite-sample analysis (especially in the setting of stochastic games) does appear novel, and provides a step forward in our understanding of equilibrium learning in zero-sum (stochastic) games.

Thus, taking everything into account (and despite the above limitations), the paper's analysis and results provide a number of fruitful insights into an active research field. In view of this, the committee's view is that the merits of the paper outweigh its flaws, hence my "accept" recommendation. At the same time, I should stress that this decision is made with the understanding that the authors commit to revise their paper as indicated above for the camera-ready phase. With this proviso, I am happy to recommend acceptance.

**References for the authors:**
1. Josef Hofbauer and Ed Hopkins, *Learning in perturbed asymmetric games,* Games and Economic Behavior 52 (2005), no. 1, 133–152.
2. David S. Leslie and E. J. Collins, *Convergent multiple-timescales reinforcement learning algorithms in normal form games,* The Annals of Applied Probability 13 (2003), no. 4, 1231–1251.
3. Richard S. Sutton and A. G. Barto, *Reinforcement learning: An introduction,* MIT Press, Cambridge, MA, 1998.